# Early intermittent hyperlipidaemia alters tissue macrophages to fuel atherosclerosis

Minoru Takaoka[1], Xiaohui Zhao[1,22], Hwee Ying Lim[2,3,22], Costan G. Magnussen[4,5,6,22], Owen Ang[2,3], Nadine Suffee[7], Patricia R. Schrank[8], Wei Siong Ong[2,3], Dimitrios Tsiantoulas[1,9], Felix Sommer[10], Sarajo K. Mohanta[11], James Harrison[1], Yaxing Meng[6], Ludivine Laurans[7], Feitong Wu[6], Yuning Lu[1], Leanne Masters[1], Stephen A. Newland[1], Laura Denti[12], Mingyang Hong[11], Mouna Chajadine[7], Markus Juonala[13,14], Juhani S. Koskinen[4,5,14,15], Mika Kähönen[16,17,18], Katja Pahkala[4,5], Suvi P. Rovio[4,5], Juha Mykkänen[4,5], Russell Thomson[6,19], Tsuneyasu Kaisho[20], Andreas J. R. Habenicht[11], Marc Clement[1], Alain Tedgui[7], Hafid Ait-Oufella[7], Tian X. Zhao[1], Meritxell Nus[1], Christiana Ruhrberg[12], Soraya Taleb[7], Jesse W. Williams[8], Olli T. Raitakari[4,5,21,22], Véronique Angeli[2,3,22] & Ziad Mallat[1,7 ✉]

Hyperlipidaemia is a major risk factor of atherosclerotic cardiovascular disease (ASCVD). Risk of cardiovascular events depends on cumulative lifetime exposure to low-density lipoprotein cholesterol (LDL-C) and, independently, on the time course of exposure to LDL-C, with early exposure being associated with a higher risk[1]. Furthermore, LDL-C fluctuations are associated with ASCVD outcomes[2–4]. However, the precise mechanisms behind this increased ASCVD risk are not understood. Here we find that early intermittent feeding of mice on a high-cholesterol Western-type diet (WD) accelerates atherosclerosis compared with late continuous exposure to the WD, despite similar cumulative circulating LDL-C levels. We find that early intermittent hyperlipidaemia alters the number and homeostatic phenotype of resident-like arterial macrophages. Macrophage genes with altered expression are enriched for genes linked to human ASCVD in genome-wide association studies. We show that LYVE1[+] resident macrophages are atheroprotective, and identify biological pathways related to actin filament organization, of which alteration accelerates atherosclerosis. Using the Young Finns Study, we show that exposure to cholesterol early in life is significantly associated with the incidence and size of carotid atherosclerotic plaques in mid-adulthood. In summary, our results identify early intermittent exposure to cholesterol as a strong determinant of accelerated atherosclerosis, highlighting the importance of optimal control of hyperlipidaemia early in life, and providing insights into the underlying biological mechanisms. This knowledge will be essential to designing effective therapeutic strategies to combat ASCVD.

A large body of evidence supports a causal role of cholesterol in the development of atherosclerosis, in part through activation of the NLRP3–IL-1β pathway[5,6]. Despite these advances, the precise mechanisms that are responsible for the development of inflammatory plaques in response to cholesterol overload are not well understood.

Besides the circulating concentration of LDL-C, exposure duration and therefore the area under the LDL-C versus age curve is an independent predictor of incident ASCVD events[1]. Notably, the time course of area of cholesterol accumulation is a strong independent determinant of ASCVD events, with early age accumulation being associated with a greater cardiovascular disease risk[1]. Furthermore, periodic variations in circulating cholesterol levels are associated with incident ASCVD, independently of mean LDL-C concentration[2,3], even under statin treatment[4]. The questions therefore regard (1) how early and intermittent variations in serum cholesterol increase ASCVD risk; and (2) how this

can be modelled to understand the underlying mechanisms. Here we have begun to answer these important questions.

## Early intermittent WD and atherosclerosis

Animal models of atherosclerosis rely on the induction of hyperlipidaemia[7,8], the extent and size of atherosclerotic plaques being proportional to the achieved levels of circulating cholesterol[7–10]. Since Ignatowski developed the first animal model of diet-induced atherosclerosis in 1908, these models have mostly relied on inducing sustained elevations in circulating cholesterol levels for a defined period of time until the experiment is terminated for analysis of plaque size and characteristics. As such, these models address the effect of 'late' hyperlipidaemia (that is, late during lifetime) on the development of atherosclerosis, but ignore the potential effect of lifelong variable exposure to circulating

cholesterol on disease development and progression, as is the case in humans.

To address these deficiencies, we designed an experimental protocol of diet-induced atherosclerosis that accounts for these parameters. In this model of early intermittent hyperlipidaemia, the area under the cholesterol–time curve (the cumulative cholesterol exposure) is unchanged when compared to the current models of late sustained hyperlipidaemia, but is spread over the lifetime of the animal (Fig. 1a and Extended Data Fig. 1a).

In a first series of experiments, we subjected LDL-receptor-deficient ($Ldlr^{-/-}$) male mice to 6 weeks of late continuous Western-type cholesterol-rich diet (cWD) versus 6 weeks of early intermittent WD (iWD) (Fig. 1a and Extended Data Fig. 1a). We measured the plasma cholesterol levels repetitively during the experiment (Extended Data Fig. 1b–d) and calculated the averaged cumulated plasma cholesterol levels over the duration of the experiment (Fig. 1b). The overall cholesterol load (cholesterol × time) was similar between the two groups of mice. The weight, heart rate, blood pressure and plasma corticosterone levels were also similar between the two groups (Extended Data Fig. 1e–h). On the basis of the paradigm that cumulative exposure to cholesterol is the major determinant of atherosclerosis, we should have detected atherosclerotic lesions of similar sizes in the two groups of mice. However, we were surprised by the substantial increase of plaque size in mice subjected to the iWD compared with the cWD (Fig. 1c,d). The results were reproducible in female mice (Extended Data Fig. 2a,b), although the fold increase (1.3 to 2.5) in lesion size was more variable than in male mice. Analysis of the lesion composition (Extended Data Fig. 2c–g) revealed a pro-inflammatory phenotype with increased accumulation of MOMA2+ macrophages and CD3+ T cells, and large necrotic cores. Despite an increase in the αSMA+ area, suggesting higher smooth muscle cell plaque content in the iWD group, collagen accumulation as measured by Sirius Red staining was not different between the two groups, suggesting altered plaque healing in iWD mice (Extended Data Fig. 2f,g). We tested several periods of iWD and cWD and always obtained an acceleration of atherosclerosis in iWD mice (Extended Data Fig. 3a,b). $Ldlr^{-/-}$ mice subjected to 12 weeks of iWD still showed substantially larger atherosclerotic plaques compared with mice that were subjected to 12 weeks of late cWD (Fig. 1e,f), with very large necrotic cores associated with defective efferocytosis and increased accumulation of apoptotic debris (Extended Data Fig. 3c–j). The lesion size tended to be smaller (although not statistically different) when iWD was started at 22 weeks of age compared with at 6 weeks of age (Extended Data Fig. 3k–l). In summary, early exposure to WD is substantially more pro-atherogenic than late exposure.

WD feeding affects the composition of the gut microbiota, which in turn may affect the development of atherosclerosis[11]. We assessed the gut microbiota composition using 16S rRNA sequencing of the faeces after both 3 and 6 weeks of iWD and cWD (Extended Data Fig. 4a). The gut microbiota composition was similar between the iWD and cWD groups after 3 weeks of WD, but differed slightly between the two groups after 6 weeks (Extended Data Fig. 4a–c). We therefore designed an experiment to deplete the gut microbiota with oral antibiotics during the last 3 weeks of iWD (Extended Data Fig. 4d). Antibiotic treatment significantly depleted the gut microbiota (Extended Data Fig. 4e–g). We found that atherosclerosis was still accelerated in iWD versus cWD under antibiotics (Fig. 1g), although to a lesser degree than in the absence of antibiotics, which may in part be due to lower levels of plasma cholesterol in the iWD group compared with the cWD group under antibiotics (Extended Data Fig. 4h). Together, our results suggest a limited effect of gut microbiota on iWD-induced acceleration of atherosclerosis.

We also examined the role of adaptive immunity and found that iWD-induced acceleration of atherosclerosis still occurred in $Ldlr^{-/-}Rag2^{-/-}$ mice (deficient in T and B cells) (Fig. 1h), despite similar circulating cholesterol levels (Extended Data Fig. 4i). Although the roles of specific T and B cell subsets require additional studies, our

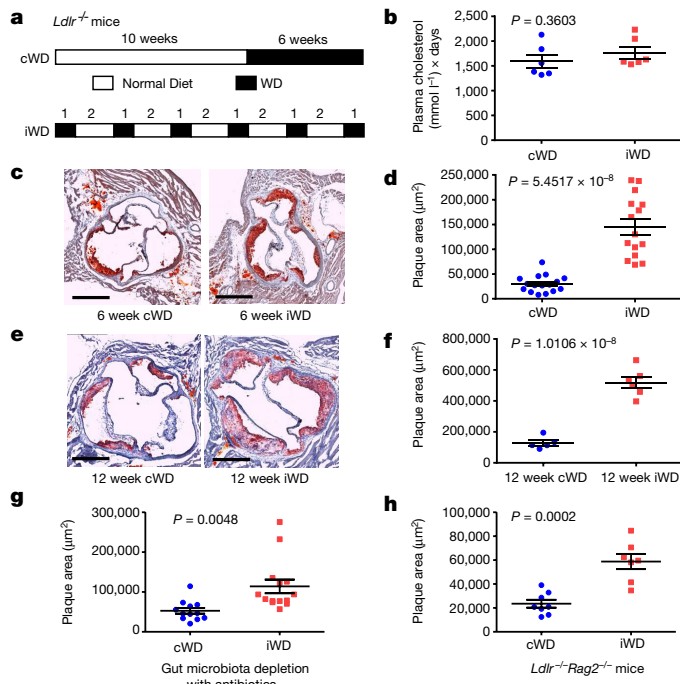

**Fig. 1 | Early intermittent hyperlipidaemia accelerates atherosclerosis in mice. a**, The experimental set-up. LDL receptor-deficient ($Ldlr^{-/-}$) male mice on a normal diet were fed a WD for 6 weeks either continuously (cWD) or intermittently (iWD). **b**, Calculation of the estimated area under the curve of plasma cholesterol levels over the whole period of the experiments. $n = 6$ per group. **c**, Representative photomicrographs of Oil Red O staining of the aortic sinus in the two groups of mice at the end of the experiment. Scale bars, 500 μm. **d**, The mean atherosclerotic plaque size in the aortic sinus of the two groups of mice. $n = 16$ (cWD) and $n = 15$ (iWD). **e**,**f**, $Ldlr^{-/-}$ male mice on a normal diet were put under a cWD or iWD diet for 12 weeks. **e**, Representative photomicrographs of Oil Red O staining of the aortic sinus in the two groups of mice et the end of the experiment. Scale bars, 500 μm. **f**, The mean atherosclerotic plaque size in the aortic sinus of the two groups of mice. $n = 5$ (cWD) and $n = 6$ (iWD). **g**, $Ldlr^{-/-}$ male mice on a normal diet were put on a cWD versus iWD diet for 6 weeks. To deplete gut microbiota, the two groups of mice were treated with oral antibiotics (Methods) starting from week 9 until the end of the experiment. The mean atherosclerotic plaque size in the aortic sinus of the two groups of mice at the end of the experiment is shown. $n = 12$ (cWD) and $n = 14$ iWD. **h**, $Ldlr^{-/-}Rag2^{-/-}$ male mice on a normal diet were put on a cWD versus iWD diet for 6 weeks. The mean atherosclerotic plaque size in the aortic sinus of the two groups of mice at the end of the experiment is shown. $n = 8$ (cWD) and $n = 7$ (iWD). Data are mean ± s.e.m. Statistical analysis was performed using two-tailed unpaired $t$-tests (**d** and **f**–**h**). $P$ values are indicated on the graphs.

results suggest a critical role of non-adaptive immune responses in this process.

## iWD reshapes arterial macrophages

We conducted a series of additional experiments and found that the atherosclerotic lesion size was not significantly different between the iWD and cWD groups after 3 weeks of WD (Extended Data Fig. 5a,b). We therefore selected this timepoint to perform RNA-sequencing (RNA-seq) analysis of aortae from both groups of mice with the aim of identifying causal processes involved in the subsequent acceleration of atherosclerosis in iWD mice. We focussed on macrophages, given the prominent role of these innate immune cells in atherosclerosis[12]. We conducted two separate experiments with nine mice per group and analysed the data together with a total of six separate pools (three mice each) per group. In total, 746 genes were differentially expressed (DEGs) in aortic macrophages between iWD- and cWD-fed mice (Fig. 2a,b and

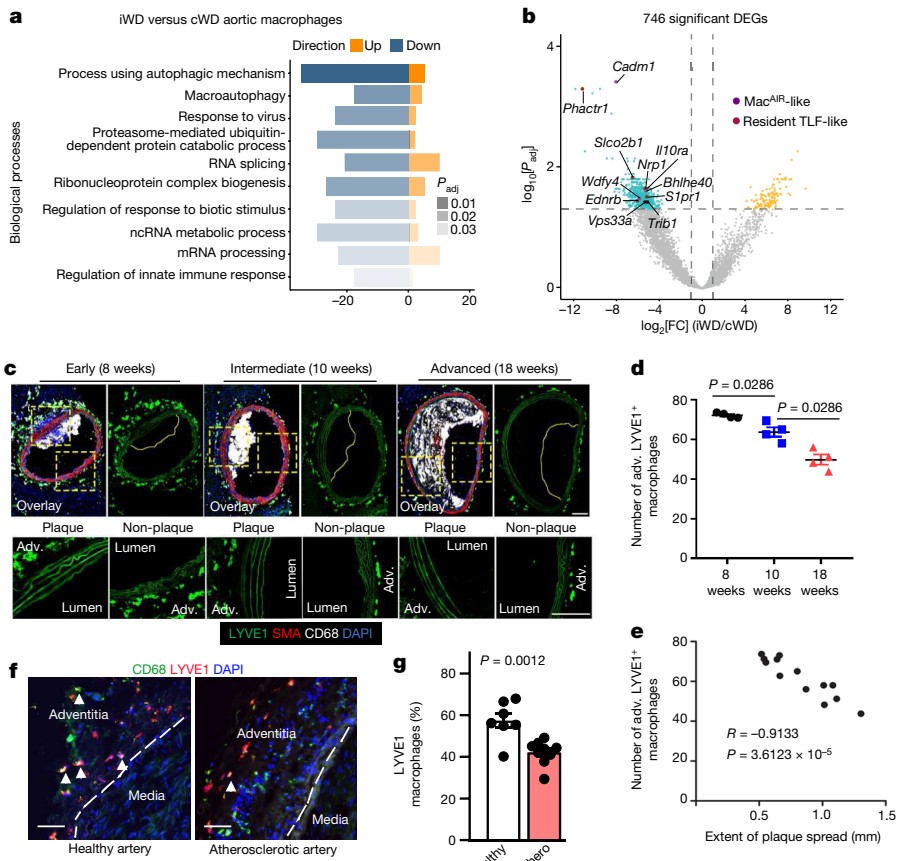

**Fig. 2 | Early intermittent hyperlipidaemia alters resident-like arterial macrophages.** *Ldlr*[−/−] male mice were put on 3 weeks of cWD versus 3 weeks of iWD (Extended Data Fig. 5a,b). The aortas (*n* = 6 separated pools of 3 mice each per group) were collected and digested for flow cytometry cell sorting of aortic macrophages and then analysed using bulk RNA-seq (Methods). **a**, Ten selected GO pathways (biological process) corresponding to the 746 significant DEGs between the iWD and cWD groups (the full list is shown in Supplementary Table 2). Blue and orange colour represents overall gene direction. Grey shading shows the adjusted *P* value (*P*$_{adj}$), calculated using one-sided Fisher exact tests followed by adjustment using the Benjamini–Hochberg procedure. **b**, Volcano plot for DEGs; yellow/cyan colour highlights upregulated genes in iWD/cWD (*P*$_{adj}$ < 0.05; two-sided Wald test with Benjamini–Hochberg adjustment) and |log$_2$[fold change]| ≥ 1, and grey colour displays non-significant DEGs. Some significant DEGs related to resident/TLF-like (red) or Mac$^{AIR}$-like (violet) macrophages are highlighted (a full list is provided in Supplementary Table 1).

**c**–**e**, *Apoe*[−/−] mice at 6 weeks of age were put on a cWD diet for 8, 10 or 18 weeks to develop early-stage, intermediate or advanced atherosclerosis. Atherosclerotic lesions of the innominate and brachiocephalic arteries were stained with DAPI (nuclei), and antibodies against LYVE1, CD68 and smooth muscle alpha-actin (SMA). **c**, Representative photomicrographs are shown. Scale bars, 100 μm. **d**,**e**, Quantification (mean ± s.e.m.) of adventitial (adv.) LYVE1$^+$CD68$^+$ macrophages (*n* = 4 per group; two-tailed Mann–Whitney *U*-test) (**d**) and correlation with the atherosclerotic plaque size (*n* = 13) (**e**). *R* is the Spearman correlation coefficient (two-tailed). **f**,**g**. Tissue sections of healthy (*n* = 7 individuals) and atherosclerotic (athero.; *n* = 9 individuals) (Methods) arteries were stained with DAPI (nuclei) and antibodies against LYVE1 and CD68. **f**, Representative photomicrographs. Scale bars, 30 μm. **g**, The percentages of adventitial LYVE1$^+$CD68$^+$ macrophages among adventitial CD68$^+$ macrophages. Statistical analysis was performed using two-tailed unpaired *t*-tests.

Supplementary Table 1). Gene Ontology (GO) term enrichment identified reduced (macro)autophagy as a potential major causal process (Fig. 2a and Supplementary Table 2), which fits with the known role of autophagy in limiting plaque inflammation, necrosis and progression in mice[13]. Notably, this defective autophagy pathway includes several autophagy-related genes (for example, *Atg12* and *Wdfy4*, which operate in the *Wdfy3* pathway[14], and *Vps33a*; Fig. 2b and Supplementary Table 1) recently identified as top-ranked regulators of macrophage efferocytosis[14]. This is consistent with the increased accumulation of apoptotic debris and the larger necrotic cores in plaques of iWD-fed mice (Extended Data Fig. 3), and the known role of impaired macrophage efferocytosis in promoting the formation of large necrotic and inflammatory atherosclerotic lesions[15].

## iWD alters resident-like macrophages

Notably, we found that most of the highly significant macrophage DEGs (Fig. 2b and Supplementary Table 1) were enriched in resident-like macrophages, for example, aortic-intima-resident-like (Mac$^{AIR}$) macrophages, adventitial and intimal TLF-like (expressing TIMD4 and/or LYVE1 and/or FOLR2) macrophages, and were downregulated under a iWD versus a cWD (Fig. 2b and Supplementary Table 3). This is typically the case of *Cadm1*, which is enriched in Mac$^{AIR}$-like macrophages[16], and *Phactr1*, which is enriched in resident-like TLF$^+$ aortic macrophages[17].

To examine whether downregulation of the resident-macrophage-associated genes was due to a loss or a phenotypic alteration of resident arterial macrophages, we fate-mapped resident arterial macrophages using *Cx3cr1*$^{creERT2/+}$*Rosa26*$^{LSL-tdTomato}$*Ldlr*[−/−] mice[16]. After treatment with tamoxifen to induce tdTomato expression (in resident macrophages and blood monocytes), the mice were rested for 2 weeks on a normal diet to allow for clearance of labelled blood monocytes. The mice were then placed on 3 weeks of cWD versus 3 weeks of iWD (Extended Data Fig. 5c). The total numbers of intimal macrophages (CD45$^+$CD68$^+$) were similar between the cWD and iWD groups (Extended Data Fig. 5d,e), consistent with the comparable plaque size between the two groups at this timepoint (Extended Data Fig. 5a,b), but the percentage and the

number of tdTomato$^+$ intimal Mac$^{AIR}$ macrophages was significantly reduced in iWD mice (Extended Data Fig. 5f,g). Thus, there is an accelerated loss of Mac$^{AIR}$ macrophages under an iWD, which is compensated by an increased accumulation of unlabelled monocyte-derived macrophages. These data also indicate that the accelerated depletion of Mac$^{AIR}$ macrophages may in part be responsible for the reduced expression of Mac$^{AIR}$-associated genes in the arterial macrophage dataset. By contrast, the number of adventitial tdTomato$^+$ macrophages was similar between the two groups (Extended Data Fig. 5h), suggesting a predominantly phenotypic alteration of these macrophages at this early timepoint.

To gain further insights into the regulatory mechanisms that may drive the change in macrophage gene expression in the aortas of iWD versus cWD mice, we performed transcription factor (TF) enrichment analysis and found significant enrichment of binding motifs for 27 TFs in aortic macrophage DEGs compared with non-DEGs (Extended Data Fig. 6a and Supplementary Table 4). Most of these TFs are members of the ETS TF family and have recently been shown to affect gene expression within expression quantitative trait loci in humans, and to drive the regulatory effects of inherited human genetic variation[18]. The most enriched ETS TF member was SPIB (Extended Data Fig. 6a), previously identified as a multitissue key driver of a network of genes with causal relationship to coronary heart disease[19]. SPIB is not expressed in myeloid cells but its highly related TF, SPIC, which shares the same DNA-binding specificity, is expressed in macrophages and was one of the most enriched TFs in the aortic macrophage DEGs (Extended Data Fig. 6a). Notably, SPIC was recently identified in a single-cell-based integrated analysis of mononuclear phagocytes of mouse and human atherosclerosis as one of top-3-ranked TFs for mouse TLF$^+$ and Mac$^{AIR}$ macrophages, with notable activity in human LYVE1$^+$ macrophages[20]. Pathway analysis of our data showed that aortic macrophage DEGs enriched for SPIC binding motifs mostly associate with resident-like macrophages and are involved in biological processes related to autophagy, ribosome biogenesis, RNA processing and actin cytoskeleton reorganization (Extended Data Fig. 6b and Supplementary Table 5), recapitulating the GO term enrichment of the whole set of aortic macrophage DEGs (Fig. 2a and Supplementary Table 2). We therefore generated $Lyz2^{cre+/-}Spic^{flox/flox}$ mice (and $Lyz2^{cre+/-}Spic^{flox/+}$ controls) to assess the role of myeloid-specific expression of SPIC on atherogenesis. We found that reconstitution of $Ldlr^{-/-}$ mice with $Lyz2^{cre+/-}Spic^{flox/flox}$ bone marrow substantially accelerated atherosclerosis in response to cWD compared with reconstitution with $Lyz2^{cre+/-}Spic^{flox/+}$ control bone marrow (Extended Data Fig. 6c,d), despite similar circulating cholesterol levels (Extended Data Fig. 6e). However, iWD-induced acceleration of atherosclerosis was limited in mice with myeloid-specific deletion of $Spic$ (Extended Data Fig. 6f). Thus, macrophage expression of SPIC, a TF that was previously identified as a potential regulator of resident-like macrophages[20] and of which the binding motifs are enriched in the arterial macrophage DEGs that are downregulated in iWD mice (Extended Data Fig. 6a and Supplementary Table 4), protects against the development of atherosclerosis. Taken together, our data suggest that alteration of resident-like arterial macrophages is involved in the acceleration of atherosclerosis in response to early intermittent hyperlipidaemia.

### Effect of LYVE1$^+$ macrophage deletion

To examine the specific role of resident arterial macrophages in atherosclerosis, we focussed on $Lyve1$-expressing macrophages, which reside in the adventitia of healthy arteries[21] and are part of intimal resident-like TLF macrophages in atherosclerotic lesions[22], although intimal TLF macrophages often do not express LYVE1 protein. Three weeks of iWD did not affect the accumulation of resident adventitial macrophages compared with the cWD (Extended Data Fig. 5h), but it profoundly reduced the expression of a substantial number of their prototypical genes (Fig. 2b and Supplementary Table 3). We reasoned

that such a profound alteration of gene expression may potentially compromise macrophage survival in the longer term. We therefore assessed the accumulation of LYVE1$^+$ macrophages at various stages of atherosclerosis in mice (Fig. 2c). We detected a significant progressive reduction in aortic LYVE1$^+$ macrophages over time, which was strongly and inversely correlated with the extent of atherosclerotic plaques (Fig. 2d,e). This is consistent with the results of our reanalysis of the single-cell RNA-seq (scRNA-seq) data of published mouse atherosclerosis studies[16,22–24], which shows that prototypical genes associated with arterial resident-like macrophages, including $Lyve1$ and $Nrp1$, are progressively downregulated with increasing durations on a high-fat diet (Extended Data Fig. 7a,b). We also analysed human healthy and atherosclerotic coronary arteries and found a significant reduction in the accumulation of resident-like LYVE1$^+$ macrophages in atherosclerotic compared with healthy human arteries (Fig. 2f,g), supporting the clinical relevance of our findings.

To directly address the role of $Lyve1$-expressing macrophages in the development of atherosclerosis, we generated $Lyve1^{cre+/WT}Csf1r^{flox/flox}$ mice (and $Csf1r^{flox/flox}$ controls) on an apolipoprotein-E-deficient ($Apoe^{-/-}$) background to selectively ablate LYVE1$^+$ macrophages[21] and assessed the effect of their genetic deletion on atherosclerosis. LYVE1$^+$ macrophages were substantially reduced in $Apoe^{-/-}Lyve1^{cre+/WT}$ $Csf1r^{flox/flox}$ mice compared with in $Apoe^{-/-}Csf1r^{flox/flox}$ mice (Extended Data Fig. 8a,b), and this was associated with a significant increase in atherosclerotic plaque size (Fig. 3a,b), despite similar plasma cholesterol levels (Extended Data Fig. 8c). Analysis of plaque composition revealed comparable smooth muscle cell and collagen content (Extended Data Fig. 8d,e). However, we found increased accumulation of intimal CD68$^+$ macrophages (Fig. 3c), and increased necrotic core area (Fig. 3d), which was consistent with impaired efferocytosis within the plaques of $Apoe^{-/-}Lyve1^{cre+/WT}Csf1r^{flox/flox}$ mice (Extended Data Fig. 8f,g). We also used another model of hyperlipidaemia. $Csf1r^{flox/flox}$ and $Lyve1^{cre+/WT}Csf1r^{flox/flox}$ mice were injected with AAV8-D377Y-mPCSK9 and subjected to 6 weeks of cWD or 6 weeks of iWD (Extended Data Fig. 8h,i). We found that, in control $Csf1r^{flox/flox}$ mice, the iWD led to a significant increase in plaque size and necrotic core area compared with the cWD (Fig. 3e–h). However, these iWD-dependent effects were blunted in mice with deletion of LYVE1$^+$ macrophages (Fig. 3i–l). Our data indicate that resident-like arterial macrophages are altered during the progression of atherosclerosis, and their alteration further exacerbates disease progression.

### iWD alters human ASCVD pathways

To further understand the mechanisms that are responsible for iWD-dependent acceleration of atherosclerosis, we assessed the relevance of our findings to the mechanisms of human ASCVD. We found that macrophage DEGs (Fig. 2b and Supplementary Table 1) were enriched in genes shown (or predicted) to be causally involved in human ASCVD in genome-wide association studies (GWAS) (Fig. 4a and Supplementary Table 6). Among the DEGs that could be assigned to a specific macrophage subset, a large proportion belonged to MAC$^{AIR}$/ TREM2-like or resident-like TLF$^+$ macrophages (Fig. 4b and Supplementary Table 7). This is the case for $Phactr1$ (Fig. 2b), the deficiency of which in macrophages accelerates atherosclerotic plaque necrosis in mice[17], and the expression of which in human arteries is controlled by a genetic variant that is causally linked to human ASCVD[25–27] and angiographically determined burden of coronary atherosclerosis[28].

To identify new targets involved in iWD-dependent acceleration of atherosclerosis of relevance to human ASCVD, we analysed publicly available scRNA-seq data[29] and found that 22 out of the GWAS-ASCVD-associated DEGs showed significant differential expression in macrophages of symptomatic compared with asymptomatic carotid atherosclerotic plaques (Fig. 4c and Supplementary Table 8). Here we focussed on $Nrp1$ for several additional reasons. We were

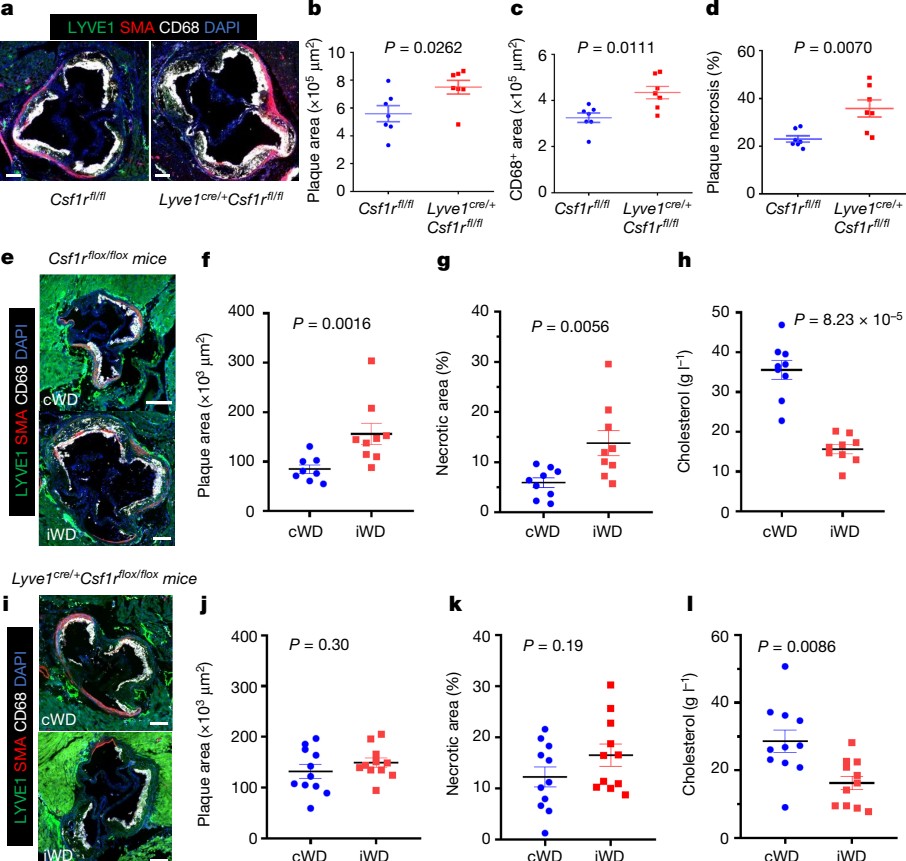

**Fig. 3 | Deletion of LYVE1⁺ resident macrophages accelerates atherosclerosis.**
**a**–**d**, *Apoe⁻/⁻ Lyve1^cre+/− Csf1r^flox/flox* mice (*n* = 7) and *Apoe⁻/⁻ Csf1r^flox/flox* littermate controls (*n* = 7) were put on a cWD diet for 20 weeks. **a**, Representative photomicrographs of aortic sinus atherosclerotic plaques from the two groups of mice are shown, stained with DAPI and antibodies against LYVE1, CD68 and SMA. **b**, The mean atherosclerotic plaque size in the aortic sinus of the two groups of mice. **c**, Quantification of the CD68⁺ area in aortic sinus lesions of the two groups of mice. **d**, The percentage of necrotic area (Methods) in the aortic sinus lesions of the two groups of mice. **e**–**l**, *Csf1r^flox/flox* (**e**–**h**) and

*Lyve1^cre+/− Csf1r^flox/flox* (**i**–**l**) female mice were injected with AAV8-D377Y-mPCSK9 to induce hyperlipidaemia. The mice were then subjected to 6 weeks of cWD or 6 weeks of iWD. *Csf1r^flox/flox*: *n* = 8 (cWD) and *n* = 9 (iWD) mice; *Lyve1^cre+/− Csf1r^flox/flox*: *n* = 11 (cWD) and *n* = 11 (iWD) mice (Methods and Extended Data Fig. 8h). **f,g,j,k**, Quantification of the atherosclerotic lesion size (**f** and **j**) and necrotic core area (**g** and **k**) in the aortic root. **h,l**, Plasma cholesterol levels at the end of the experiment. Data are mean ± s.e.m. Statistical analysis was performed using two-tailed Mann–Whitney *U*-tests (**b**–**d**,**g**,**h** and **j**–**l**). *P* values are indicated on the graphs. For **a**,**e**,**i**, scale bars, 200 µm.

interested in genes that operate in the same biological pathway as *Phactr1*, and our data indicate that the latter is part of an actin filament organization pathway that shows differential enrichment in arterial macrophages of cWD compared with iWD mice (Extended Data Fig. 6b and Supplementary Table 5). Notably, this pathway includes other DEGs that are involved in actin polymerization (such as *Actr2* and *Cyfip2*) and were recently shown, like PHACTR1, to regulate macrophage efferocytosis[14] and to be linked to ASCVD in GWAS[28]. Notably, *Nrp1* is one of the actin filament organization DEGs[30,31] (Supplementary Table 5), and previous work suggested that *Phactr1* may be induced downstream of NRP1 signalling[32]. Furthermore, our meta-analysis of the available scRNA-seq data on arterial macrophages shows that *NRP1* is preferentially expressed in resident-like TLF⁺ macrophages both in mice (Fig. 4d) and in humans (Fig. 4e), with *NRP1^high* macrophages being enriched in biological pathways related to cellular migration and actin cytoskeleton reorganization (Fig. 4f and Supplementary Tables 9 and 10). Other biological pathways of importance to atherosclerosis, for example, receptor-mediated endocytosis, regulation of leukocyte chemotaxis, apoptotic processes and complement activation, were also enriched in *NRP1^high* macrophages (Fig. 4f and Supplementary Tables 9 and 10).

To further assess the clinical relevance of the actin filament organization pathway to the causal pathways leading to ASCVD in humans,

we focussed on the results of a recent study in which the investigators systematically characterized risk variants and genes for coronary artery disease in over a million participants[33]. Combining several complementary approaches, 220 candidate causal genes were prioritized. We applied GO analysis to those 220 genes and found that the most represented biological pathways were pathways related to lipid metabolism, cell migration and vasculature development (Supplementary Table 11 and Extended Data Fig. 9). Genes related to actin filament organization, including *NRP1*, were over-represented in the pathways related to cell migration, further highlighting the relevance of our findings obtained in mice to the pathophysiology of ASCVD in humans. *NRP1* was also represented in pathways related to extrinsic apoptotic signalling and apoptotic cell clearance (Supplementary Table 11).

Together, our data point to actin filament organization and NRP1 expression in macrophages as potentially important targets, the alteration of which may be causally linked to iWD-dependent acceleration of atherosclerosis. However, the role of myeloid NRP1 in atherosclerosis is unclear.

## Effect of *Nrp1* deletion in macrophages

A subset of lymphatic and blood endothelial cells express both LYVE1 and NRP1, precluding the use of *Lyve1^cre+/−* mice to address the role of

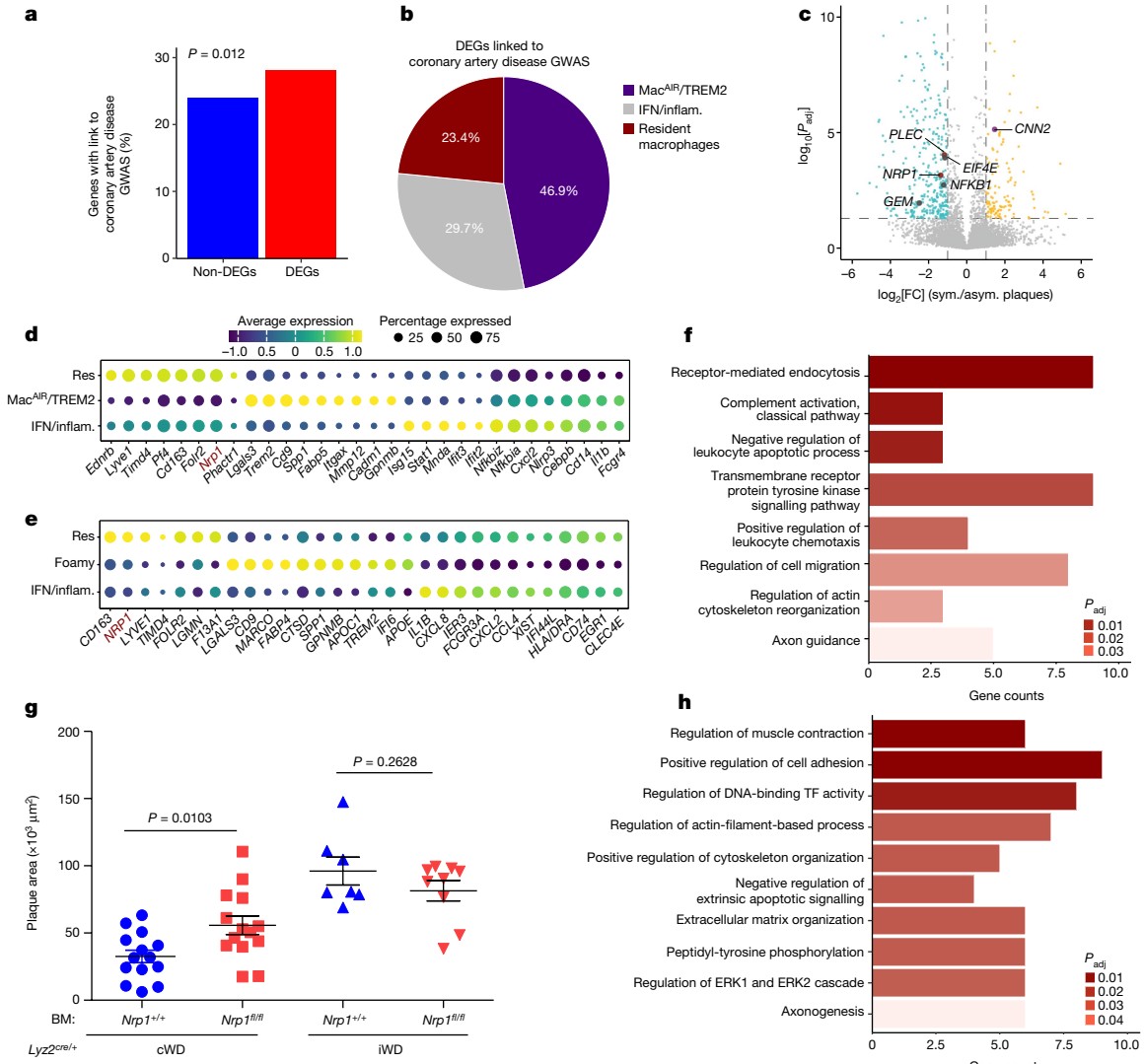

**Fig. 4 | Alteration of ASCVD pathways in resident arterial macrophages accelerates atherosclerosis. a**, The percentage of genes with link to coronary artery disease (CAD) in GWAS for significant ($P_{adj} \leq 0.05$ and $|\log_2[\text{fold change}]| \geq 1$) DEGs (209 out of 746) and non-significant ($P_{adj} > 0.1$ and $|\log_2[\text{fold change}]| < 1$) DEGs (831 out of 3,461); $P_{adj}$ values were calculated using two-sided two-proportions $z$-tests. In total, 64 out of 209 DEGs linked to CAD-GWAS could be assigned to three types of arterial macrophages (the gene list is provided in Supplementary Table 7). **b**, The distribution of 64 CAD-GWAS-related DEGs among macrophage subtypes. **c**, Volcano plot for CAD-GWAS-associated macrophage DEGs that are differentially expressed in symptomatic (sym.) versus asymptomatic (asym.) human carotid atherosclerotic plaques, obtained after reanalysis of available scRNA-seq data[29]. Yellow/cyan colour highlights upregulated genes in symptomatic/asymptomatic plaques. Significant DEGs related to resident/TLF-like (red), Mac^AIR/TREM2-like (violet) and IFN/inflammatory-like (IFN/inflam.; grey) macrophages are highlighted. $P$ values were calculated using two-sided Wald tests followed by adjustment using Benjamini–Hochberg correction. **d,e**, Dot plots of macrophage markers from the meta-analysis of scRNA-seq data of mouse (**d**) and human (**e**) atherosclerotic plaques. NRP1 is shown for comparison. Res, resident. **f,h**, Selected GO (biological process) pathways for significant DEGs enriched in human *NRP1^high* versus *NRP1^low* (**f**) and mouse *Nrp1^+/+* versus *Nrp1^−/−* (**h**) macrophages. Gradient red colour represents $P_{adj}$ values calculated using one-side Fisher's exact tests followed by adjustment with Benjamini–Hochberg correction. The bar size represents the gene count in each pathway. **g**, *Ldlr^−/−* male mice were lethally irradiated and reconstituted with bone marrow (BM) from *Lyz2^cre+/−Nrp1^flox/flox* mice and *Lyz2^cre+/−Nrp1^+/+* controls. Mice were put on a cWD ($n = 14$ (*Lyz2^cre+/−Nrp1^flox/flox*) and $n = 14$ (*Lyz2^cre+/−Nrp1^+/+*)) or iWD ($n = 9$ (*Lyz2^cre+/−Nrp1^flox/flox*) and $n = 7$ (*Lyz2^cre+/−Nrp1^+/+*)) for 6 weeks (Methods). The mean atherosclerotic plaque size in the aortic sinus is shown. Data are mean ± s.e.m. Statistical analysis was performed using two-tailed unpaired *t*-tests. $P$ values are indicated on the graphs.

NRP1 in resident-like macrophages. *Lyz2^cre+/−* mice have previously been used successfully to delete *Nrp1* in myeloid cells and tissue resident-like macrophages[34]. We therefore generated *Lyz2^cre+/−Nrp1^flox/flox* mice (and *Lyz2^cre+/−Nrp1^+/+* controls) and performed a bone marrow transplant experiment into *Ldlr^−/−* mice to assess the role of myeloid-specific expression of NRP1 on atherogenesis (Extended Data Fig. 10a). We successfully reconstituted *Ldlr^−/−* mice with NRP1-deficient and NRP1-sufficient macrophages (Extended Data Fig. 10b). iWD accelerated the development of atherosclerosis compared with cWD in *Ldlr^−/−* mice with *Lyz2^cre+/−Nrp1^+/+* bone marrow (Fig. 4g). Reconstitution of *Ldlr^−/−* mice with *Lyz2^cre+/−Nrp1^flox/flox* bone marrow increased the development of atherosclerosis under the cWD compared with reconstitution with *Lyz2^cre+/−Nrp1^+/+* bone marrow (Fig. 4g), independently of changes in circulating cholesterol levels (Extended Data Fig. 10c). By contrast, deletion of *Nrp1* in myeloid cells had no effect on the development of atherosclerosis under iWD (Fig. 4g). Our findings suggest that the protective effect of NRP1-expressing macrophages in cWD is not maintained under iWD, consistent with the substantial downregulation

**Table 1 | Association of non-HDL-C with plaque presence and plaque area**

| | Carotid plaque | | | | | | | | log-transformed carotid plaque area (mm²) | | | | | | | |
|---|---|---|---|---|---|---|---|---|---|---|---|---|---|---|---|---|
| | Model 1 (n=2,062) | | | | Model 2 (n=1,761) | | | | Model 1 (n=817) | | | | Model 2 (n=689) | | | |
| | RR | 95% CrI | Relative weight (%) | 95% CrI | RR | 95% CrI | Relative weight (%) | 95% CrI | β | 95% CrI | Relative weight (%) | 95% CrI | β | 95% CrI | Relative weight (%) | 95% CrI |
| Lifetime effect | 1.22 | 1.14–1.30 | | | 1.23 | 1.14–1.33 | | | 0.27 | 0.22, 0.33 | | | 0.27 | 0.21, 0.33 | | |
| Life-period, age | | | | | | | | | | | | | | | | |
| Childhood, 6 to 12 years | 1.04 | 1.001–1.14 | 21.4 | 0.7–64.1 | 1.06 | 1.002–1.17 | 27.0 | 1.0–73.8 | 0.06 | 0.002–0.16 | 20.6 | 0.8–57.6 | 0.06 | 0.002–0.17 | 21.7 | 0.8–60.3 |
| Adolescence, 15 to 18 years | 1.07 | 1.004–1.19 | 35.5 | 1.9–82.4 | 1.07 | 1.003–1.19 | 34.2 | 1.7–81.4 | 0.16 | 0.04–0.25 | 57.4 | 17.1–90.4 | 0.14 | 0.03–0.25 | 51.9 | 10.7–88.0 |
| Young adulthood, 21 to 24 years | 1.09 | 1.01–1.20 | 43.1 | 3.4–87.1 | 1.09 | 1.005–1.20 | 38.8 | 2.4–84.3 | 0.06 | 0.003–0.15 | 21.9 | 1.2–55.1 | 0.07 | 0.004–0.17 | 26.5 | 1.6–62.2 |
| Life-course model[a] | Accumulation | | | | Accumulation | | | | Accumulation | | | | Accumulation | | | |

Association of non-HDL-C in three life-periods (childhood, adolescence, young adulthood) with plaque (dichotomous, no/yes) and plaque area (continuous measure) among those with plaque in mid-adulthood (in 2018) aged 41 to 56 years. β, relative risk (RR) and relative weights are shown for a 1 s.d. increase in non-HDL-C. Model 1 was adjusted for sex and year of birth in all participants. Model 2 adjusted for sex, year of birth, area under the curve between ages 6 and 24 years for body mass index, HDL-C and systolic blood pressure, area under the curve between ages 9 and 24 years for blood glucose and physical activity index, education (years studied), ever smoked daily before age 24 years, family history of cardiovascular disease in all participants. CrI, credible interval.
[a]Indicated by Euclidian distance.

of *Nrp1* expression in aortic macrophages under iWD (Fig. 2b and Supplementary Table 1). Bulk RNA-seq analysis of NRP1-deficient and NRP1-sufficient peritoneal macrophages from reconstituted *Ldlr*[−/−] mice showed that *Nrp1* deletion altered macrophage phenotype and affected biological pathways related to actin-filament-based processes, cytoskeleton organization, cell adhesion and apoptosis (Fig. 4h and Supplementary Tables 12 and 13). Consistently, we show that *Nrp1* deletion in macrophages reduces efferocytosis (Extended Data Fig. 10d). Taken together, our data implicate NRP1 as part of a biological pathway that affects the arterial macrophage cytoskeleton and becomes altered in response to early intermittent hyperlipidaemia, leading to acceleration of atherosclerosis.

## Lifetime cholesterol and carotid plaque

Our results led us to hypothesize that exposure to cholesterol early in life (childhood, adolescence and young adulthood) may be a strong determinant of atherosclerosis at mid-adulthood, independently of cumulative lifetime exposure. We therefore analysed data from the Cardiovascular Risk in Young Finns Study (YFS)[35], which is a prospective cohort study that began in 1980 when the participants were aged 3–18 years. Follow-up clinics were conducted regularly thereafter. We performed analyses on two subsets of participants in the YFS when carotid ultrasound studies were performed: 2,653 participants who attended either the 2001 or 2007 follow-ups when aged 24–45 years, and 2,062 participants who attended in 2018 when aged 41–56 years. The participant characteristics are shown in Supplementary Tables 14 and 15. Carotid plaques (n = 88 at the 2001/2007 visits, and n = 817 at the 2018 visit) were defined as protruding focal >50% arterial wall thickenings[36]. The associations of non-high-density lipoprotein cholesterol (non-HDL-C) from three early life periods (childhood, adolescence, young adulthood) with the relative risk of carotid plaque (dichotomous outcome, no/yes) and carotid plaque area (continuous outcome) in mid-adulthood are shown in Supplementary Tables 16 and 17 for the 2001/2007 follow-ups and in Table 1 and Supplementary Table 18 for the 2018 follow-up. Data stratification by sex for carotid plaque outcomes is shown in Supplementary Tables 16 and 17 for the 2001/2007 follow-ups and in Supplementary Tables 19–22 for the 2018 follow-up. The lifetime effect indicated that higher non-HDL-C accumulated across the observed life stages was associated with higher risk of plaque presence (Table 1 and Supplementary Table 16), and higher plaque area (Table 1 and Supplementary Table 17). These findings were consistent in minimally and fully adjusted analyses, including sex, year of birth and potential confounding variables. The results also remained consistent after imputation for missing values in the adjusted model (Supplementary Table 18). Data stratification by sex suggested that the associations were slightly stronger for male than for female participants (Supplementary Tables 16, 17 and 19–22). For each model, the Euclidian distance generally indicated that the accumulation life course model best represented the association with mid-adulthood plaque presence and plaque area (Supplementary Tables 16, 17 and 19–22, Supplementary Methods and Supplementary Data). This was despite some variation in the relative contributions of non-HDL-C levels at each life stage, as shown in Table 1, Supplementary Tables 16–22 and accompanying ternary plots (Supplementary Methods and Supplementary Data), noting that the credible intervals for each life stage were wide and overlapped. Although there was some suggestion that exposure to non-HDL-C at adolescence and young adulthood was contributing most to the observed lifetime effects, exposure to non-HDL-C levels before adulthood (that is, childhood and adolescence) consistently contributed more than half of the lifetime effect to plaque outcomes in mid-adulthood. This suggests that exposure to cholesterol before adulthood contributes to the development of atherosclerosis in mid-adulthood.

## Discussion

Hyperlipidaemia is a major causal risk factor for atherosclerosis. We show that early intermittent hyperlipidaemia is a major determinant of atherosclerotic plaque size compared with late hyperlipidaemia, even when adjusted for cumulative plasma cholesterol levels over time. Atherosclerosis is known as an age-associated arterial disease. Our results show that events (here, high circulating lipid levels) occurring early in life substantially affect the subsequent development and progression of atherosclerosis. Our observation may explain, at least in part, the association between early (and periodic) time course of cholesterol elevation and subsequent cardiovascular events in adulthood[1–3]. Our results may also be relevant to the increased risk of

cardiovascular events after statin discontinuation and re-exposure to high cholesterol levels[37].

Understanding why atherosclerosis is accelerated in response to early and intermittent hyperlipidaemia may lead to the identification of biological mechanisms that are involved in disease development and progression. As a first step, we show that early intermittent hyperlipidaemia impairs critical autophagy- and efferocytosis-related pathways in arterial macrophages with causal links to atherosclerosis in mice and GWAS ASCVD in humans. We also show that most of these pathways are enriched in resident-like arterial macrophages, and that early intermittent hyperlipidaemia accelerates atherosclerosis, at least in part through the loss of the atheroprotective properties of TLF+ macrophages. We identify biological pathways in arterial macrophages, in particular genes that are involved in actin cytoskeleton organization such as *NRP1*, that contribute to atheroprotection, in part through the promotion of efferocytosis. Other relevant genes that we identified in these pathways include *PHACTR1*, which has causal links to ASCVD in humans[25–27] and has previously been shown to limit atherosclerosis through improved efferocytosis[17].

Our results indicate that resident arterial macrophages become altered very early during the development of atherosclerosis. Intermittent hyperlipidaemia prevents the newly recruited monocyte-derived macrophages from acquiring atheroprotective tissue-residency characteristics, leading to substantial disease acceleration. Additional work will be required to assess the differential roles of monocyte-, yolk-sac- or erythromyeloid-progenitor-derived resident macrophages in this process. Nevertheless, our results are in agreement with the concept that the balance of homeostatic versus disease-associated signals is a major determinant of the local diversity of macrophage cell states and, therefore, tissue homeostasis[38].

We found that intermittent hyperlipidaemia can still accelerate atherosclerosis after extended durations of a high-fat diet, long after the early alteration of resident macrophages. This suggests that alteration of resident macrophages sets the stage for further acceleration of atherosclerosis by additional mechanisms, which become revealed by the absence of resident macrophages. This hypothesis merits further investigation.

Our study has limitations. *Cx3cr1*^creERT2/+ mice could not be used to assess the role of resident macrophages in atherosclerosis because *Cx3cr1* heterozygosity affects the development of atherosclerosis[39]. Mice were lethally irradiated before reconstitution with the bone marrow of interest to address the role of macrophage SPIC and NRP1 expression. Resident arterial macrophages are ablated by lethal irradiation but most (approximately 80%) of them reconstitute from the donor bone marrow[40]. Besides macrophages, the use of *Lyz2*^cre+/– also affects gene expression in neutrophils. However, compared to macrophages, *Nrp1* and *Spic* show very low or no expression, respectively, in blood neutrophils. Thus, the results using *Lyz2*^cre+/–*Nrp1*^flox/flox and *Lyz2*^cre+/–*Spic*^flox/flox mice can mostly be attributed to the roles of NRP1 and SPIC in macrophages. Note also that efferocytosis was not directly assessed in plaque-derived macrophages. Finally, although the human data show that hyperlipidaemia early in life determines future atherosclerotic plaque development, these data are observational and are subject to bias and confounding, and additional studies will be required to address the effect of cholesterol variability in childhood or adolescence on plaque development and progression.

In summary, we provide evidence that exposure to high circulating lipid levels in early life is a major determinant of atherosclerosis development and progression in adulthood. This finding has the potential to radically change our understanding of the mechanisms of accelerated atherosclerosis. We unravel major homeostatic roles of resident arterial macrophages in limiting the development of atherosclerosis, and show that early intermittent hyperlipidaemia alters these homeostatic mechanisms to accelerate atherosclerosis. Our results have implications regarding the optimal timing of initiation of lipid-lowering therapies[41], the earlier being the better[42], and the need for sustained lifelong control of circulating lipid levels[43,44].

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

¹Department of Medicine, Section of CardioRespiratory Medicine, University of Cambridge, Heart and Lung Research Institute, Cambridge, UK. ²Immunology Translational Research Programme, Yong Loo Lin School of Medicine, Department of Microbiology and Immunology, National University of Singapore, Singapore, Singapore. ³Immunology Programme, Life Sciences Institute, National University of Singapore, Singapore, Singapore. ⁴Research Centre of Applied and Preventive Cardiovascular Medicine; University of Turku, Turku, Finland. ⁵Centre for Population Health Research, University of Turku and Turku University Hospital, Turku, Finland. ⁶Baker Heart and Diabetes Institute, Melbourne, Victoria, Australia. ⁷Université Paris Cité, Institut National de la Santé et de la Recherche Médicale, U970, PARCC, Paris, France. ⁸Department of Integrative Biology & Physiology, Center for Immunology, University of Minnesota, Minneapolis, MN, USA. ⁹Department of Laboratory Medicine, Medical University of Vienna, Vienna, Austria. ¹⁰Institute of Clinical Molecular Biology, University of Kiel and University Hospital Schleswig Holstein (UKSH), Kiel, Germany. ¹¹Institute for Cardiovascular Prevention, Ludwig-Maximilians-Universität München (LMU), Munich, Germany. ¹²Institute of Ophthalmology, University College London, London, UK. ¹³Department of Medicine, University of Turku, Turku, Finland. ¹⁴Division of Medicine, Turku University Hospital, Turku, Finland. ¹⁵Department of Medicine, Satakunta Central Hospital, Pori, Finland. ¹⁶Department of Clinical Physiology, University of Tampere, Tampere, Finland. ¹⁷Faculty of Medicine and Health Technology, University of Tampere, Tampere, Finland. ¹⁸Finnish Cardiovascular Research Center Tampere, University of Tampere, Tampere, Finland. ¹⁹Analytical Edge, Hobart, Tasmania, Australia. ²⁰Department of Immunology, Institute of Advanced Medicine, Wakayama Medical University, Wakayama, Japan. ²¹Department of Clinical Physiology and Nuclear Medicine, Turku University Hospital, Turku, Finland. ²²These authors contributed equally: Xiaohui Zhao, Hwee Ying Lim, Costan G. Magnussen, Olli T. Raitakari, Véronique Angeli. ✉e-mail: zm255@medschl.cam.ac.uk

## Methods

### Mice

All in vivo experiments using mice were approved by local Institutional Review Boards: the Home Office, UK, under PPL PA4BDF775; the Ethical Committee of INSERM (APAFIS, 29371) and the French Ministry of Agriculture (MESRI 674 29371); the Institutional Animal Care and Use Committee (IACUC) at National University of Singapore (protocol number R21-1562); and under animal protocol 2111–39587A for experiments performed in the United States. All mice were on the C57BL/6 background. Both males and females were used. Atherosclerosis experiments were started at the age of 6 weeks unless otherwise stated. *Lyz2*[cre+/-], *Ldlr*[-/-] and *Rag2*[-/-] mice were initially from Charles River. *Spic*[flox/flox] mice were from T. Kaisho and W. Ise[45]. *Lyz2*[cre+/-]*Nrp1*[flox/flox] and *Lyz2*[cre+/-]*Nrp1*[+/+] mice were generated by the group of Christiana Ruhrberg, University College London, UK. *Cx3cr1*[creERT2/+]*Rosa26*[LSL-Tomato]*Ldlr*[-/-] mice were generated by J. Williams. *Apoe*[-/-] mice were purchased from Comparative Medicine (National University of Singapore). *Lyve1*[cre/cre] and *Csf1r*[flox/flox] mice were purchased from Jackson Laboratory. *Apoe*[-/-]*Lyve1*[cre/WT]*Csf1r*[flox/flox] mice were generated by crossbreeding *Lyve1*[cre/cre] with *Apoe*[-/-]*Csf1r*[flox/flox] mice. In some experiments, 8–10-week-old *Lyve1*[cre/WT]*Csf1r*[flox/flox] and *Csf1r*[flox/flox] female mice were injected intraperitoneally with $0.5 \times 10^{12}$ viral genome copies of AAV8-D377Y-mPCSK9 virus per mouse to induce hypercholesterolemia. AAV8-D377Y-mPCSK9 viruses were purchased from and generated by Phoenix Laboratory of Gene Therapy and Cell Therapy (National University of Singapore). Mice were randomly assigned to their respective experimental groups. Sample size was determined to detect a significant difference ($\alpha = 0.05$ and 80% power) of at least 30% in lesion size between groups. Analyses were performed in a blinded manner. Main atherosclerosis experiments were repeated several times and showed consistent results (see the main text). All animals were maintained in pathogen-free animal facilities and were handled in accordance with protocols approved by local institutional animal care and use committees. Mice were housed under a 12 h–12 h dark–light cycle, at ambient temperature of 22 °C and 45–65% humidity. No wild animals and no field-collected samples were used in the study.

### Bone marrow transplants

*Ldlr*[-/-] mice were irradiated with two doses of 5.5 Gy (separated by 4 h) followed by reconstitution with $0.5–1 \times 10^7$ sex-matched donor bone marrow cells. Animals were randomly assigned to receive the appropriate bone marrow. Mice were then maintained for a 4-week recovery period before being continued on normal or WD (21% fat, 0.15% cholesterol).

### Measurement of plasma cholesterol levels

Blood was obtained from through the saphenous vein (during the experimental period) or the vena cava (at euthanasia) on EDTA. Plasma was separated by centrifugation at 12,000 rpm and kept at −80 °C. Lipids (and corticosterone) were measured by the Core Biochemical Assay laboratory at the University of Cambridge and, in some experiments, using a commercially available cholesterol colorimetric assay kit (BioVision).

### Histology, immunohistochemistry and morphometry of mouse atherosclerotic lesions

Tissues were collected into 4% PFA overnight before washing into PBS. Aortic root and innominate artery cryosections were stained with Oil Red O (Sigma-Aldrich) for atherosclerotic lesion size, Picrosirius Red (Sigma-Aldrich) for collagen content and haematoxylin and eosin and DAPI for acellular debris (necrosis). For immunofluorescence staining, cryosections (kept at −80 °C) were dried for 1 h followed by rehydration in PBS for 10 min before the staining. The samples were then permeabilized in 0.1% Triton X-100, 0.1% citrate in PBS for 30 min. The samples were then washed in PBS and incubated with the blocking solution

(staining solution (PBS, 1% BSA, 2 mM EDTA, 0.01% $NaN_3$) + 5% serum of secondary antibody species) for 30 min followed by incubation with primary antibodies diluted in the blocking solution overnight at 4 °C. The samples were washed with PBS and incubated with secondary antibodies diluted in the staining solution for 3 h. The samples were washed in PBS and nuclei was counterstained with DAPI and mounted with CC mount (Sigma-Aldrich). Quantification of lesion size and composition was performed using ImageJ (National Institutes of Health) according to accepted guidelines[46]. The LYVE1[+] macrophage number in aortic sections was quantified using colocalization of pan-macrophage marker CD68 or the tissue-resident M2 macrophage markers CD206 with LYVE1. We have previously shown that the majority of adventitial resident LYVE1[+] macrophages also co-express CD206[21]. The adventitial area of the innominate artery section was demarcated as the area between GFP autofluorescence of the external elastic laminae and the border of the periadventitial fat. The extent of plaque spread was determined by the measurement of external elastic laminae length beneath the plaque. Necrotic area was determined by demarcating DAPI-negative regions within plaques of the aortic sinus sections. Data analysis was conducted in a blind manner where appropriate. Primary antibodies used for immunofluorescence analysis included anti-MOMA2 (rabbit monoclonal, Bio-Rad, MCA519G; 1:200), anti-CD68 (rat monoclonal, Bio-Rad, MCA1957, 1:300), anti-CD206 (rat monoclonal, Bio-Rad, MCA2235, 1:200), anti-CD3 (rabbit polyclonal, DAKO/Agilent, A045229-2, 1:200), anti-LYVE1 (rabbit monoclonal, Abcam, ab218535, 1:600), anti-CD31 (Armenian-hamster monoclonal, Millipore, MAB1398Z, 1:200), anti-collagen Type I (rabbit polyclonal, Millipore, ABT257, 1:300) and Cy3-conjugated anti-smooth muscle actin (1A4, Sigma-Aldrich, C6198; 1:1,000). Rat IgG (eBioscience) and rabbit IgG (Jackson ImmunoResearch) were used for isotype controls. AF488- (1:300), AF555- (1:300), Cy3- (1:500) and AF647 (1:300)-conjugated antibodies (Jackson ImmunoResearch) were used for fluorescence detection. The apoptosis assay was performed using the Roche In Situ Cell Death Detection Kit, TMR Red (Roche). The number of extracellular (not internalized) apoptotic cells was determined on three ×40 magnification fields of aortic sinus lesions as TUNEL[+]DAPI[+] double-positive cells within the necrotic core of atherosclerotic plaques.

To examine whether NRP1 expression in macrophages affects efferocytosis, bone-marrow-derived macrophages from *Lyz2*[cre+/-]*Nrp1*[flox/flox] and *Lyz2*[cre+/-]*Nrp1*[+/+] mice were incubated with fluorescent dye PKH26-labelled apoptotic Jurkat cells (E6-1; ATCC) induced by ultraviolet irradiation. After 45 min of incubation, macrophages were washed with PBS and then fixed with 4% paraformaldehyde, rinsed with PBS and imaged. The Jurkat cell line used in this experiment tested negative for mycoplasma at first provision. However, it has not been retested later on and has not been authenticated.

### Microbiota analysis using 16S amplicon sequencing

DNA was isolated from faecal material of the experimental mice using the DNeasy PowerSoil Pro Kit (Qiagen) according to the manufacturer's protocol. Extracted DNA was eluted from the spin-filter silica membrane with 100 μl of elution buffer and stored at −80 °C. 16S profiling and MiSeq sequencing was performed as described earlier[47,48] with the following modifications: the V3–V4 region of the 16S gene was amplified using the dual barcoded primers 341F (GTGCCAGCMGCC GCGGTAA) and 806R (GGACTACHVGGGTWTCTAAT). Each primer contained additional sequences for a 12 base Golay barcode, Illumina adaptor and a linker sequence[49].

PCR was performed using the Phusion Hot Start Flex 2× Master Mix (NEB) in the GeneAmp PCR system 9700 (Applied Biosystems) and the following program: 98 °C for 3 min; 25× 98 °C for 20 s, 55 °C for 30 s and 72 °C for 45 s; then 72 °C for 10 min and hold at 4 °C. The performance of the PCR reactions was checked using agarose gel electrophoresis. Normalization was performed using the SequalPrep Normalization Plate Kit (Thermo Fisher Scientific) according to the manufacturer's

instructions. Equal volumes of SequalPrep-normalized amplicons were pooled and sequenced on the Illumina MiSeq ($2 \times 300$ nucleotides) system. MiSeq sequencing data were analysed using MacQIIME v.1.9.1 (http://www.wernerlab.org/software/macqiime). In brief, all sequencing reads were trimmed, retaining only nucleotides with a Phred quality score of at least 20, then paired-end assembled and mapped onto the different samples using the barcode information. Rarefaction was performed at 4,062 reads per sample to normalize all samples to the minimum shared read count and to account for differential sequencing depth. Sequences were assigned to operational taxonomic units (OTUs) using uclust and the greengenes reference database (gg_13_8 release) with 97% identity. Representative OTUs were picked and taxonomy assigned using uclust and the greengenes database. Quality filtering was performed by removing chimeric sequences using ChimeraSlayer and by removing singletons and sequences that did not align with PyNAST. The reference phylogenetic tree was constructed using FastTree 2. Unequal read counts were normalized by rarefaction to 5,257 and 9,200 (ABx experiment) reads per sample. The relative abundance was calculated by dividing the number of reads for an OTU by the total number of sequences in the sample. $\beta$-Diversity was calculated using Bray−Curtis dissimilarity and visualized by principal coordinate plots.

Gut microbiota was depleted using a combination of oral antibiotics (metronidazole 1 g l$^{-1}$, amoxicillin 0.5 g l$^{-1}$, vancomycin 0.5 g l$^{-1}$, neomycin 1 g l$^{-1}$) dissolved in the drinking water with sucralose (4 g l$^{-1}$). The success of bacterial depletion was determined by 16S qPCR on the isolated faecal DNAs as described previously[50]. 16S copy numbers were normalized per g of faecal material and to the untreated control.

### Fate mapping of and quantification of resident arterial macrophages

$Cx3cr1^{creER/+}R26^{lsl\text{-}tdTomato}Ldlr^{-/-}$ reporter mice were orally gavaged with 200 µl (20 mg ml$^{-1}$) tamoxifen in corn oil to induce tdTomato-expression in CX3CR1-expressing Mac$^{AIR}$ and adventitia-resident macrophages. Mice were then rested on a normal diet for 2 weeks to allow for monocyte turnover in the blood before initiating a 7 week experimental feeding period. The control mice were rested for four additional weeks on a normal diet followed by three weeks on a WD (TD.88137; adjusted calories diet, 42% from fat, Envigo). Mice in the intermittent feeding group underwent three cycles of iWD; this consisted of 1 week of WD followed by 2 weeks on a chow diet, with euthanasia after the third week of WD feeding. Aortae were collected, fixed in a 4% paraformaldehyde with 30% sucrose solution and then stained for macrophages using CD45 (30-F11, BioLegend, 0.5 µg ml$^{-1}$) and CD68 (FA-11, BioLegend, 1 µg ml$^{-1}$) antibodies overnight. Adventitia and aortic intima were whole-mount imaged using the Leica Sp8 laser-scanning confocal microscope using LAS X software. Images were processed using Imaris software (Bitplane), and quantification of cells was performed using the spot function. Adventitia samples were collected at three different regions of interest (approximately 1 mm × 1 mm each), and quantification of resident adventitia cell density was averaged for each mouse. Macrophages in the aortic intima were quantified by CD68 expression, and Mac$^{AIR}$ cells were identified for co-expression of tdTomato. Sample imaging and quantification were performed with sample identification blinded to the researcher.

### Total RNA extraction, cDNA synthesis and qPCR

Total RNA was extracted from cells using the RNeasy mini kit (Qiagen) and cDNA was synthesized using the QuantiTect Reverse Transcription Kit. Quantitative PCR (qPCR) was performed using the SYBR Green qPCR mix (Eurogentec) on the Roche Lightcycler. $RplpO$ (also known as $36B4$) was used as a reference gene. Data were analysed using the $\Delta\Delta C_t$ method.

The primer list was as follows: mouse $Nrp1$ forward, 5′-CATCTCC CGGTTACCCTCATTCTT-3′; mouse $Nrp1$ reverse, 5′-GCGGCCGCCT TCATTCTC-3′; $RplpO$ forward, 5′-TCTCCAGAGGCACCATTGAAA-3′; $RplpO$ reverse, 5′-CTCGCTGGCTCCCACCTT-3′.

### Artery tissue digestion
Animals were perfused with cold PBS to remove any trace of blood. Aortas were collected and cleaned of contaminated tissues under a dissection microscope. Aortas were minced with scissors and subjected to two different enzymatic digestion methods. For batch 1, collagenase D (0.2 mg ml$^{-1}$, Sigma-Aldrich), dispase (1 U ml$^{-1}$, Stem Cell Technologies), DNase I grade II (0.2 mg ml$^{-1}$, Sigma-Aldrich) and elastase (1 mg ml$^{-1}$, Worthington) were used. For batch 2, collagenase type I (450 U ml$^{-1}$, Sigma-Aldrich), collagenase type XI (125 U ml$^{-1}$, Sigma-Aldrich), hyaluronidase type 1 (60 U ml$^{-1}$, Sigma-Aldrich), DNase I (60 U ml$^{-1}$, Sigma-Aldrich) were used, then incubated for 45 min at 37 °C. Cell suspensions were pipetted up and down until disappearance of all residual piece of tissue and filtered through a 70 µm cell strainer.

### RNA-seq and analyses

**RNA-seq.** For aortic macrophages, single-cell suspensions of aortic cells were prepared. Cell suspensions were incubated with Zombie Aqua Fixable Viability Kit (BioLegend) for 10 min at room temperature and the cells were then stained with fluorescently labelled anti-mouse antibodies against CD45, CD11b and CD64 at 4 °C. Aortic macrophages were identified as CD45$^+$CD11b$^+$CD64$^+$. Peritoneal macrophages were obtained by intraperitoneal injection of cold PBS from $Ldlr^{-/-}$ mice reconstituted with $Lyz2^{cre+/-}Nrp1^{flox/flox}$ or $Lyz2^{cre+/-}Nrp1^{+/+}$ bone marrow. Peritoneal exudates were allowed to adhere to the six-well plates by culturing them for 2 h at 37 °C. Nonadherent cells were removed by gently washing with warm PBS and the remaining peritoneal macrophages were used for bulk RNA-seq.

RNA extraction and residual DNA removal after DNase digestion was performed using the RNeasy mini kit (Qiagen) according to the manufacturer's instructions. After quality checking using the Agilent Bioanalyser 2100 system, total RNA was made into sequencing libraries. For aortic macrophages, we used the Takara SMART-Seq v4 Ultra Low Input RNA Kit. In brief, total RNA was fragmented before reverse transcription. Second-strand cDNA was PCR synthesized with the incorporation of SMART technology. cDNA originally from rRNA was removed selectively before Illumina-compatible barcoded libraries were generated by PCR amplification. Indexed libraries were normalized, pooled and were sequenced on the Illumina HiSeq 4000 platform, single-end reads (SE50). For peritoneal macrophages, total RNA was purified using poly-T oligo beads and fragmented before reverse transcription. After adenylating 3′ ends and ligating the adapter, cDNA fragments were amplified by PCR. Indexed libraries were normalized, pooled and were sequenced on the Illumina NovaSeq 6000 platform, paired-end reads (PE50). All sequencing was performed at the Genomics Core Facility, Cancer Research UK Cambridge Institute.

**RNA-seq analyses.** For iWD and cWD $Ldlr^{-/-}$ mice, single-end, unstranded sequencing was performed using SMART-Seq v4 plus Nextera (Ultra Low Input RNA Kit (Takara)) with read lengths of 50 bp. Raw fastq data have been deposited in ArrayExpress under accession number E-MTAB-12759. There are two separate experiments for this study with iWD and cWD groups, each group has three independent pools per experiment, and each pool comprises three different mice. All raw data were merged and analysed together, with a total of six different pools for each group. The sample summary table is provided in Supplementary Table 23a. The quality control, alignment and gene quantification analysis were run using the nextflow (v.21.05.0.edge)[51] RNA-seq pipeline with nf-core/rnaseq (v.3.2, https://nf-co.re/rnaseq)[52] with the option '--aligner star_salmon' and Ensembl reference genome with the annotation for mouse GRCm39. The alignment results are with a mean of 72% reads uniquely mapping and mean of 34 million reads per sample (Supplementary Table 23a). Corresponding software/packages and versions from nextflow pipeline are listed in Supplementary Table 24. All scripts, with details of software versions, expression raw

count files and results are available at GitHub (https://github.com/CAD-ZM-BFX/Takaoka_Mallat) and Zenodo (https://doi.org/10.5281/zenodo.13137366).

Counts extracted with htseq-counts were used to perform the DEG analysis using DESeq2 (v.1.36.0; R package, v.4.2.1; https://www.r-project.org/)[53]. In total, 746 genes were identified as significant DEGs when comparing the iWD group to the cWD group using the cut-off threshold $P_{adj} < 0.05$ and $|\log_2[\text{fold change}]| \geq 1$ (an absolute fold change of 2). The list of significant DEGs is provided in Supplementary Table 1 and a volcano plot is presented in Fig. 2b. The two-sided Wald-test $P$ value was then adjusted using the Benjamini–Hochberg false-discovery rate multiple-test correction method. GO and KEGG pathway analysis were performed using R package clusterProfiler (v.4.4.4)[54,55] on the significant DEGs ($n = 746$); all identified pathways are listed in Supplementary Table 2, with ten selected pathways plotted in Fig. 2a. The enrichment $P$ value was calculated using the hypergeometric distribution with the one-side Fisher's exact test.

$$P(X \geq x) = 1 - P(X \leq x - 1) = 1 - \sum_{i=0}^{x-1} \frac{\binom{M}{i}\binom{N-M}{n-i}}{\binom{N}{n}}$$

In this equation, $N$ is the total number of genes in the background distribution, $M$ is the number of genes within that distribution that are annotated (either directly or indirectly) to the gene set of interest, $n$ is the size of the list of genes of interest and $x$ is the number of genes within that list which are annotated to the gene set. The background distribution by default is all of the genes that have annotation. $P$ values are adjusted for multiple comparison using Benjamini–Hochberg correction.

TF-binding motif enrichment analysis was performed using Rcistarget package (v.1.16.0). Both significant DEGs and redefined ($P_{adj} > 0.1$ and $|\log_2[\text{fold change}]| < 1$) non-significant DEG after TF analysis were performed. We identified 73 common enriched binding motifs corresponding to 27 TFs (Supplementary Table 4) and we performed two-sample proportion $z$-tests (two-tailed) and adjusted the $P$ values using Benjamini–Hochberg correction. We generated a dot plot by plotting the difference in normalized enrichment score (NES) between significant and non-significant DEGs (Extended Data Fig. 6a). Moreover, SPIC was selected and GO analysis was performed on significant DEGs ($n = 199$) showing enrichment for SPIC binding motifs (Extended Data Fig. 6b and Supplementary Table 5).

To assign significant DEGs to specific mouse macrophage cell types, we performed integrated analysis of the published scRNA-seq data on mouse atherosclerosis. We selected four main mouse studies from a previous article[20] (Supplementary Table 25). Both individual and integrated study analyses were applied for the data using R package Seurat (v.5.0.0)[56] with integration method scVI[57]. We applied FeaturePlot to a set of key mouse macrophage genes (*Csf1r*, *Sepp1*, *Pf4*, *F13a1*, *Folr2*, *Nrp1*, *Lyve1*, *Cd163*, *Ednrb*, *Timd4*, *Phactr1*, *Egr1*, *Lgals3*, *Ctsd*, *Cxcl2*, *Ccl4*, *Trem2*, *Spp1*, *Cd9*, *Gpnmb*, *Il1b*, *Cd74*, *Stat1*, *Ifit3*, *Mnda*, *Irf7*, *Isg15*, *Ifit2* and *Fcgr4*) to identify the macrophage cluster. We then reclustered the macrophage cell types into six main subtypes of macrophages identified as resident-like, IFN-, inflammation-, Mac$^{AIR}$- and TREM2-like and CCR2-MHCII. In early atherosclerosis, Mac$^{AIR}$ macrophages are the first macrophages that become foam cells, and Mac$^{AIR}$- and TREM2-like macrophages share common markers and gene programs[16]. Therefore, they were grouped together for some analyses. For mouse macrophages, we identified *NRP1*$^{high}$ and *NRP1*$^{low}$ groups, then performed GO pathway analysis using enrichR (R package, v.3.1)[58–60] (Supplementary Table 10).

Further differential expression analysis between chow diet and high-fat diet of mouse scRNA-seq data across different time courses of high-fat diet was performed for three main subsets of macrophages using the FindMarkers function with the MAST method in Seurat (v.5.0.0, R package)[56]. A $\log_2[\text{fold change}]$ heat map was generated

with key genes in each annotated subtype of macrophage across time (Extended Data Fig. 7a). A proportion bar plot presents the general information for three main macrophage subtypes and the change in their proportions with increased duration of high-fat diet (Extended Data Fig. 7b).

GWAS analysis was performed using the extracted list related to atherosclerosis/CAD (coronary artery disease) from the GWAS Catalog (https://www.ebi.ac.uk/gwas/) (Supplementary Table 26) and from supplementary table 13 of ref. 28. The GWAS list human symbols were converted to mouse symbols using g:Profiler2 orthology (v.0.2.1)[61] in R (Supplementary Table 27). We then performed the overlap analysis between the GWAS list and our DEGs list (significant DEGs and non-significant DEGs; Fig. 4a,b and Supplementary Tables 6 and 7). To assign the 210 significant DEGs that are linked by coronary artery GWAS to specific macrophage cell types, we performed integrated analysis of the published scRNA-seq data on human atherosclerosis. Six human studies from cardiovascular PlaqView were selected (https://www.plaqview.com/; Supplementary Table 28). Integrated analysis was performed first using R package Seurat (v.5.0.0)[56], and the macrophage cluster was identified by applying the FeaturePlot for a set of key human macrophage genes (*CSF1R*, *PF4*, *FOLR2*, *NRP1*, *LYVE1*, *LGALS3*, *CXCL2*, *SPP1*, *GPNMB*, *IL1B*, *APOC1*, *APOE*, *FCGR4*, *IFIT2* and *FABP4*). We then reclustered the macrophage cell types into five main subtypes of macrophages identified as resident-like, IFN-, inflammation-, foamy-like and mixed subtypes. Next, we assigned our GWAS-related 210 significant DEGs (Supplementary Table 6) to each macrophage cell type key marker list (Fig. 4b and Supplementary Table 7). For both human macrophages, we identified *NRP1*$^{high}$ and *NRP1*$^{low}$ groups, then performed GO pathway analysis using enrichR (R package, v.3.1)[58–60] (Supplementary Table 9).

We analysed data from a recent GWAS study in which the investigators systematically characterized risk variants and genes for CAD in over a million participants[33]. Two hundred and twenty candidate causal genes at 279 CAD associations were prioritized, combining eight complementary approaches. We used clusterProfiler (v.4.4.4, R package)[54,55] and performed GO enrichment analysis of biological process orthology. In total, 178 out of 220 causal genes contributed to the GO enrichment analysis, and 867 enriched biological processes were identified (Supplementary Table 11). Selected pathways and corresponding causal gene network plots are shown in Extended Data Fig. 9 using the cnetplot function in R (enrichplot v.1.22.0; https://bioconductor.org/packages/release/bioc/html/enrichplot.html). Significant differentially expressed genes in our RNA-seq data of aortic macrophages from iWD versus cWD (Supplementary Table 1) are shown (Extended Data Fig. 9).

## RNA-seq of peritoneal macrophages
Paired-end (PE50), reverse-stranded sequencing for *Nrp1* WT versus *Nrp1*-KO peritoneal macrophages in reconstituted *Ldlr*$^{-/-}$ male mice was performed using the Illumina NovaSeq6000 platform. Raw fastq data have been deposited in ArrayExpress under accession number E-MTAB-12761. DEG analysis was performed using the same workflow as mentioned for iWD versus cWD. The sample summary and mapping statistics table is provided in Supplementary Table 23b. There were 113 significant DEGs identified ($P_{adj} < 0.05$ and $|\log_2[\text{fold change}]| \geq 0.6$; Supplementary Table 12). GO was performed using clusterProfiler (R package, v.4.4.4)[54,55] with the 113 DEGs, and the list of pathways is given in Supplementary Table 13.

## Immunohistochemistry of human coronary healthy and atherosclerotic arteries
Samples of human aortic tissue were obtained from the INSERM cardiovascular tissue Biobank (member of European BBMRI-RIC organization). The study was performed according to the Guidelines of the World Medical Association Declaration of Helsinki. The local ethics committee of the INSERM Paris (01-024) approved the study, and written informed consent for permission was given by all patients.

Human coronary arteries were obtained from patients with coronary artery disease from explanted hearts of transplant recipients (average age, 59 years) INSERM. Coronary artery disease classification was based on coronary angiography before transplant and macroscopic aspect at dissection to confirm the presence of atherosclerotic plaque. Furthermore, control healthy human coronary arteries were obtained from explanted hearts of organ donors (average age, 56 years). Paraffin sections were stained with H&E to assess atherosclerosis. For immunostaining, antisera were first optimized using different dilutions to determine the best staining results with minimal background. Immunostainings were performed as previously described[62] using the following marker antibodies: goat anti-human LYVE1 (R&D, AF2089, 1:100) and mouse anti-human CD68 (DAKO, M0876, 1:50). After antigen retrieval and incubation with primary antibodies, visualization was performed using secondary antibodies (donkey anti-goat Cy3, Jackson Immuno-Research, 705-165-147, 1:300 for LYVE1 and donkey anti-mouse Alexa 647, Jackson ImmunoResearch, 705-605-150, 1:300 for CD68) as previously described[62,63]. DAPI was used to stain the cell nuclei. For negative controls, staining was performed without primary antibodies. Stained slides were imaged using the Leica TCS SP8 3X or by DM6 B Thunder Imager (Leica Microsystems). Images were acquired under identical microscope settings using sequential acquisition of different channels to avoid cross-talk between fluorophores, and non-overlapping fluorophores were used for colocalization analysis. All images were prepared as TIF files and quantified using ImageJ software. For quantification of LYVE1$^+$ macrophages in the adventitia, LYVE1$^+$CD68$^+$ macrophages among total adventitial CD68$^+$ cells were quantified in 5–8 images per section in three sections per sample and expressed as percentage of LYVE1$^+$ macrophages from adventitial CD68$^+$ cells.

## Participants of the YFS

The Cardiovascular Risk in Young Finns Study is a prospective cohort study that began in 1980 when the participants were aged 3–18 years. Follow-up clinics were conducted in 1983, 1986, 1989 (sub-sample), 1992 (sub-sample), 2001, 2007, 2011 and 2018. The primary sample for this analysis included up to 2,062 participants who attended the most recent 2018 follow-up when aged 41–56 years and who had carotid ultrasound studies performed. Moreover, a secondary sample included 2,643 participants who attended either the 2001 or 2007 follow-ups then aged 24–45 years when carotid ultrasound studies were also performed. Ethics approval for the baseline study was granted by the University of Turku Faculty of Medicine Ethical Committee (letter dated 14/11/1978, minutes 3/1978) and for the most recent follow-up, granted by the Hospital District of Southwest Finland (Ethics approval number ETMK:68/1801/2017).

## Measures performed in the YFS

At each timepoint, venous blood samples were taken after an overnight fast (12 h) using standard methods to measure serum total cholesterol and HDL-C that were calibrated to shifts in assay kits over time. Non-HDL-C was calculated as total cholesterol minus HDL-C. In 2001 and 2007, B-mode ultrasound with 13.0 MHz linear array transducers (Sequoia 512, Acuson) was performed on the left common carotid artery and the beginning of the bifurcation; and, in 2018, using the GE Logiq S8 system (GE Vingmend Ultrasound A/S) on both the left and right common carotid artery, bifurcation and internal carotid artery. Carotid plaques were defined as focal structures protruding into the arterial lumen of at least 0.5 mm or 50% of the surrounding intima–media thickness value or demonstrating a thickness of 1.5 mm as measured from the media–adventitia interface to the intima–lumen interface[64]. Carotid artery plaque measures were performed off-line by a single reader blinded to the participant's details. Where ultrasound data were available on participants at both 2001 and 2007 follow-ups, data from 2007 was used. The prevalence of plaque was 39.6% in 2018 (817 out of 2,062) and 3.3% (88 out of 2,643) in 2001/07.

## Statistics and reproducibility

Statistical analyses were performed using Graph Pad Prism 7 (Graph Pad Software). Experimental groups were compared using two-tailed Student's unpaired $t$-tests or Mann–Whitney tests as appropriate. Data are presented as mean ± s.e.m. $P < 0.05$ was considered to be significant.

**Statistical analyses in the YFS.** *The BRLM.* The Bayesian relevant life-course exposure model (BRLM) was used to identify relative contribution of non-HDL-C measured in childhood (aged 6, 9 and 12 years), adolescence (aged 15 and 18 years), and young adulthood (aged 21 and 24 years) on the atherosclerosis outcome variables in mid-adulthood (carotid plaque presence with Poisson regression and plaque area with linear regression). Detailed methods on the BRLM have been provided elsewhere[65,66]. The BRLM estimates an overall or total lifetime effect, representing the maximum accumulated effect of non-HDL-C measured at different stages across the observed life-course (in this case childhood, adolescence and young adulthood) on each outcome variable in mid-adulthood ($\beta$ for continuous outcomes, and relative risk for categorical outcomes). The BRLM also allows estimation of the relative weights and their posterior distributions for non-HDL-C measured in childhood, adolescence and young adulthood that represent the direct or relative contribution of non-HDL-C at each life stage to each outcome. These weight parameters help to determine the life-course model that is best supported by the data. The life-course association of non-HDL-C and each outcome forms one of three prevailing hypotheses: (1) an accumulation life-course model (non-HDL-C at each life stage is having an equal contribution); or subclasses of an accumulation model; (2) a sensitive life-course model (non-HDL-C levels at each life stage has different importance); or (3) a critical life-course model (non-HDL-C at only one life stage is of importance). Using the accumulated effect and relative weight, the BRLM also estimates the life-stage-specific effect. Here we fit the BRLM using a Cauchy (0, 2.5) prior for the lifetime effect and a non-informative Dirichlet (1, 1, 1) prior for weights.

*The IGC model.* The individual growth curve (IGC) model[67] was used to interpolate over ages with missing non-HDL-C values[68]. The IGC model is an advanced multilevel mixed-effect regression model that enables the simultaneous modelling of the interindividual differences in intraindividual systematic changes over time (repeated measurements at the individual level). The non-HDL-C trajectory in our study was best described by an IGC model with a cubic age polynomial (age$^{67}$) random intercept, quadratic age random slope and inclusion of sex as a modifier. If observed non-HDL-C data at the ages used in the BRLM were missing, we used individual-level data derived from the IGC model to fill these gaps. These non-HDL-C levels at each timepoint were transformed into age- and sex-specific $z$ scores. Life-stage averaged non-HDL-C $z$ scores were calculated as the mean values between ages 6 to 12 years for childhood (($z_{6\text{ years old}} + z_{9\text{ years old}} + z_{12\text{ years old}}$)/3), ages 15 and 18 years for adolescence (($z_{15\text{years old}} + z_{18\text{ years old}}$)/2), and ages 21 and 24 years for young adulthood (($z_{21\text{years old}} + z_{24\text{ years old}}$)/2). In our primary analyses, we show the association for a 1 s.d. increase in non-HDL-C $z$-score accounts for variations in non-HDL-C that naturally occur with age and sex. We have done this to ensure a uniform metric for comparison across each life-stage. However, we also provide data for a per 1 mmol l$^{-1}$ increase in non-HDL-C in the supplement, which might be more clinically relevant. We routinely adjusted for potential confounders in a minimally adjusted model (model 1) that included sex and year of birth, and a fully adjusted model (model 2) that included sex, year of birth, education (years studied), ever smoked daily before age 24 years, family history of cardiovascular disease and areas under the curve of measurements available up to age 24 years for body mass index, HDL-C, systolic blood pressure, fasting blood glucose and physical activity index. All models were additionally performed stratified by sex. Owing to missing covariates, there was a reduction in sample size between models 1 and 2. Methods for imputing missing values have been described

(https://github.com/MadathilSA/BayesianRelevantLifeCourseExposure/blob/master/Manuscript III/HeNCeIndiaSmkModelImput.jags). Multiple imputation was undertaken for the covariates of youth age, ever smoked, family history of cardiovascular disease, education (number of years studied), and cumulative body mass index, systolic blood pressure and HDL-C, using predictive mean matching via the R library mice. Three imputations were made, and the means and credible intervals were calculated using the draws across all imputations.

The rstan package of R studio (v.3.5.3, R Foundation for Statistical Computing) was used to fit the BRLM in the probabilistic programming language, Stan[69]. The Lme4 package of R studio (v.3.5.3, R Foundation for Statistical Computing) was used to perform the IGC modelling.

### Reporting summary

Further information on research design is available in the Nature Portfolio Reporting Summary linked to this article.

## Data availability

All the associated raw data presented in this paper are available from the corresponding author on reasonable request. Raw RNA-seq data are accessible through EMBL-EBI ArrayExpress under accession numbers E-MTAB-12759 and E-MTAB-12761. All of the scRNA-seq datasets used in this study are publicly available. For mouse datasets, see Supplementary Table 25. For human datasets, see Supplementary Table 28. Source data are provided with this paper.

## Code availability

Custom codes for RNA-seq analysis used in this paper are available at GitHub and Zenodo[70] (https://github.com/CAD-ZM-BFX/Takaoka_Mallat, https://doi.org/10.5281/zenodo.13137366).

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

**Acknowledgements** Z.M. is supported by a BHF chair grant (CH/10/001/27642), a BHF Special Project Grant (SP/F/20/150010), the BHF Centre of Research Excellence (RE/18/1/34212), the Leducq Foundation (20CVD03) and the NIHR Cambridge Biomedical Research Centre (NIHR203312). The views expressed are those of the authors and not necessarily those of the NIHR or the Department of Health and Social Care; M.N. by a BHF Intermediate Fellowship (G103314). S.A.N. is supported by a BHF Project Grant (PG/21/10816); and T.X.Z. through the BHF (CH/10/001/27642) and by the BMA Foundation, and the Academy of Medical Sciences (SGL025/1071). This work is also supported by a NMRC-funded grant to H.Y.L. (OFYIRG20nov-0013) and to V.A. (OF-IRG19Nov-0112). J.W.W. and P.R.S. were supported by the American Heart Association (CDA855022) and the US National Institutes of Health (R01 AI165553); D.T. by the European Research Council (ERC Starting grant), the Austrian Science Fund (I4963, P35233) and the European Research Area Network on Cardiovascular Diseases (I4647); F.S. by the German Research Foundation (DFG) through the grant SO1141/10-1 and the Research Unit FOR5042 "miTarget—The Microbiome as a Target in Inflammatory Bowel Diseases" (project P5); S.K.M. through a Corona Foundation grant (S199/10087/2022), and a DFG grant (SFB 1123/Z1); and C.R. by a Wellcome Investigator Award (095623/Z/11/Z). This work is also supported by the Academy of Finland (grant numbers 286284, 134309 (Eye), 126925, 121584, 124282, 129378 (Salve), 117787 (Gendi) and 41071 (Skidi)); the Social Insurance Institution of Finland; Competitive State Research Financing of the Expert Responsibility area of Kuopio, Tampere and Turku University Hospitals (grant number X51001); the Turku University Foundation; Tampere University Hospital Supporting Foundation; the Juho Vainio Foundation; Paavo Nurmi Foundation; the Finnish Foundation of Cardiovascular Research; Orion-Farmos Research Foundation; Sigrid Juselius Foundation; Emil Aaltonen Foundation; Yrjö Jahnsson Foundation; Signe and Ane Gyllenberg Foundation; Diabetes Research Foundation of Finnish Diabetes Association; Tampere Tuberculosis Foundation; the Finnish Cultural Foundation; and EU Horizon 2020 (grant number 755320 for TAXINOMISIS). J.S.K. is supported by the European Research Council (grant number 742927 for MULTIEPIGEN project), Maud Kuistila Memorial Foundation, Päivikki and Sakari Sohlberg Foundation, Finnish Medical Foundation, Paulon Foundation and Licentiate of Medicine Paavo Ilmari Ahvenainen Foundation; and C.G.M. by a National Health and Medical Research Council (NHMRC) Investigator Grant (APP1176494). The contents of the published material are solely the responsibility of the individual authors and do not reflect the views of the NHMRC. We thank W. Ise for his contribution to generating the *Spic*^flox/flox mice; J.-B. Michel for providing tissue sections of healthy and atherosclerotic human coronary arteries; D. Cuchet-Lourenco and E. F. Warner for providing Jurkat cells and assistance with BMDM experiments, respectively; the staff at the Phenotyping Hub and Biochemical Assay Laboratory of Cambridge University Hospitals, the University of Cambridge Biomedical Services at the Anne McLaren Building, the Cancer Research UK Cambridge Institute Genomics Core facility and the Genomics & Bioinformatics Core, Wellcome-MRC Institute of Metabolic Science-Metabolic Research Laboratories and Medical Research Council Metabolic Diseases Unit, University of Cambridge; the members of the Advanced Imaging and Histology cores of Life Sciences Institute, NUS; and S. U. Gan and S. K. Chuan for providing AAV8-D377Y-mPCSK9.

**Author contributions** M.T. led and performed most of the experiments on atherosclerosis and contributed to the RNA-seq data. X.Z. analysed the RNA-seq data. V.A. and H.Y.L. designed the experiments related to LYVE1+ expressing adventitial macrophages, and H.Y.L., O.A. and W.S.O. performed the corresponding experiments and analysed the data. C.G.M. and O.T.R. organized data collection, designed the analysis plan of the YFS and interpreted the data, and Y.M. and F.W. performed those analyses. NS performed and analysed atherosclerosis experiments. P.R.S. and J.W.W. performed fate-mapping experiments and analysed data. D.T. performed atherosclerosis and RNA-seq experiments. F.S. performed 16S amplicon sequencing and analysed/interpreted corresponding data. M.N. contributed to analysis of the microbiota depletion experiments. S.K.M., M.H. and A.J.R.H. performed and analysed experiments related to adventitia LYVE1+ macrophages in human coronary arteries, and T.X.Z. provided artery samples. J.H., Y.L., L.M. and S.A.N. contributed to the atherosclerosis and RNA-seq experiments. L.L., M. Clement, M. Chajadine and S.T. maintained genetically modified mice and helped in atherosclerosis experiments. L.D. and C.R. maintained, mated and genotyped genetically modified mice for bone marrow donation. M.J. and J.S.K. did the plaque measurements in the YFS. M.K. contributed to the set-up of the YFS. K.P., S.P.R. and J.M. collected data from the 2018 visit. R.T. conducted statistical analyses on the YFS. T.K. generated and provided genetically modified mice. A.T. and H.A.-O. contributed to data interpretation. Z.M. conceived the idea, designed, initiated and supervised the project, established the required collaborations and wrote the manuscript. All of the authors reviewed and edited the manuscript.

**Competing interests** The authors declare no competing interests.

**Additional information**
**Correspondence and requests for materials** should be addressed to Ziad Mallat.

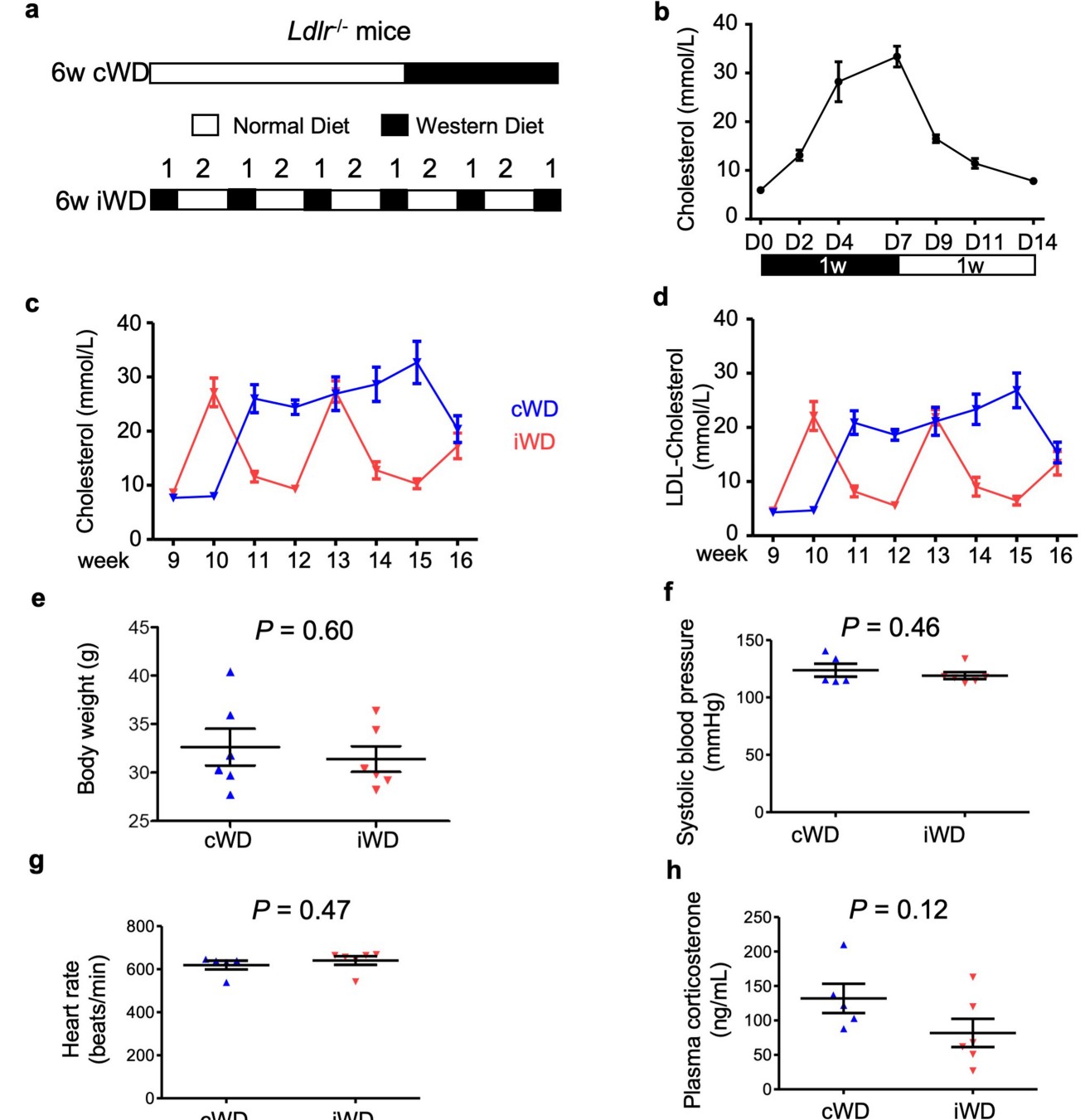

**Extended Data Fig. 1 | Induction of early intermittent hyperlipidaemia in mice. a**, Experimental set up. LDL receptor-deficient (*Ldlr*⁻/⁻) male mice on normal diet were fed a Western-type diet (WD) for 6 weeks (w) either continuously (cWD) or intermittently (iWD). **b**, **c**, Calculation of plasma cholesterol levels in mice subjected to iWD over a period of 6 weeks. Here, examples are given for measurements that started at the end of week 9 and the beginning of week 10 and lasted until end of week 15. **b**, shows cholesterol measurement over 14 consecutive days (n = 4 per time point). **c**, shows weekly cholesterol

measurements in *Ldlr*⁻/⁻ mice under cWD (during the last 6 weeks, n = 6) and iWD (n = 6). **d**, weekly LDL cholesterol measurements in *Ldlr*⁻/⁻ mice under cWD (during the last 6 weeks, n = 6) and iWD (n = 6). **e**-**h**, Body weight (n = 6 per group) (**e**), systolic blood pressure (**f**), heart rate (**g**), and plasma corticosterone levels (**h**) in randomly selected cWD (n = 5) or iWD (n = 6) fed *Ldlr*⁻/⁻ mice at the end of the experiment. Mean ± s.e.m.; two-tailed unpaired *t*-test (**e**-**h**). *P* values are indicated on the graphs.

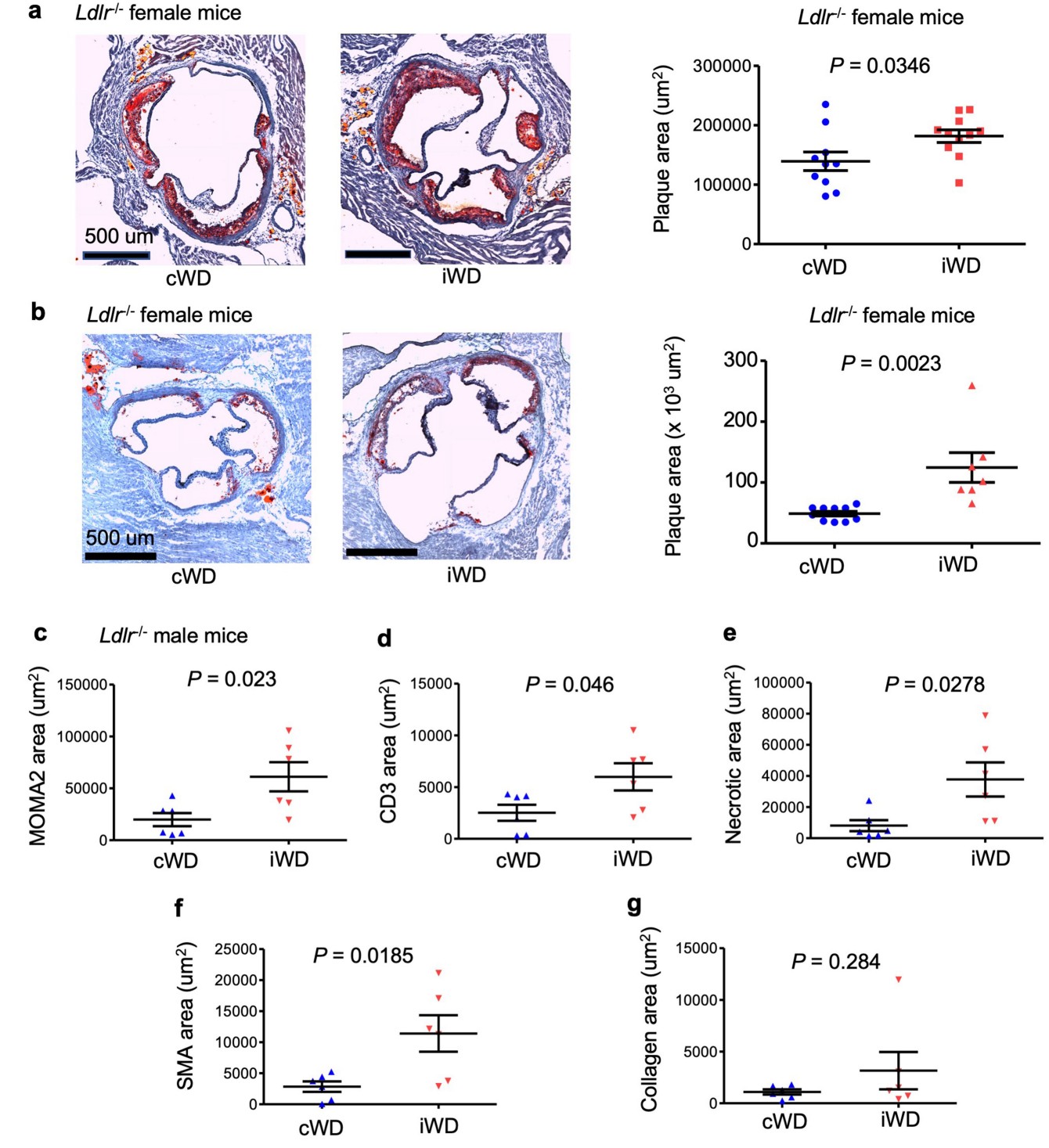

**Extended Data Fig. 2 | Early intermittent hyperlipidaemia accelerates atherosclerosis in mice and increases plaque inflammation and necrosis.** **a**, LDL receptor-deficient (*Ldlr$^{-/-}$*) female mice were fed a Western-type diet (WD) for 6 weeks (w) either continuously (cWD) or intermittently (iWD). Representative photomicrographs (left panels) and mean atherosclerotic plaque size (right panel) in the aortic sinus of the 2 groups of mice are shown (n = 10 cWD and n = 11 iWD). **b**, *Ldlr$^{-/-}$* female mice irradiated and reconstituted with WT bone marrow and subjected to 6 weeks of cWD (n = 10) or iWD (n = 8). Representative photomicrographs (left panels) and mean atherosclerotic plaque size (right panel) in the aortic sinus of the 2 groups of mice are shown. **c-g**, *Ldlr$^{-/-}$* male mice were put on 6 weeks cWD (n = 6) versus 6 weeks iWD (n = 6). Mean plaque area staining positively for MOMA2 (**c**), CD3 (**d**), acellular debris (necrosis) (**e**), αSMA (**f**), and Sirius red (collagen) (**g**) in one set of a series of experiments. Mean ± s.e.m.; two-tailed unpaired *t*-test. *P* values are indicated on the graphs.

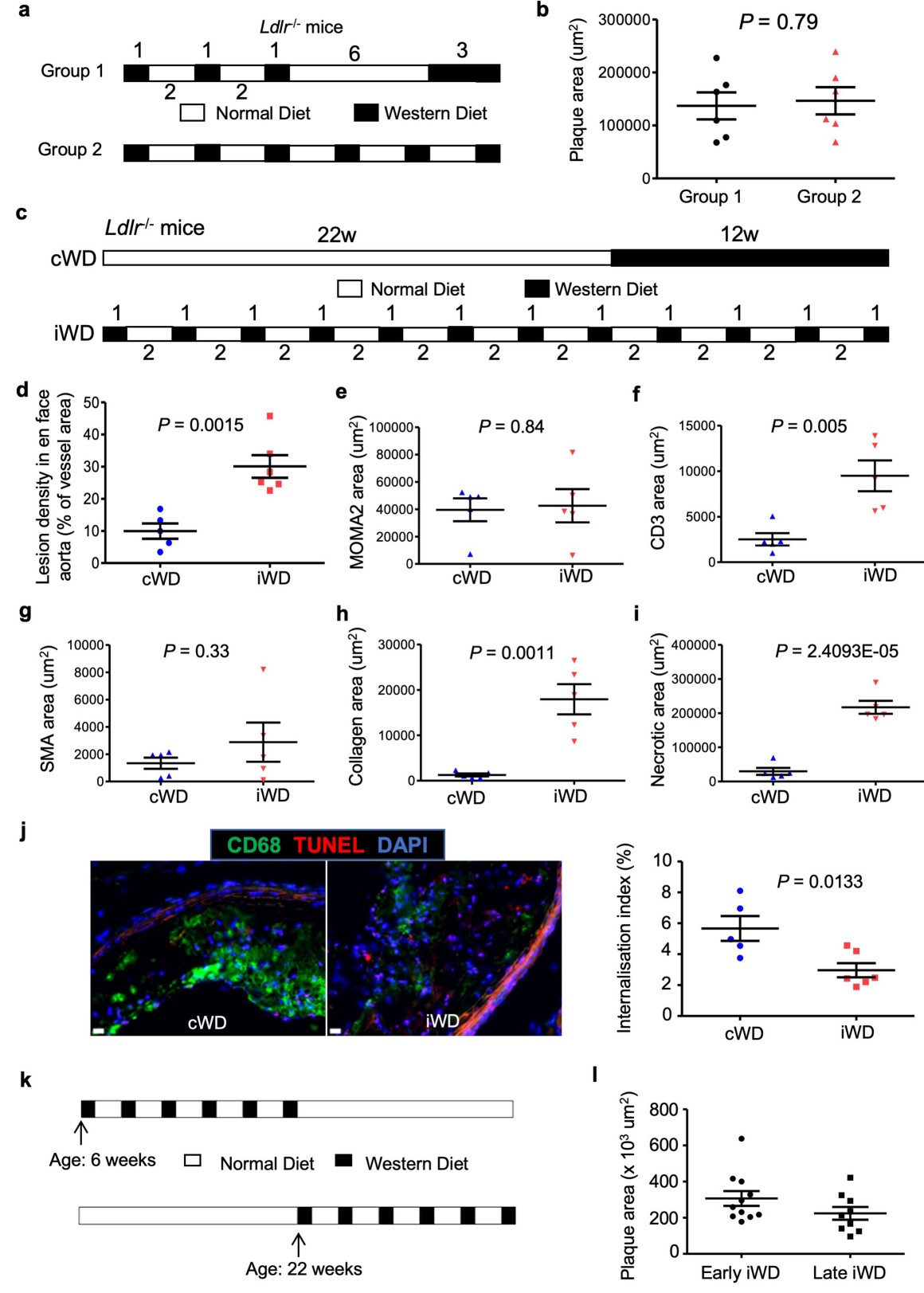

**Extended Data Fig. 3 | Various regimens of early intermittent hyperlipidaemia accelerate atherosclerosis in mice. a**, Experimental set up. In group 1, LDL receptor-deficient (*Ldlr*[-/-]) male mice first received 3 intermittent weeks of Western diet (WD) at the beginning of the experiment, then 3 weeks of continuous WD just before the end of the experiment (n = 6). Group 2 mice received 6 weeks of intermittent WD (n = 6). **b**, Mean atherosclerotic plaque size in the aortic sinus of the 2 groups of mice (n = 6 per group). **c-j**, Experimental set up (**c**). *Ldlr*[-/-] male mice were put on 12 weeks of cWD (n = 5) versus 12 weeks of iWD (n = 6). **d**, Mean atherosclerotic plaque size on en face aorta in the 2 groups of mice (n = 5 to 6 per group). Mean plaque area staining positively for MOMA2 (**e**), CD3 (**f**), αSMA (**g**), Sirius red (collagen) (**h**), and acellular debris (necrosis) (**i**), (n = 5 per group). **j**, Representative photomicrographs of TUNEL staining (red) and CD68 staining (green) (scale bar = 10 μm) and quantification of the proportion of internalized apoptotic cells in plaques from the 2 groups of mice (n = 5 to 6 per group). **k**, Experimental set up. *Ldlr*[-/-] male mice were put on 6 weeks iWD starting at 6 or 22 weeks of age (n = 11 and n = 9 mice, respectively). **l**, Mean atherosclerotic plaque size in the 2 groups of mice. Mean ± s.e.m.; two-tailed unpaired *t*-test (**b**, **d-j**, **l**). *P* values are indicated on the graphs.

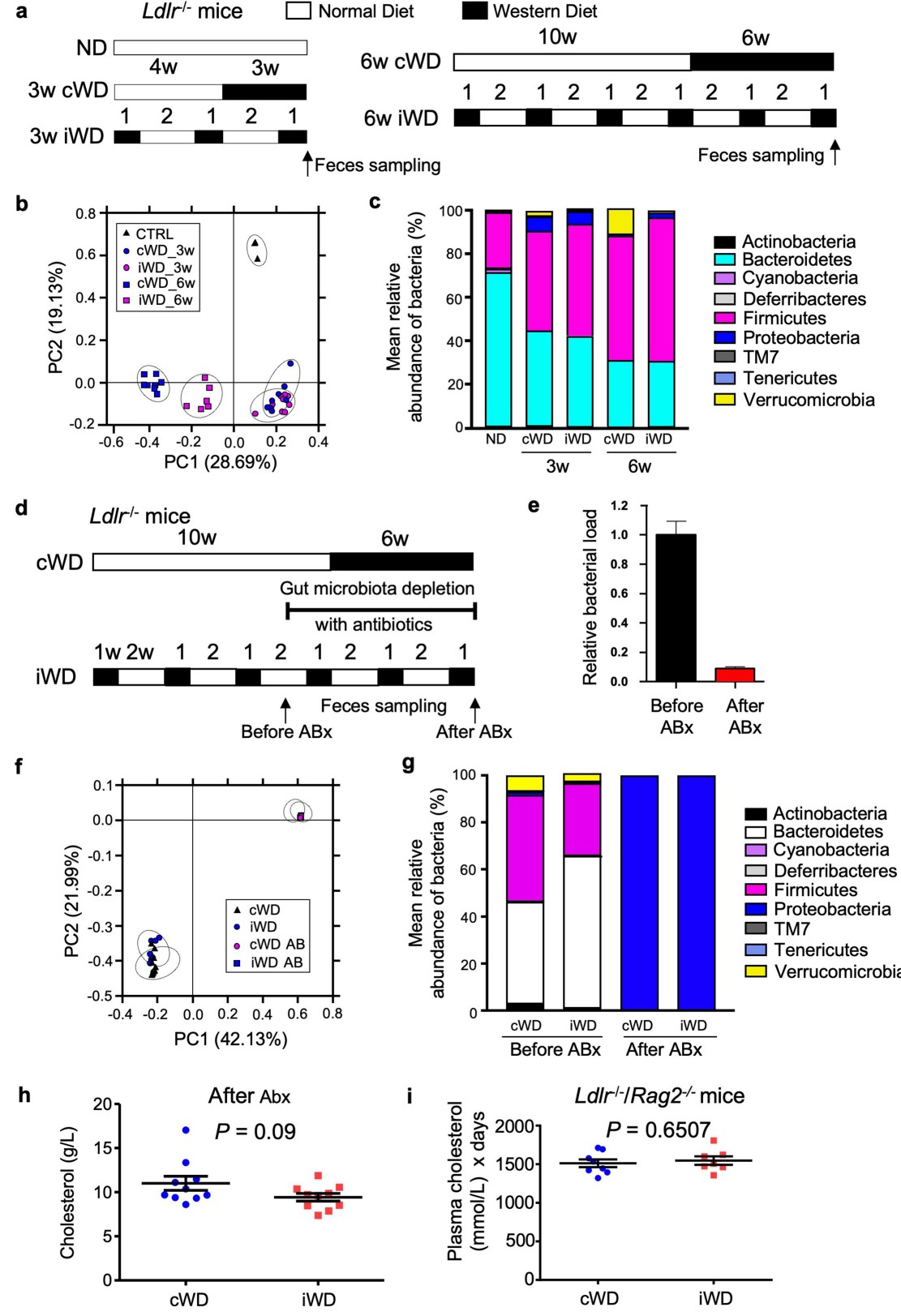

**Extended Data Fig. 4** | See next page for caption.

**Extended Data Fig. 4 | Early intermittent hyperlipidaemia, gut microbiota, and accelerated atherosclerosis in mice. a**, Experimental set up. LDL receptor-deficient (*Ldlr*[−/−]) male mice were fed a normal (ND, control, n = 4) diet or a Western-type diet (WD) for 3 or 6 weeks (w) either continuously (cWD) or intermittently (iWD) (n = 8 cWD 3w, n = 10 iWD 3w, n = 8 cWD 6w, n = 6 iWD 6w). Faecal samples were collected during and at the end of the experiment for analysis of gut microbiota using 16S rRNA sequencing (**b**-**c**). **b**, Principal coordinate analysis reveals distinct separation in microbiota composition due to diet. **c**, WD-induced changes in microbiome composition on the phylum level (mean relative abundance). **d**, Experimental set up. *Ldlr*[−/−] male mice were put on 6 weeks cWD versus 6 weeks iWD. Treatment with oral antibiotics (see Methods) to deplete the gut microbiota was started at the beginning of week 9. Faecal samples collected at the end of week 8 (before treatment with antibiotics), and at the end of the experiment (after treatment with antibiotics) were analysed using 16S rRNA sequencing (n = 10 per group) (**e**-**g**). Bacterial quantification via 16S qPCR (**e**), principal coordinate analysis (**f**), and proportional abundance of bacteria (**g**) demonstrate a great impact of the antibiotic treatment (ABx) on microbiome abundance and composition. **h**, Plasma levels of total cholesterol at the end of the experiment (n = 10 per group). **i**, *Ldlr*[−/−]/*Rag2*[−/−] male mice were put on 6 weeks of cWD (n = 8) versus iWD (n = 7). Calculation of plasma cholesterol levels over the whole period of the experiment. Mean ± s.e.m.; two-tailed unpaired *t*-test (**h**, **i**). *P* values are indicated on the graphs.

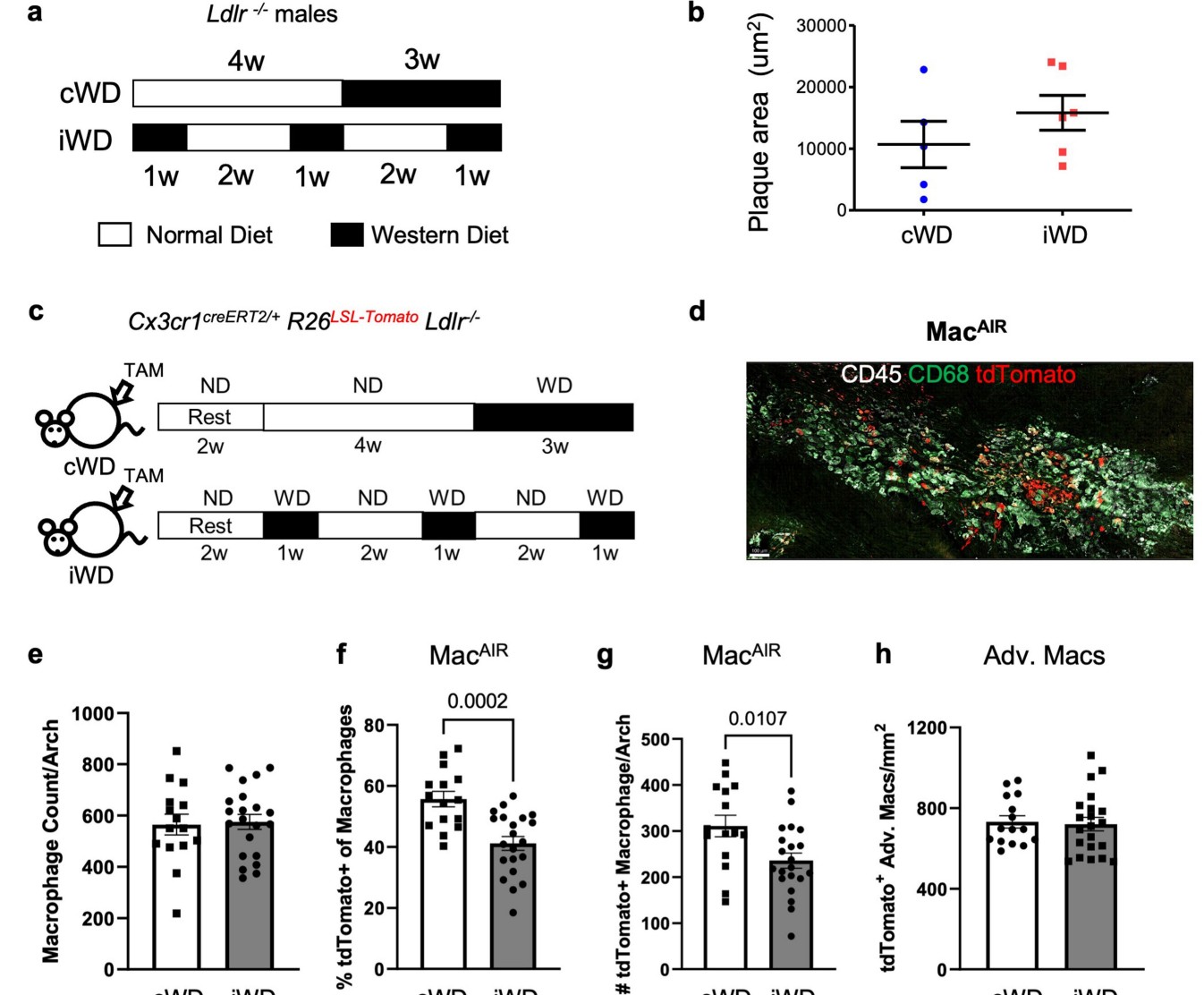

**Extended Data Fig. 5 | Impact of early intermittent hyperlipidaemia on atherosclerosis and resident arterial macrophages in mice. a**, LDL receptor-deficient (*Ldlr*[−/−]) female mice were fed a Western-type diet (WD) for 3 weeks (w) either continuously (cWD) or intermittently (iWD). **b**, Mean atherosclerotic plaque size in the aortic sinus of the 2 groups of mice is shown (n = 5 cWD and n = 6 iWD). **c**, Experimental set up. *Cx3cr1*[creERT2/+] *Rosa26*[LSL-Tomato] *Ldlr*[−/−] mice on normal diet (ND) received tamoxifen treatment (TAM) to induce Tomato expression. The mice were rested for 2 weeks on normal diet to allow for clearance of labelled blood monocytes. The mice were then placed on 3 weeks of cWD versus 3 weeks of iWD. Mice were killed at the end of the experiment for analysis of fate-mapped Tomato⁺ resident arterial macrophages (n = 15 cWD; n = 21 iWD). **d**, Representative photomicrographs showing staining for macrophages (CD45⁺ CD68⁺) to quantify the numbers of intimal Mac^AIR macrophages. **e**, Total CD45⁺ CD68⁺ macrophage count in the aortic arch of the 2 groups of mice at the end of the experiment. **f**, Percentage of Tomato⁺ CD45⁺ CD68⁺ macrophages in the intima (Mac^AIR) (among intimal CD45⁺ CD68⁺ macrophages) of the 2 groups of mice. **g**, Number of Tomato⁺ CD45⁺ CD68⁺ intimal (Mac^AIR) macrophages. **h**, Number of Tomato⁺ adventitial (adv.) macrophages. n = 15 cWD and n = 21 iWD (**e**-**h**). Mean ± s.e.m.; two-tailed unpaired *t*-test (**b**, **e**-**h**). *P* values are indicated on the graphs.

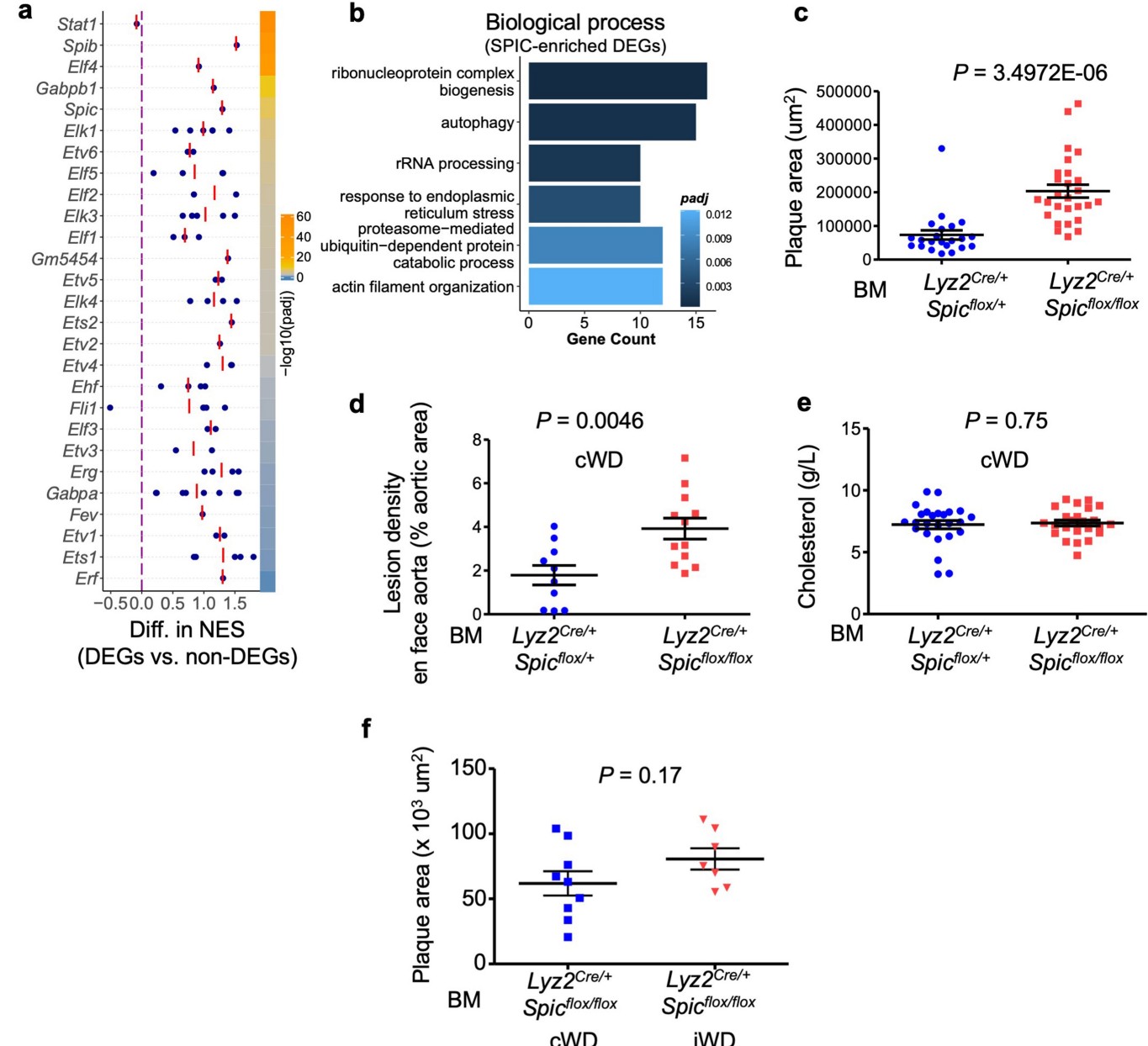

**Extended Data Fig. 6 | Impact of macrophage expression of SPIC on the development of atherosclerosis in mice. a**, Dotplot (n = 1 to 8) with mean (red bar) for common transcription factor (TF) binding motifs enriched in significant DEGs and their corresponding normalized enrichment score (NES) difference with nonsignificant DEGs. The order of the TFs corresponds to the mean of *Padj* value. Two-tailed proportion *z*-test performed then adjusted by Benjamini-Hochberg (BH) correction. SPIC is highlighted. Detailed information is provided in Supplementary Table 4. **b**, Barplot of selected 6 Gene Ontology (Biological Process) pathways for significant DEGs enriched in SPIC binding motifs. Gradient blue colour represents the padj values (one-sided Fisher exact test then adjusted by BH correction), and the size of the bar represents the number of genes that contribute to the corresponding pathways. Detailed information is provided in Supplementary Table 5. **c**, *Ldlr*[-/-] female mice were

lethally irradiated and reconstituted with bone marrow (BM) from *Lyz2*[Cre+/-] *Spic*[flox/flox] mice (n = 27) and *Lyz2*[Cre+/-] *Spic*[flox/+] controls (n = 22) (see Methods). After 5 weeks of recovery mice were put on cWD for 8 weeks to assess the role of myeloid-specific expression of SPIC on atherogenesis. Mean atherosclerotic plaque size in the aortic sinus of the 2 groups of mice is shown. **d**, Mean atherosclerotic plaque size in the en face aorta of the 2 groups of mice (n = 12 *Lyz2*[Cre+/-] *Spic*[flox/flox] and n = 10 *Lyz2*[Cre+/-] *Spic*[flox/+]). **e**, Mean plasma total cholesterol levels at the end of the experiment (n = 24 *Lyz2*[Cre+/-] *Spic*[flox/flox] and n = 25 *Lyz2*[Cre+/-] *Spic*[flox/+]). **f**, *Ldlr*[-/-] female mice were lethally irradiated and reconstituted with *Lyz2*[Cre+/-] *Spic*[flox/flox] bone marrow. After recovery, mice were put on 6 weeks of cWD (n = 9) or 6 weeks iWD (n = 7). Mean atherosclerotic plaque size in the aortic sinus of the 2 groups of mice. Mean ± s.e.m.; two-tailed unpaired *t*-test (**c-f**). *P* values are indicated on the graphs.

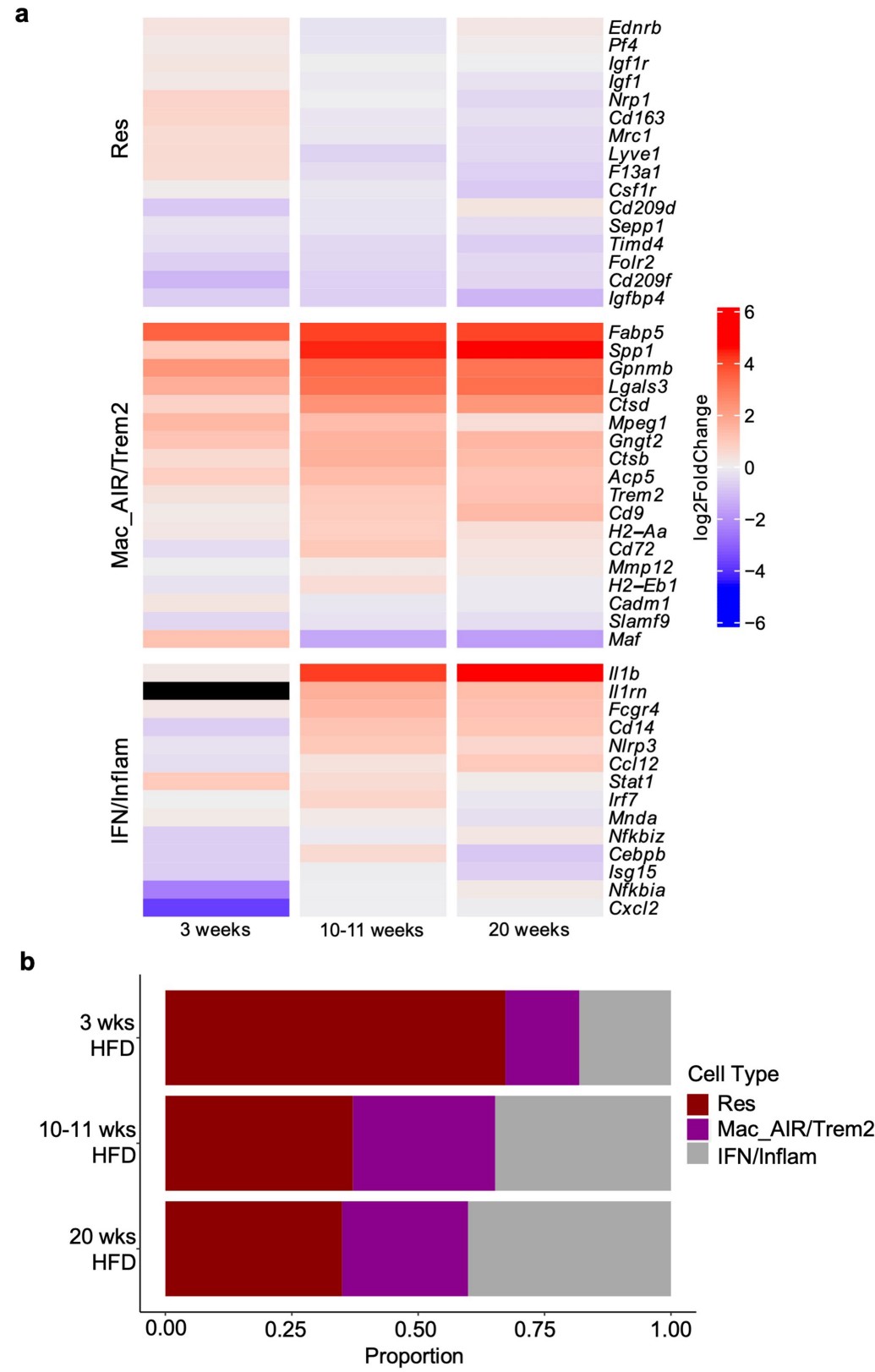

**Extended Data Fig. 7 | Time-dependent changes in the expression of prototypical genes associated with distinct arterial macrophage subsets in *Ldlr*⁻/⁻ mice subjected to continuous western diet. a**, A heatmap of selected markers for each macrophage subtype (Res = Resident; MacAIR/Trem2; IFN/Inflam = interferon/inflammatory) derived from analysis of the mouse public data set on scRNASeq of atherosclerotic lesions in *Ldlr*⁻/⁻ mice subjected to various durations of continuous high fat diet (HFD). Hierarchical clustering is shown based on log2FoldChange. **b**, Barplot for three macrophage subtype proportions for each duration of continuous HFD.

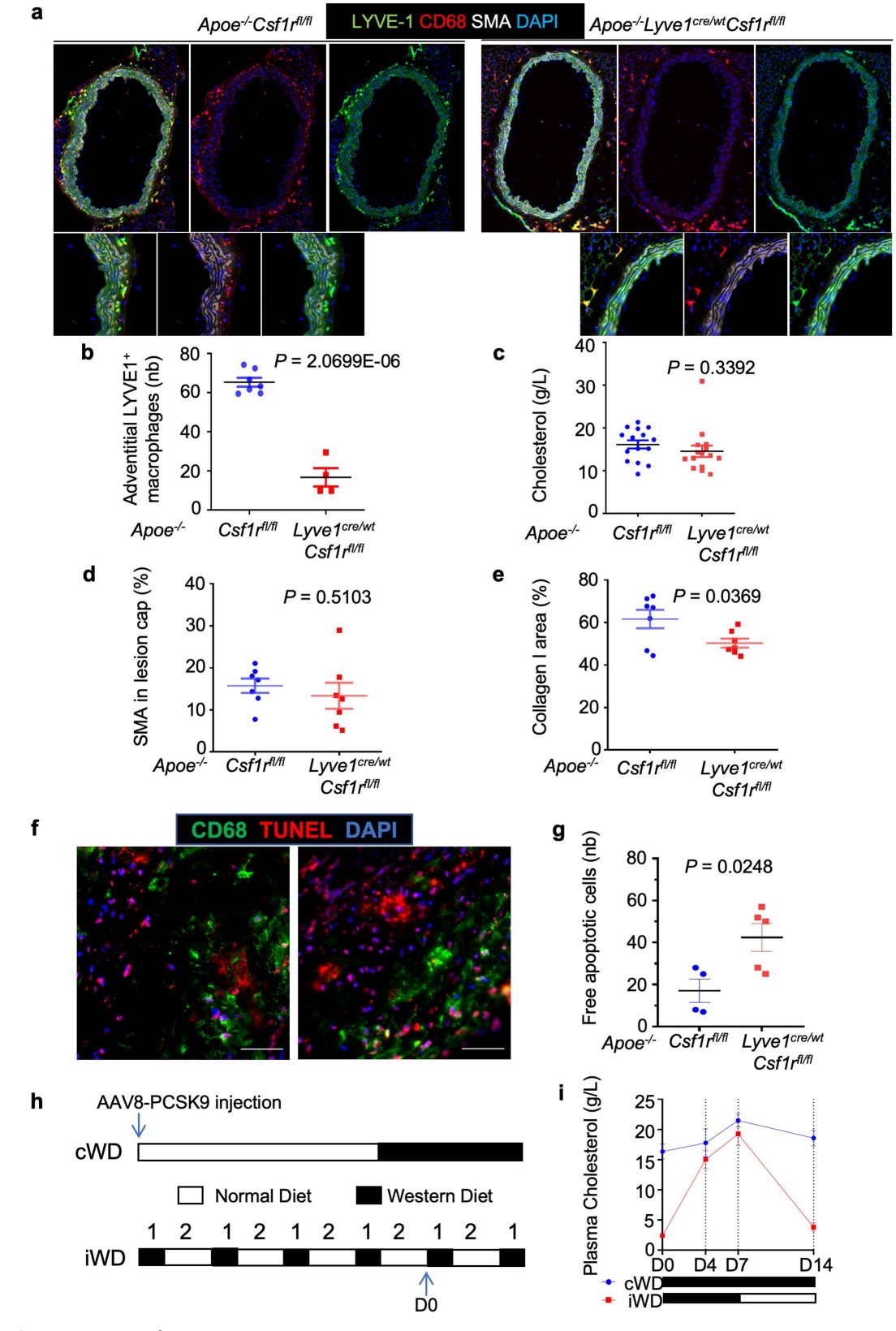

**Extended Data Fig. 8** | See next page for caption.

**Extended Data Fig. 8 | Deletion of LYVE1$^+$ resident macrophages in atherosclerotic mice. a**, Representative examples of staining for LYVE1, CD68, SMA and DAPI in arteries of $Apoe^{-/-}$ $Csf1r^{flox/flox}$ and $Apoe^{-/-}$ $Lyve1^{Cre+/-}$ $Csf1r^{flox/flox}$ mice. **b**, Quantification of adventitial LYVE1$^+$ macrophages in the 2 groups of mice (n = 7 $Csf1r^{flox/flox}$ and n = 4 $Lyve1^{Cre+/-}$ $Csf1r^{flox/flox}$). **c**, Plasma levels of total cholesterol at the end of the experiment in the different groups of mice fed on cWD for 18 weeks (n = 15 $Apoe^{-/-}$ $Csf1r^{flox/flox}$ and n = 15 $Apoe^{-/-}$ $Lyve1^{Cre+/-}$ $Csf1r^{flox/flox}$ mice. **d**, Quantification of SMA$^+$ area in lesion cap of the aortic sinus (n = 7 per group). **e**, Quantification of collagen I area in the plaques of the aortic sinus (n = 7 per group). Mean ± s.e.m.; two-tailed unpaired $t$-test (**b**-**e**). $P$ values are indicated on the graphs. **f**, Representative photomicrographs (scale bar = 50 μm) of plaques stained with anti-CD68, DAPI, and TUNEL. **g**, Number of free apoptotic (TUNEL$^+$DAPI$^+$) cells in aortic lesions of $Apoe^{-/-}$ $Csf1r^{flox/flox}$ (n = 4) and $Apoe^{-/-}$ $Lyve1^{Cre+wt}$ $Csf1r^{flox/flox}$ (n = 5) were quantified (see Methods). Mean ± s.e.m.; two-tailed unpaired $t$-test. $P$ values are indicated on the graphs. **h**, Experimental design. $Csf1r^{flox/flox}$ and $Lyve1^{Cre+/-}$ $Csf1r^{flox/flox}$ female mice were injected with AAV8-D377Y-mPCSK9 to induce hyperlipidaemia. The mice were then subjected to 6 weeks of cWD or 6 weeks of iWD. $Csf1r^{flox/flox}$ mice (n = 8 cWD and n = 9 iWD), and $Lyve1^{Cre+/-}$ $Csf1r^{flox/flox}$ mice (n = 11 cWD and n = 11 iWD) (see Methods). **i**, Representative values of plasma cholesterol levels at the indicated time points. Mean ± s.e.m. (n = 5 per group).

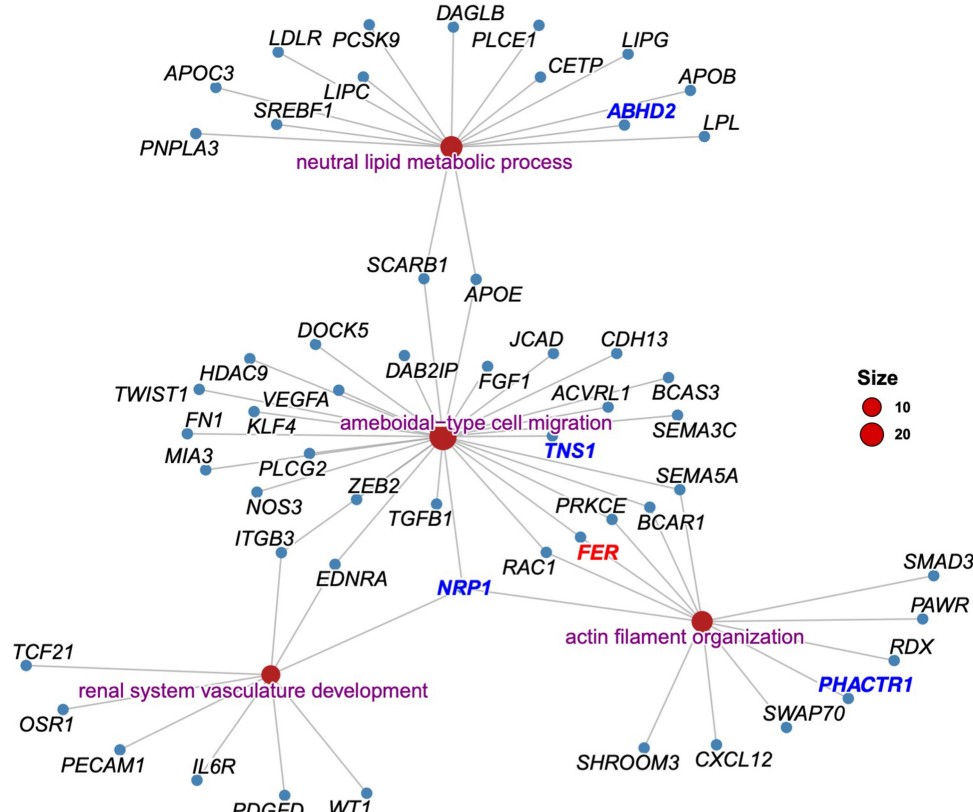

**Extended Data Fig. 9 | Gene ontology enrichment analysis on 220 human coronary artery disease causal genes.** Gene ontology enrichment analysis of biological processes (BP) was conducted on 220 coronary artery disease causal genes prioritized in Aragam et al.[33]. A total of 178 out of 220 causal genes contributed to the GO enrichment analysis, and 867 enriched BP were identified (see Supplementary Table 11). The most enriched pathways and their corresponding causal gene network are plotted here using cnetplot function in R. Significant differentially expressed genes in our RNASeq data of aortic macrophages from iWD vs cWD (see Supplementary Table 1) are shown here (red: upregulated in iWD, blue: downregulated in iWD). NRP1 is represented in the most enriched biological pathways, which are related to cell migration, actin filament organization, and vascular development.

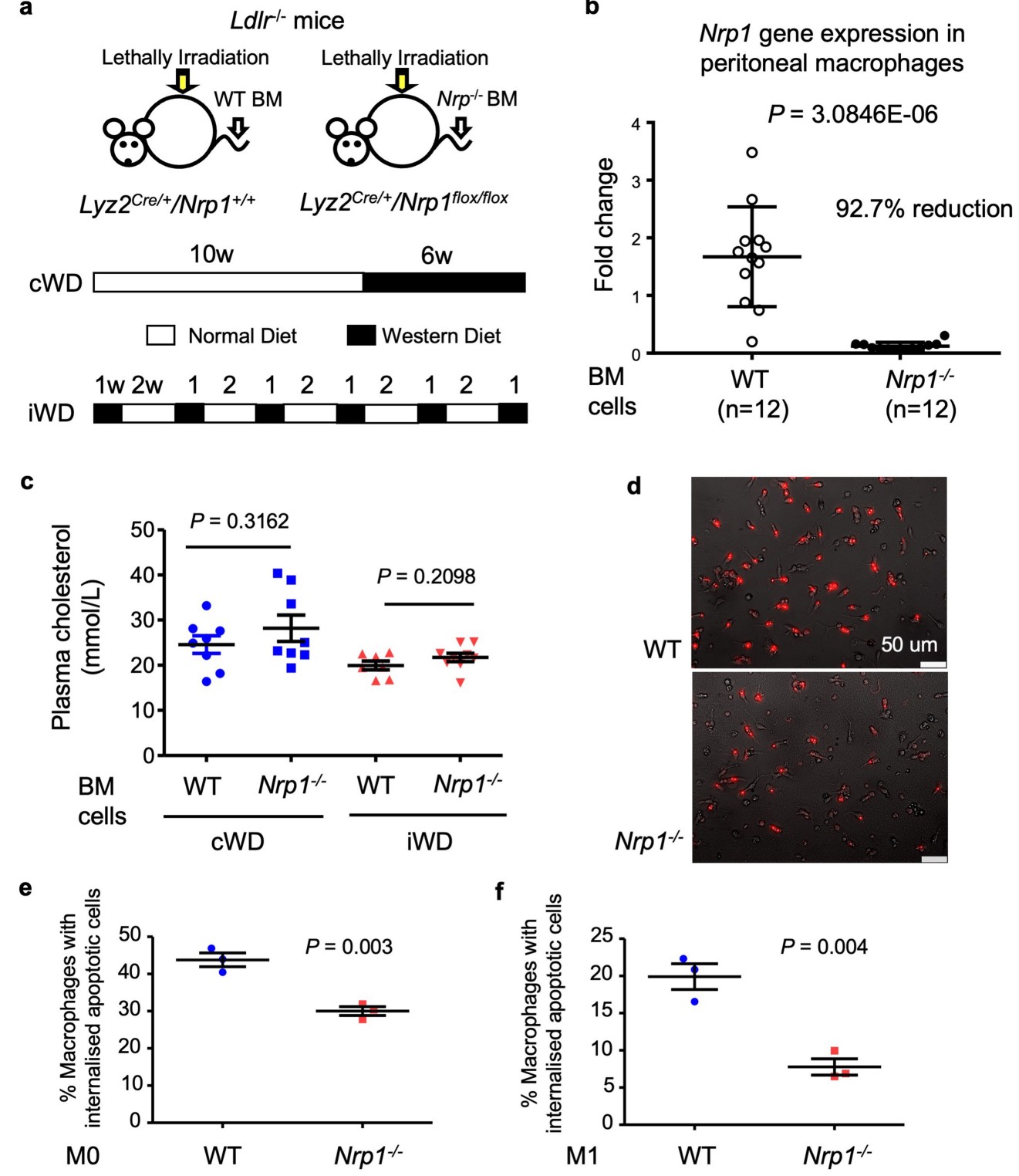

**Extended Data Fig. 10 | Deletion of NRP1 expression in myeloid cells of atherosclerotic mice. a**, Experimental set up. LDL receptor-deficient (*Ldlr*[-/-]) male mice were lethally irradiated and reconstituted with *Lyz2*[*Cre+/–*] *Nrp1*[*flox/flox*] (*Nrp1*[-/-]) or *Lyz2*[*Cre+/–*] *Nrp1*[+/+] (WT) bone marrow. After recovery, mice were put on a Western-type diet (WD) for 6 weeks (w) either continuously (cWD) or intermittently (iWD). **b**, *Nrp1* gene expression by qPCR in peritoneal macrophages at the end of the experiment (n = 12 per group). Fold change is relative to *Rplp0* (36B4) expression. **c**, Mean plasma total cholesterol levels at the end of the experiment in the 4 groups of mice (n = 8 cWD WT and *Nrp*[-/-], n = 7 iWD WT, and n = 9 iWD *Nrp*[-/-]). **d**, Bone-marrow-derived macrophages were generated from *Lyz2*[*Cre+/–*] *Nrp1*[+/+] (WT) and *Lyz2*[*Cre+/–*] *Nrp1*[*flox/flox*] mice and cultured with PKH26-labelled apoptotic Jurkat cells (see Methods). **e**, **f**, Quantification of the percentage of macrophages (M0 in **e** and M1 in **f**) with internalized apoptotic cells; two separate experiments with n = 3 biological replicates/group. Mean ± s.e.m.; two-tailed unpaired *t*-test. *P* values are indicated on the graphs.

# Reporting Summary

## Statistics

For all statistical analyses, confirm that the following items are present in the figure legend, table legend, main text, or Methods section.

| n/a | Confirmed | |
|---|---|---|
| ☐ | ☒ | The exact sample size (*n*) for each experimental group/condition, given as a discrete number and unit of measurement |
| ☐ | ☒ | A statement on whether measurements were taken from distinct samples or whether the same sample was measured repeatedly |
| ☐ | ☒ | The statistical test(s) used AND whether they are one- or two-sided<br>*Only common tests should be described solely by name; describe more complex techniques in the Methods section.* |
| ☐ | ☒ | A description of all covariates tested |
| ☐ | ☒ | A description of any assumptions or corrections, such as tests of normality and adjustment for multiple comparisons |
| ☐ | ☒ | A full description of the statistical parameters including central tendency (e.g. means) or other basic estimates (e.g. regression coefficient) AND variation (e.g. standard deviation) or associated estimates of uncertainty (e.g. confidence intervals) |
| ☐ | ☒ | For null hypothesis testing, the test statistic (e.g. *F*, *t*, *r*) with confidence intervals, effect sizes, degrees of freedom and *P* value noted<br>*Give P values as exact values whenever suitable.* |
| ☒ | ☐ | For Bayesian analysis, information on the choice of priors and Markov chain Monte Carlo settings |
| ☒ | ☐ | For hierarchical and complex designs, identification of the appropriate level for tests and full reporting of outcomes |
| ☒ | ☐ | Estimates of effect sizes (e.g. Cohen's *d*, Pearson's *r*), indicating how they were calculated |

*Our web collection on statistics for biologists contains articles on many of the points above.*

## Software and code

Policy information about availability of computer code

| Data collection | Reported in manuscript and in Supplementary Table 24 |
|---|---|
| Data analysis | Reported in Methods section. Softwares used in the research are publicly available and are described in Supplementary Table 24. Code deposition released to publicly available with Github link https://github.com/CAD-ZM-BFX/Takaoka_Mallat (DOI: https://doi.org/10.5281/zenodo.13137366). |

For manuscripts utilizing custom algorithms or software that are central to the research but not yet described in published literature, software must be made available to editors and reviewers. We strongly encourage code deposition in a community repository (e.g. GitHub). See the Nature Portfolio guidelines for submitting code & software for further information.

## Data

Policy information about availability of data

All manuscripts must include a data availability statement. This statement should provide the following information, where applicable:
- Accession codes, unique identifiers, or web links for publicly available datasets
- A description of any restrictions on data availability
- For clinical datasets or third party data, please ensure that the statement adheres to our policy

All the associated raw data presented in this paper are available from the corresponding author upon reasonable request. Source data are provided with this paper.

Raw RNA-sequencing data is publicly accessible with accession number E-MTAB-12759 (https://www.ebi.ac.uk/biostudies/arrayexpress/E-MTAB-12759) and E-MTAB-12761(https://www.ebi.ac.uk/biostudies/arrayexpress/studies/E-MTAB-12761).

## Human research participants

Policy information about studies involving human research participants and Sex and Gender in Research.

| | |
|---|---|
| Reporting on sex and gender | Reported |
| Population characteristics | This is described in Methods and Supplementary Methods and patients characteristics are provided in Supplementary Tables 14 and 15. "Multiple imputation was undertaken for the covariates of youth age, ever smoked, family history of cardiovascular disease, education (number of years studied), and cumulative body mass index, systolic blood pressure and high-density cholesterol, using predictive mean matching via the R library, mice". |
| Recruitment | Described |
| Ethics oversight | Ethics approval for the baseline study was granted by the University of Turku Faculty of Medicine Ethical Committee (letter dated 14/11/1978, minutes 3/1978) and for the most recent follow-up, granted by the Hospital District of Southwest Finland (Ethics approval number ETMK:68/1801/2017). |

Note that full information on the approval of the study protocol must also be provided in the manuscript.

# Field-specific reporting

Please select the one below that is the best fit for your research. If you are not sure, read the appropriate sections before making your selection.

☒ Life sciences ☐ Behavioural & social sciences ☐ Ecological, evolutionary & environmental sciences

For a reference copy of the document with all sections, see nature.com/documents/nr-reporting-summary-flat.pdf

# Life sciences study design

All studies must disclose on these points even when the disclosure is negative.

| | |
|---|---|
| Sample size | We stated in the manuscript that "Sample size was determined to detect a significant difference (alpha=0.05 and 80% power) of at least 30% in lesion size between groups." |
| Data exclusions | No data exclusion |
| Replication | We indicated that "Main atherosclerosis experiments were repeated several times and showed consistent results (see Results)". |
| Randomization | We indicated that "Mice were randomly assigned to their respective experimental groups." The human data are derived from the Young Finns Study cohort and randomization is not applicable. |
| Blinding | For Atherosclerosis studies, we indicated that "Analyses were performed in a blinded manner". For staining quantification, we stated that "Data analysis was conducted in a blind manner where appropriate". For human carotid plaque measurements, we indicated that "Carotid artery plaque measures were performed off-line by a single reader blinded to participant's details". |

# Reporting for specific materials, systems and methods

We require information from authors about some types of materials, experimental systems and methods used in many studies. Here, indicate whether each material, system or method listed is relevant to your study. If you are not sure if a list item applies to your research, read the appropriate section before selecting a response.

## Materials & experimental systems

| n/a | Involved in the study |
|---|---|
| ☐ | ☒ Antibodies |
| ☐ | ☒ Eukaryotic cell lines |
| ☒ | ☐ Palaeontology and archaeology |
| ☐ | ☒ Animals and other organisms |
| ☒ | ☐ Clinical data |
| ☒ | ☐ Dual use research of concern |

## Methods

| n/a | Involved in the study |
|---|---|
| ☒ | ☐ ChIP-seq |
| ☒ | ☐ Flow cytometry |
| ☒ | ☐ MRI-based neuroimaging |

# Antibodies

| | |
|---|---|
| Antibodies used | Antibodies are described in detail in Methods.<br>Primary mouse antibodies used for immunofluorescence analysis included anti-MOMA2 (rabbit monoclonal; Biorad MCA519G; 1:200), anti-CD68 (rat monoclonal; Bio-Rad MCA1957; 1:300), anti-CD206 (rat monoclonal; Bio-Rad MCA2235; 1:200), anti-CD3 (rabbit polyclonal, DAKO/Agilent A045229-2; 1:200), anti-LYVE-1 (rabbit monoclonal; Abcam ab218535; 1:600), anti-CD31 (Armenian-Hamster monoclonal; Millipore MAB1398Z; 1:200), anti-Collagen Type I (rabbit polyclonal; Millipore ABT257; 1:300), and Cy3-conjugated anti-smooth muscle actin (clone 1A4; Sigma-Aldrich C6198; 1:1000). CD45 (clone 30-F11, BioLegend, 0.5 ug/mL) and CD68 (clone FA-11, BioLegend, 1 ug/mL). Rat IgG (eBioscience) and rabbit IgG (Jackson ImmunoResearch) were used for isotype controls. AF488- (1:300), AF555- (1:300), Cy3- (1:500), AF647 (1:300)-conjugated antibodies (Jackson ImmunoResearch) were used for fluorescence detection.<br>Primary human antibodies were: CD68 (Agilent DAKO, M087601-2, Clone PG-M1, Monoclonal Mouse Anti-Human, 1:50); Lyve1 (R&D Systems, AF2089, Polyclonal, Goat Anti-Human, 1:100). |
| Validation | Validation of antibodies was provided by the manufacturer, and this quality control data can be accessed through the supplier's website. In addition, all antibody clones have been widely used and tested in research with numerous prior publications validating their specificity.<br>Well-validated human antibodies were purchased from established commercial vendors, including Agilent, and, R&D Systems. Unless otherwise noted, antibodies were used at manufacturer- and primary literature-validated concentrations for the relevant assays, as detailed below:<br>CD68 (Agilent DAKO, M087601-2, Clone PG-M1, Monoclonal Mouse Anti-Human, 1:50): https://www.agilent.com/store/productDetail.jsp?catalogId=M087601-2&catId=SubCat3ECS_86401;<br>Lyve1 (R&D Systems, AF2089, Polyclonal, Goat Anti-Human, 1:100): https://www.rndsystems.com/products/human-lyve-1-antibody_af2089.<br>Well-validated mouse antibodies were purchased from established commercial vendors, including Bio-Rad, Merck Millipore, Abcam, Agilent DAKO and Sigma-Aldrich. Unless otherwise noted, antibodies were used at manufacturer- and primary literature-validated concentrations for the relevant assays, as detailed below:<br>MOMA2 (Monoclonal, Rat anti-mouse; Bio-Rad MCA519G, 1:200) https://www.bio-rad-antibodies.com/monoclonal/mouse-macrophages-monocytes-antibody-moma-2-mca519.html?f=purified<br>CD68 (Monoclonal, Rat anti-mouse; Bio-Rad MCA1957, 1:300) https://www.bio-rad-antibodies.com/monoclonal/mouse-cd68-antibody-fa-11-mca1957.html?f=purified<br>CD206 (Monoclonal, Rat anti-mouse ; Bio-Rad MCA2235, 1:200) https://www.bio-rad-antibodies.com/monoclonal/mouse-cd206-antibody-mr5d3-mca2235.html?f=purified<br>CD3 (Polyclonal, Rabbit anti-human/mouse DAKO/Agilent A045229-2, 1:200) https://www.agilent.com/store/en_US/Prod-A045229-2/A045229-2<br>LYVE-1 (Monoclonal, Rabbit anti-mouse; Abcam ab218535, 1:600) https://www.abcam.com/en-us/products/primary-antibodies/lyve1-antibody-epr21771-ab218535<br>CD31 (Monoclonal, Armenian-Hamster anti-mouse; Merck Millipore MAB1398Z, 1:200) https://www.merckmillipore.com/SG/en/product/Anti-PECAM-1-Antibody-clone-2H8-Azide-Free,MM_NF-MAB1398Z?ReferrerURL=https%3A%2F%2Fwww.google.com%2F<br>Collagen Type I (Polyclonal, Rabbit anti-mouse; Merck Millipore ABT257, 1:300) https://www.merckmillipore.com/SG/en/product/Anti-Pro-Collagen-Type-I-A1-COL1A1,MM_NF-ABT257<br>Smooth muscle actin (Monoclonal, Cy3-conjugated anti-mouse, clone 1A4; Sigma-Aldrich C6198, 1:1000) https://www.sigmaaldrich.com/SG/en/product/sigma/c6198?srsltid=AfmBOopf92OIh30s832a0Cvc6x3BCYiW3LEVIb_gxpzGwLQJ7bSM0Apn<br>CD45 (clone 30-F11, BioLegend, 0.5ug/mL; catalog #103126) and CD68 (clone FA-11, BioLegend, 1 ug/mL, catalog #137020). |

# Eukaryotic cell lines

Policy information about cell lines and Sex and Gender in Research

| | |
|---|---|
| Cell line source(s) | Jurkat cells clone E6-1, ATCC. |
| Authentication | Jurkat cells have not been authenticated by our lab. |
| Mycoplasma contamination | The Jurkat cell line used in the experiment tested negative for mycoplasma at first provision. However, it has not been re-tested later on. |
| Commonly misidentified lines<br>(See ICLAC register) | n/a |

# Animals and other research organisms

Policy information about studies involving animals; ARRIVE guidelines recommended for reporting animal research, and Sex and Gender in Research

| | |
|---|---|
| Laboratory animals | All mice were on a C57BL/6 background. Both males and females were used. Atherosclerosis experiments were started at the age of 6 weeks unless otherwise stated. Lyz2Cre+/–, Ldlr–/– and Rag2–/– mice were initially from Charles River. Spicflox/flox mice were from Tsuneyasu Kaisho, Department of Immunology, Institute of Advanced Medicine, Wakayama Medical University, Japan, and Dr. Wataru Ise, Laboratory of Lymphocyte Differentiation, World Premier International Immunology Frontier Research Center, Osaka University, Japan44. Lyz2Cre+/– Nrp1flox/flox and Lyz2Cre+/– Nrp1+/+ mice were generated by the group of Christiana Ruhrberg, University College London, UK. Cx3cr1creERT2/+ Rosa26LSL-Tomato Ldlr–/– mice were generated by Jesse Williams, University of Minnesota Medical School, Minneapolis, USA. Apoe-/- mice were purchased from Comparative Medicine (National University of Singapore). Lyve1cre/cre and Csf1rflox/flox mice were purchased from Jackson Laboratory. Apoe-/-Lyve1cre/wtCsf1rflox/flox mice |

were generated by crossbreeding Lyve1cre/cre with Apoe-/-Csf1rflox/flox mice.
Mice were housed in 12h/12h dark/light cycle, at ambient temperature of 22 degrees Celsius and 45%-65% humidity.

Wild animals

We indicated that "No wild animals and no field-collected samples were used in the study".

Reporting on sex

Reported. We indicated that "Both males and females were used". Sex is clearly identified in legends of figures and extended dat figures.

Field-collected samples

We indicated that "No wild animals and no field-collected samples were used in the study".

Ethics oversight

We stated that " All in vivo experiments using mice were approved by local Institutional Review Boards: the Home Office, UK, under PPL PA4BDF775; the Ethical Committee of INSERM (APAFIS #29371) and the French Ministry of Agriculture (MESRI 674 #29371); the Institutional Animal Care and Use Committee (IACUC) at National University of Singapore (protocol number R21-1562); and under animal protocol 2111-39587A for experiments performed in the USA".

Note that full information on the approval of the study protocol must also be provided in the manuscript.

