## [Peer Review File · Nature]

Manuscript Title: Early intermittent hyperlipidaemia alters tissue macrophages to fuel atherosclerosis

Editorial Notes:

Redactions – unpublished data

Reviewer Comments & Author Rebuttals

Reviewer Reports on the Initial Version:

Referees' comments:

Referee #1 (Remarks to the Author):

Takaoka and colleagues demonstrate in a murine model of atherosclerosis that early intermittent hyperlipidemia leads to increased atherosclerosis compared to continuous Western diet exposure. To identify the mechanisms that could explain what causes the excess atherosclerosis burden found in early intermittent hyperlipidemia, the authors use comparative transcriptomic analysis, transcription factor binding analysis and identify links to human genetic susceptibility genes. This discovery work was then followed up by in vivo analysis using fate mapping, cell-specific gene deletion and cell depletion approaches.

Together, these types of studies suggest that early intermittent hyperlipidemia reduces atheroprotective functions in resident arterial macrophages, which causes the accelerated atherosclerosis phenotype. This work challenges the prevailing notion that atherosclerosis is mainly driven by cumulative exposure to excess cholesterol and suggests that intermittent hyperlipidemia, even early in life, can contribute significantly to atherogenesis. Notably, the mouse atherosclerosis studies are backed up by human translational studies in human healthy and atherosclerotic coronary arteries showing a reduction of resident LYVE1 positive macrophages in the adventitia of atherosclerotic coronary arteries. Furthermore, epidemiological data back up the overall conclusions.

Together, the authors identify new atheroprotective functions of resident macrophages in vessels that are altered due to early intermittent hyperlipidemia. This discovery is of general interest to the readership of Nature.

Critiques remain:

1. While the authors identify pathways related to actin filament organization in resident macrophages, they fail to explain how these pathways mediate the protection from atherogenesis induction. How do SPIC and NRP1 alter macrophage function in the context of atherogenesis?
2. The identification of potentially atheroprotective macrophages is of interest to the field, but the function of these cells has not been studied, and it remains ill-defined what these cells do to protect from atherogenesis.

3. Much of the work is focused on early hyperlipidemia, but I see no comparison towards hyperlipidemia at later time points of murine life. It would be beneficial to compare these paradigms at different ages of mice.

SEP

Referee #2 (Remarks to the Author):

This manuscript describes studies in mice in which intermittent western-type diet (iWD) feeding results in greater atherosclerosis than continuous WD (cWD) feeding for the same overall period of WD and similar effects on integrated plasma cholesterol. The authors rule out effects on the gut microbiome as a contributor to the difference in atherosclerosis. Based on aorta RNAseq, they show that iWD has an effect on the phenotype of resident arterial macrophages compared with cWD. Studies of macrophage-specific deletion of Spic and Lyve1+ under cWD conditions confirm their contributions to atherosclerosis in this context. Macrophage-specific deletion of Nrp1 exacerbated atherosclerosis in cWD but not iWD fed mice. A Finnish study of cIMT implicates the importance of non-HDL-cholesterol earlier in life in carotid atherosclerosis.

Specific comments:

1. RNAseq of aortas suggests an effect of iWD on autophagy and efferocytosis. Studies of the plaques showing that macrophage efferocytosis is actually impaired would help to confirm this observation based on gene expression data.
2. The studies with Spic deletion and Lyve1+ resident macrophage deletion appear to have been only under cWD and not under iWD conditions, despite the focus of this manuscript on the latter.
3. Nrp1 deletion exacerbated atherosclerosis in cWD but not iWD. It is hard to understand how Nrp1 may have a mechanistic role in the increased atherosclerosis in iWD fed mice.
4. The human data are interesting and consistent with previous data regarding the importance of non-HDL cholesterol exposure in childhood, but do not directly address the issue of intermittent high fat diet.

Referee #3 (Remarks to the Author):

Takoaka et al have submitted a paper focused on the role of intermittent western diet (iWD) triggering loss of homeostatic arterial macrophages, which then leads to increased atherosclerosis. Mechanistically, the authors link this to impaired efferocytosis and actin dynamics. The study is primarily performed in animal models and with validation in human population health studies.

The authors focus on the concept that the time spent with elevated LDL "area under the curve" , in particular during childhood, elevates risk. The initial animal studies are quite interesting. The authors compared LDLr KO mice fed a iWD vs continuous WD (cWD) and observed markedly more atherosclerosis with iWD despite similar elevation in total cholesterol. This affect was much more striking in male mice vs females. The authors found increased CD3+ T cells and macrophages in the lesions. Depletion of gut microbiota did explain the acceleration, and neither did loss of adaptive immune cells (Rag2KO).

The authors focus on resident LYVE1+ macrophages, and interesting fate mapping studies indicated decreased abundance of LYVE1+ macrophages in the Intima, which transcriptional changes suggested impaired efferocytic clearance mechanisms underlie early LDL exposure induced accelerated atherosclerosis. The authors provide evidence some of dysregulated genes in RNA-seq data correlated with GWAS related human atherosclerosis, and also examine a Finnish population study that demonstrates exposure to cholesterol in in childhood links with increased carotid atherosclerosis later in life.

Overall, the idea of the study and its clinical impact (if correct), is both novel and important. This novelty is related to 1) Marked atherosclerosis seen in a diet that has intermittent (iWD) component; 2) the role of resident, arterial MFs in this process through a defined mechanism. The study is easy to read and figures understandable. While the initial phenotype is quite interesting, in terms of magnitude of increased atherosclerosis with iWD feeding, the logic of the remainder of the study with respect the hypothesis is hard to follow. The authors raise several critical mechanistic points, however each one is not particular well explored, which reduces enthusiasm for the study.

Some of the data, for example targetting SpiC and Nrp1 - have no real relation to each other, and no obvious relationship to accelerated atherosclerosis seen in the iWD feeding - which is the novel component. The true role of resident macrophages, how iWD diet accelerates atherosclerosis, and depletion of Lyve1 MFs, deletion of SpiC and Nrp1 - all appear to be rather separate observation because they are performed under different conditions. In fact, the authors move away from iWD almost entirely in the second half of the animal experiments, thus it is not clear they are actually dissecting the mechanism. If the authors could further explore a mechanism within the context of the accelerated atherosclerosis program induced by iWD diet, that would be of greater interest than the current version of the paper.

Major Comments

1. Conceptually, the authors are examining accelerated atherosclerosis related the iWD. In this diet system, they observe increase loss of resident arterial MFs, and show a change in SpiC - a heme related TF in macrophages in bulk RNA-seq data. The interpretations of the data are unclear. For example, they fate map resident macrophages using Cx3cr1-CreERT system, and show these resident MFs are partially reduced in number, using iWD diet - suggesting their loss in the intima is a mechanism. But, to assess the role SpiC, they use LysM-Cre, and then perform BM transplant, replacing all the macrophages,

before feeding the mice a continuous WD (inexplicably, they do not use the iWD on which the phenotype was originally based, but the continuous diet). This undermines the conclusions significantly. So much has changed, it is hard to know, if as the authors claim, SpiC is restraining increased athero from iWD is resident MFs. The authors state that end of the Fig 2, that "Taken together, our data suggest that alteration of resident arterial macrophages is involved in the acceleration of atherosclerosis in response to early intermittent hyperlipidaemia." This is not case the with data shown.

2. As stated above, Fig 2 tracks a reduction is resident (?Lyve1+) macrophages in the intima in iWD diet, but no change in adventitia in LDLR background. The authors proceed to target Lyve1 MFs using on an ApoE background (changing model, although acceptable) and examine lesions at 8, 10 and 18 weeks with continuous WD. Again, I am not sure why the diet shift? I understand the model may have been generated for other reasons, but the focus of this whole figure shifts from a drop in intima LYVE1+ MFs in the prior figure, to now the adventitial LYVE1 MFs, which is a different zone. I get the Lyve1-Cre affects both zones (intimal and adventitial), but the story here is becoming confounded and not well explain. It is incumbent on the authers to link their novel model (intermittent diet) to what is presented as a novel mechanistic study with respect to resident MFs. But - the authors then need to 1) using the same diet in each experiment to prove the point; 2) focus on the same types of MFs...whether they are intimal, or adventitial....if they think that matters. It may be the compartment (intimal vs adventitia) does not matter, but the Lyve1 targeting does, but its unclear mechanistically how the happens.

3.The data regarding Nrp1 (and the justification of use in GWAS) is not related to primary hypothesis. In fact, there is no phenotype in the iWD acceleration. The novelty here being explored is acceleration of athero in the iWD. The authors appear ot have separate pieces of data , with resident MFs loss in iWD model, a phenotype with LysM-Cre in the cWD model, another phenotype if Lyve1-Cre in cWD (now on APoE background). What is needed is a more straightforward, controlled approach to define the mechanism involved.

4. The authors suggest throughout the study that efferocytosis and actin skeleton changes underlie why macrophage behave differently and how they drive the phenotype. However, this is never actually tested in vivo, or in vitro. In fact, the Nrp1 data via RNA seq in this study is in peritoneal MFs, which is really under-related to athero. References to actin dynamics and efferocytosis - which if true, would be interesting, are currently entirely unsupported. The authors need to significantly expand this section to show a mechanism.

5. The authors have used the LYve1 Cre to deplete resident MFs and see large athero lesions. Given the variability well known to athero studies, the n-value should be increased. Moreover, can the authors include some controls to show resident Lyve1 macrophages are lost? I understand Lyve1 is not detected by IF as well (being heterozygous), but what about other numbers? Are there fewer total MFs? Do they express different markers?

6. At 3 weeks post diet induction, lesion sized had not changed, and importantly - the authors focused some mechanistic studies at this time point, before the acceleration began. When I review fig S5, the

average plaque has increased by 50% in the iWD group, although there are not enough animals to conclude, that the plaque size is similar. To make this claim, that function is being assessed before plaques are larger (environment has changed), additional numbers are needed.

7. Given the effects were markedly reduced in females, the authors should clearly state where male and female mice (and patients) were used all in the figure legends. I had understood the entire study, apart from figure one experiment in the supplement was done in males. In figure 2H, it states females, is this correct? The results do not highlight this. The authors allude to different mechanisms at play in the discussion, however given the mechanism was sorted out in males only - this should be reflected in the abstract and more strongly throughout (introduction, etc). Additional mechanistic studies demonstrating key aspects of the study should be done in female animals to see if the same programs (impair efferocytosis, etc), are altered (to a lesser degree), or its different mechanism in females. Note that supplement Fig 5A states male in figure, and females in legend, similarly in Fig S6 - states female.

8. Clinical cohort data is not well explored in the Finn study. This requires more dedicated tables (listing all variables, in a univariate fashion that may or may not link to the outcome). In such an abbreviated analysis, it is difficult to ask questions. This should be stratified by male vs female sex in humans.

9. The title of the paper is "Childhood hyperlipidemia.....". Were the animal studies started in childhood? How do animal studies relate to this?

10. The authors measured total cholesterol, what about the LDL cholesterol, TG, etc with iWD diet?

Author Rebuttals to Initial Comments:

Takaoka et al-Nature 2023-03-04115-Revision

Referee #1 (Remarks to the Author):

Takaoka and colleagues demonstrate in a murine model of atherosclerosis that early intermittent hyperlipidemia leads to increased atherosclerosis compared to continuous Western diet exposure. To identify the mechanisms that could explain what causes the excess atherosclerosis burden found in early intermittent hyperlipidemia, the authors use comparative transcriptomic analysis, transcription factor binding analysis and identify links to human genetic susceptibility genes. This discovery work was then followed up by in vivo analysis using fate mapping, cell-specific gene deletion and cell depletion approaches.

Together, these types of studies suggest that early intermittent hyperlipidemia reduces atheroprotective functions in resident arterial macrophages, which causes the accelerated atherosclerosis phenotype. This work challenges the prevailing notion that atherosclerosis is mainly driven by cumulative exposure to excess cholesterol and suggests that intermittent hyperlipidemia, even early in life, can contribute significantly to atherogenesis. Notably, the mouse atherosclerosis studies are backed up by human translational studies in human healthy and atherosclerotic coronary arteries showing a reduction of resident LYVE1 positive macrophages in the adventitia of atherosclerotic coronary arteries. Furthermore, epidemiological data back up the overall conclusions.

Together, the authors identify new atheroprotective functions of resident macrophages in vessels that are altered due to early intermittent hyperlipidemia. This discovery is of general interest to the readership of Nature.

Critiques remain:

1. While the authors identify pathways related to actin filament organization in resident macrophages, they fail to explain how these pathways mediate the protection from atherogenesis induction. How do SPIC and NRP1 alter macrophage function in the context of atherogenesis?

Response:

We now provide several additional layers of evidence that support the importance of the actin filament organisation pathway in atherogenesis.

Fig. 2g and Supplementary Table 5 show that genes within the actin cytoskeleton reorganisation pathway are enriched for Spic binding motifs, and Nrp1 is among these genes. However, we do not claim that Spic and Nrp1 are directly related to each other or that they operate exclusively in the same biological pathway.

We provide further evidence for the clinical relevance of the actin filament organisation pathway in the causal pathways leading to atherosclerotic coronary artery disease in humans. In a recent Nature Genetics paper (PMID: 36474045) in which the investigators systematically characterized risk variants and genes for coronary artery disease in over a million participants, 220 candidate causal genes were prioritised combining eight complementary approaches. We applied Gene Ontology analysis on those 220 and found that the most represented biological pathways were pathways related to lipid metabolism, cell migration, and vasculature development (new Supplementary Table 11 and new Extended Fig. 10). Genes related to actin filament organization, including NRP1, were overrepresented in the pathways related to cell migration, further highlighting the relevance of our findings obtained in mice to the pathophysiology of coronary artery disease in humans. In fact, NRP1 is also represented in pathways related to “extrinsic apoptotic signaling” and “apoptotic cell clearance” in our analysis of the 220 candidate causal genes identified in GWAS (new Supplementary Table 11).

NRP1 deletion in macrophages has previously been shown to affect macrophage migration (PMID: 25271625), which may alter macrophage accumulation in atherosclerotic arteries, and therefore plaque development and progression. Actin cytoskeleton organisation is also related to efferocytosis, and we provide new data showing that NRP1 deletion in macrophages reduces macrophage efferocytosis in vitro (new Extended data Fig. 11d-11f). This is consistent with the fact that NRP1 is known to activate RAC1, the latter being a known major actor in the efferocytosis pathway. Moreover, our new analyses indicate that RAC1 is overrepresented in the GWAS-CAD pathways related to cell migration (new Supplementary Table 11 and new Extended Fig. 10), and RAC1 is also part of an apoptotic cell clearance pathway which is significantly represented in the GWAS-CAD data analysis (new Supplementary Table 11 and new Extended Fig. 10). Overall, our data provide new mechanistic evidence for the role of NRP1 in atherosclerosis.

There are several additional ways through which NRP1 and SPIC may play protective roles in atherogenesis. Previous work has shown that NRP1 deletion in macrophages promotes inflammatory responses (PMID: 28659345; PMID: 35064377), and that Spic expression in macrophages governs anti-inflammatory responses (PMID: 30061415; PMID: 32610126).

Please, also note that we have included the results of additional experiments with Spic and Nrp1 under both cWD and iWD, and have increased the number of mice used to assess atherosclerosis. Our conclusions are strengthened.

2. The identification of potentially atheroprotective macrophages is of interest to the field, but the function of these cells has not been studied, and it remains ill-defined what these cells do to protect from atherogenesis.

Response: Please, see our response above. We now pinpoint specific and important atheroprotective mechanisms.

3. Much of the work is focused on early hyperlipidemia, but I see no comparison towards hyperlipidemia at later time points of murine life. It would be beneficial to compare these paradigms at different ages of mice.

Response:

For your information only, Angeli et al have an ongoing study on the impact of ageing (without hyperlipidaemia) on the homeostatic functions of Lyve1+ macrophages. Their data indicate that old Lyve1+ macrophages (in 2 years old mice) lose their capacity to regulate collagen production by smooth muscle cells.

Therefore, we agree that, in the future, it will be interesting to compare the impact of iWD at different time points. Such studies would require numerous experiments at various ages to determine when the alteration of Lyve1+ macrophages with ageing would affect the acceleration of atherosclerosis induced by iWD. We hope the reviewer will agree that this is out of scope (of the revision) of the current work.

Nevertheless, we were able to compare *Ldlr*^{-/-} mice subjected to 6 weeks iWD started at the age of 6 weeks with mice subjected to 6 weeks of iWD started at the age of 22 weeks. We found that lesion size tended to be lower (although not statistically different) when iWD was started at 22 weeks of age compared to 6 weeks of age (data presented below for reviewer only; but we will be happy to include the data in the manuscript if requested by the reviewer). Our data suggests that the impact of iWD may become less important with advancing age, but experiments on much older mice will be required to answer this question.

Please note that 6 weeks of iWD started early (as shown above) was still more pro-atherogenic than 12 weeks of cWD started late (see cWD plaques in Fig. 2b), despite similar age of mice at sacrifice.

Referee #2 (Remarks to the Author):

This manuscript describes studies in mice in which intermittent western-type diet (iWD) feeding results in greater atherosclerosis than continuous WD (cWD) feeding for the same overall period of WD and similar effects on integrated plasma cholesterol. The authors rule out effects on the gut microbiome as a contributor to the difference in atherosclerosis. Based on aorta RNAseq, they show that iWD has an effect on the phenotype of resident arterial macrophages compared with cWD. Studies of macrophage-specific deletion of *Spic* and *lyve1+* under cWD conditions confirm their contributions to atherosclerosis in this context.

Macrophage-specific deletion of *Nrp1* exacerbated atherosclerosis in cWD but not iWD fed mice. A Finnish study of cIMT implicates the importance of non-HDL-cholesterol earlier in life in carotid atherosclerosis.

Specific comments:

1. RNAseq of aortas suggests an effect of iWD on autophagy and efferocytosis. Studies of the plaques showing that macrophage efferocytosis is actually impaired would help to confirm this observation based on gene expression data.

Response: We provide new data showing increased accumulation of apoptotic cells in atherosclerotic plaques of mice under iWD compared to cWD (new Extended data Fig. 3j) and in mice with Lyve1+ macrophage deletion (new Extended data Fig. 8f-8g). We also show that NRP1 deletion in macrophages reduces macrophage efferocytosis (new Extended data Fig. 11d-1f).

2. The studies with Spic deletion and Lyve1+ resident macrophage deletion appear to have been only under cWD and not under iWD conditions, despite the focus of this manuscript on the latter.

Response: We have conducted new experiments using cWD and iWD in mice with Spic deletion in macrophages. The results are presented in new Extended data Fig. 6d. We now show that in these mice with Spic deletion in macrophages, iWD does not lead to significant acceleration of atherosclerosis compared to cWD. This is consistent with the hypothesis that iWD accelerates atherosclerosis by downregulating Spic (and is therefore less effective at further accelerating atherosclerosis in mice with no Spic). These results are similar to the results obtained with *Nrp1* deletion in macrophages.

We also conducted new experiments using cWD and iWD in mice with Lyve1+ resident macrophage deletion. Here, we used the AAV8-D377Y-mPCSK9 model of hyperlipidaemia, which can be done directly on the *Csf1^{flox/flox}* and *Lyve1^{Cre/+} Csf1^{flox/flox}* mice without the need for the *Apoe^{-/-}* (or *Ldlr^{-/-}*) background. We show that iWD accelerates atherosclerosis in WT (*Csf1^{flox/flox}*) mice compared to cWD, but iWD does not accelerate atherosclerosis in mice with Lyve1+ resident macrophage deletion (*Lyve1^{Cre/+} Csf1^{flox/flox}* mice). The results are presented in new Extended data Fig. 9a-9j. Taken together, the data strongly support our hypothesis.

For your information, please note that we do not claim that alteration of resident macrophages is the only mechanism by which iWD accelerates atherosclerosis. Indeed, we show that, in addition to the alteration of Lyve1+ macrophages, iWD also alters MacAIR macrophages. Other immune and vascular cells may also be altered. Thus, alteration of Lyve1+ macrophages under iWD explains part but probably not all the mechanisms through which iWD increases atherogenesis. We show that Lyve1+ macrophages end up by being substantially reduced with the progression of atherosclerosis in mice (after prolonged durations of cWD) (Fig. 3) and in humans (Fig. 3). Once Lyve1+ macrophages disappear, iWD may still accelerate atherosclerosis by other mechanisms. In fact, we show that the acceleration of atherosclerosis under iWD is still present after 12 weeks of iWD (Fig. 1e-1g), which suggests that additional mechanisms may come into play once Lyve1+ macrophages have been altered, are no longer functional or have disappeared. Thus, it is perfectly plausible that alteration of resident macrophages sets the stage for further acceleration of atherosclerosis by additional mechanisms (which become revealed by the absence of resident macrophages and may not have been pro-atherogenic in the presence of resident macrophages). This will require additional studies but is now discussed in the revised version of the manuscript.

3. Nrp1 deletion exacerbated atherosclerosis in cWD but not iWD. It is hard to understand how Nrp1 may have a mechanistic role in the increased atherosclerosis in iWD fed mice.

Response: Our reasoning is the following: iWD accelerates atherosclerosis compared to cWD; iWD is associated with a substantial reduction (log2Fold change < - 5) of Nrp1 expression in macrophages compared to cWD; Deletion of Nrp1 in macrophages (to reproduce the profound reduction of Nrp1 seen in macrophages of iWD-fed mice) accelerates atherosclerosis in cWD; Thus, the profound reduction of Nrp1 expression in macrophages of iWD fed mice may account, at least in part, for the acceleration of atherosclerosis under iWD. Nrp1 deletion in macrophages did not further accelerate atherosclerosis under iWD because Nrp1 expression is already substantially reduced in macrophages of iWD-fed mice.

We also show that NRP1 deletion in macrophages reduces macrophage efferocytosis (new Extended data Fig. 11d-1f), suggesting a possible atherogenic mechanism.

Please note that in the present paper, we focussed on the pathways that were downregulated in aortic macrophages in response to iWD and were therefore hypothesised to play an athero-protective role (iWD downregulates them to accelerate atherosclerosis). Additional studies will be required to address the role of potential pro-atherogenic pathways (that are upregulated by iWD to promote atherosclerosis). The latter can potentially be found among the genes that were upregulated in our aortic macrophages in response to iWD. This is now acknowledged in the Discussion.

4. The human data are interesting and consistent with previous data regarding the importance of non-HDL cholesterol exposure in childhood, but do not directly address the issue of intermittent high fat diet.

Response: We concur that our human data are consistent with prior childhood findings. For example, our research in this same cohort found that non-HDL-C levels during childhood, adolescence, and young adulthood were equally associated with coronary artery calcification in adulthood (<https://doi:10.1001/jamacardio.2020.7238>). The novelty of our present data is in analysing the area-under-the curve for non-HDL cholesterol very early in life during childhood (individuals aged 6-12 years) and assessing its association with the occurrence of carotid plaques and the size of plaques at mid-adulthood. We show that non-HDL cholesterol during childhood is significantly associated with the presence and size of carotid plaques at mid-adulthood, whereas carotid intima-media thickness at mid-adulthood shows stronger association with non-HDL cholesterol during young adulthood.

We have now provided additional tables with patient characteristics (new Supplementary Table 14), and additional tables presenting the data according to various statistical models (new Supplementary Tables 15 to 18). The data support our pre-clinical findings that early hyperlipidaemia substantially determines future plaque development.

Our human study, constrained by the observational nature and the limited dietary information available from all life stages, could not directly address the issue of intermittent high fat diet. The complexities of questionnaire-based dietary data collection, especially over extended periods, present substantial challenges (long-term dietary tracking, reliability of self-reported data, temporal trends in dietary patterns). Despite this, we selected non-HDL-C as a lipid biomarker of a high-saturated-fat diet, which encompasses all atherogenic cholesterol. Its established causal link with atherosclerotic cardiovascular disease ([https://doi.org/10.1016/S0140-6736\(19\)32519-X](https://doi.org/10.1016/S0140-6736(19)32519-X)) and sensitivity to changes in dietary fat composition, as evidenced in DIVAS (<https://doi.org/10.3945/ajcn.114.097089>) and STRIP (<https://doi.org/10.1016/j.jpeds.2023.113776>) clinical trials, underscores the relevance of non-HDL-C. These trials demonstrate the sensitivity of non-HDL-C to dietary shifts, supporting its use in our observational context. While a deeper examination of lipids

(lipidomics) in clinical trials has indicated potential patterns in lipid traits as proxies for specific diets, including those related to a high-saturated-fat diet (<https://doi.org/10.1161/CIRCULATIONAHA.121.056805>), such data are not currently available in our cohort. Nonetheless this would also not directly address the issue of intermittent high-fat diet and does not diminish the value of non-HDL-C as a reasonable proxy for a higher saturated-fat diet. In summary, the role of intermittent high fat diet in humans will require additional studies. This is now acknowledged in the Discussion.

Referee #3 (Remarks to the Author):

Takaoka et al have submitted a paper focused on the role of intermittent western diet (iWD) triggering loss of homeostatic arterial macrophages, which then leads to increased atherosclerosis. Mechanistically, the authors link this to impaired efferocytosis and actin dynamics. The study is primarily performed in animal models and with validation in human population health studies.

The authors focus on the concept that the time spent with elevated LDL "area under the curve" , in particular during childhood, elevates risk. The initial animal studies are quite interesting. The authors compared LDLr KO mice fed a iWD vs continuous WD (cWD) and observed markedly more atherosclerosis with iWD despite similar elevation in total cholesterol. This effect was much more striking in male mice vs females. The authors found increased CD3+ T cells and macrophages in the lesions. Depletion of gut microbiota did explain the acceleration, and neither did loss of adaptive immune cells (Rag2KO).

The authors focus on resident LYVE1+ macrophages, and interesting fate mapping studies indicated decreased abundance of LYVE1+ macrophages in the Intima, which transcriptional changes suggested impaired efferocytic clearance mechanisms underlie early LDL exposure induced accelerated atherosclerosis. The authors provide evidence some of dysregulated genes in RNA-seq data correlated with GWAS related human atherosclerosis, and also examine a Finnish population study that demonstrates exposure to cholesterol in childhood links with increased carotid atherosclerosis later in life.

Overall, the idea of the study and its clinical impact (if correct), is both novel and important. This novelty is related to 1) Marked atherosclerosis seen in a diet that has intermittent (iWD) component; 2) the role of resident, arterial MFs in this process through a defined mechanism. The study is easy to read and figures understandable. While the initial phenotype is quite interesting, in terms of magnitude of increased atherosclerosis with iWD feeding, the logic of the remainder of the study with respect the hypothesis is hard to follow. The authors raise several critical mechanistic points, however each one is not particularly well explored, which reduces enthusiasm for the study.

Some of the data, for example targeting SpiC and Nrp1 - have no real relation to each other, and no obvious relationship to accelerated atherosclerosis seen in the iWD feeding - which is the novel component. The true role of resident macrophages, how iWD diet accelerates atherosclerosis, and depletion of Lyve1 MFs, deletion of SpiC and Nrp1 - all appear to be rather separate observations because they are performed under different conditions. In fact, the authors move away from iWD almost entirely in the second half of the animal experiments, thus it is not clear they are actually dissecting the mechanism. If the authors could further explore a mechanism within the context of the accelerated atherosclerosis program induced by iWD diet, that would be of greater interest than the current version of the paper.

Major Comments

1. Conceptually, the authors are examining accelerated atherosclerosis related the iWD. In this diet system, they observe increase loss of resident arterial MFs, and show a change in SpiC - a heme related TF in macrophages in bulk RNA-seq data. The interpretations of the data are unclear. For example, they fate map resident macrophages using Cx3cr1-CreERT system, and show these resident MFs are partially reduced in number, using iWD diet - suggesting their loss in the intima is a mechanism. But, to assess the role SpiC, they use LysM-Cre, and then perform BM transplant, replacing all the macrophages, before feeding the mice a continuous WD (inexplicably, they do not use the iWD on which the phenotype was originally based, but the continuous diet). This undermines the conclusions significantly. So much has changed, it is hard to know, if as the authors claim, SpiC is restraining increased athero from iWD is resident MFs. The authors state that end of the Fig 2, that "Taken together, our data suggest that alteration of resident arterial macrophages is involved in the acceleration of atherosclerosis in response to early intermittent hyperlipidaemia." This is not case the with data shown.

Response: We provided justification for the use of LysM-cre and discussed this on lines 396-399 and 526-534. We agree that BM transplant replaces all the macrophages including most resident arterial macrophages. SpiC and Nrp1 have previously been reported to be expressed preferentially (although not exclusively) in resident tissue macrophages, and our data show that SpiC and Nrp1 preferentially operate in resident macrophages. It is true that in the first version of the manuscript we had not used the SpiC mice under iWD. We now provide this data and show that iWD does not significantly accelerate atherosclerosis when SpiC is deleted in macrophages (new Extended data Fig. 6d) (which is expected according to our hypothesis, because SpiC is already downregulated in iWD), further strengthening our conclusions.

In response to the question about the relationship between SpiC and Nrp1, we do not claim that they are directly related to each other or that they operate exclusively in the same biological pathway. Nevertheless, Fig. 2g and Supplementary Table 5 suggest that SpiC and Nrp1 may both be involved in the actin cytoskeleton reorganisation pathway. The latter is one of the most differentially regulated pathways in our dataset, and we show that it is overrepresented in the biological pathways with causal link to human coronary artery disease (see response to point 1 of Reviewer 1, new Supplementary Table 11 and new Extended data Fig. 10).

2. As stated above, Fig 2 tracks a reduction in resident (?Lyve1+) macrophages in the intima in iWD diet, but no change in adventitia in LDLR background.

Response: Fig. 2b (along with data in Supplementary Tables 1 and 3) shows that at this early time point there is a significant reduction in the expression of genes associated with resident macrophages, both intimal LYVE1(-) MacAIR macrophages, and adventitial and TLF-like LYVE1+ macrophages. Fig. 2c-2e (and Extended data Fig. 5) show that there is a reduction in the numbers of MacAIR macrophages in the intima of iWD-fed compared to cWD-fed mice, which may explain, at least in part, the reduced expression of genes associated with MacAIR macrophages (Fig. 2b along with data in Supplementary Table 1 and 3). At this same time point, the number of adventitial macrophages, which are mostly LYVE1+, is not reduced in iWD mice (and this is also consistent with the fact that *Lyve1* gene expression is not different between cWD and iWD at this time point; Supplementary Table 1). As explained in the manuscript, this result indicates that at this early time point, iWD alters the gene expression profile (and potentially the functions) of LYVE1+ macrophages but their numbers are not yet altered. In brief, both types of macrophages are involved and are worth of further investigation. Since currently it is not feasible to genetically deplete MacAIR macrophages selectively, we chose to focus on resident LYVE1+ (TLF+)

macrophages and on some of their prototypical genes that were downregulated in our data set.

The authors proceed to target Lyve1 MFs using on an ApoE background (changing model, although acceptable) and examine lesions at 8, 10 and 18 weeks with continuous WD. Again, I am not sure why the diet shift? I understand the model may have been generated for other reasons, but the focus of this whole figure shifts from a drop in intima LYVE1+ MFs in the prior figure, to now the adventitial LYVE1 MFs, which is a different zone. I get the Lyve1-Cre affects both zones (intimal and adventitial), but the story here is becoming confounded and not well explain. It is incumbent on the authors to link their novel model (intermittent diet) to what is presented as a novel mechanistic study with respect to resident MFs. But - the authors then need to 1) using the same diet in each experiment to prove the point; 2) focus on the same types of MFs...whether they are intimal, or adventitial....if they think that matters. It may be the compartment (intimal vs adventitia) does not matter, but the Lyve1 targeting does, but its unclear mechanistically how the happens.

Response: The first point is about LYVE1+ macrophages and whether they are present in intima or/and adventitia. We have now clarified in the manuscript that we are talking about resident macrophages, whether adventitial or intimal. Indeed, resident macrophages with TLF-associated genes and expressing CD206 have been described and detected both in adventitia and intima, at the media border and the shoulder region (PMID: 35858629; PMID: 37943053; PMID: 30054204; PMID: 30666513). Please note that LYVE1 protein expression is detected almost exclusively in adventitial macrophages, although *Lyve1* gene expression can also be found in resident macrophages of the intima (and therefore, use of the *Lyve1 cre* model may also affect some resident macrophages in the intima). LYVE1+ macrophages of the adventitia also affect the biology of the intima. Previous work from our groups reported essential roles of adventitial LYVE1+ macrophages in arterial remodelling and stiffness (PMID: 30566884). These processes are very likely to impact the response of the artery wall to high cholesterol load.

The second point is about using cWD versus iWD to address the role of Lyve1+ macrophages in atherosclerosis. We have also addressed this comment in response to point 2 of Reviewer 2. We conducted new experiments using cWD and iWD in mice with Lyve1+ resident macrophage deletion. Here, we used the AAV8-D377Y-mPCSK9 model of hyperlipidaemia, which can be done directly on the *Csf1^{flox/flox}* and *Lyve1^{Cre/+} Csf1^{flox/flox}* mice without the need for the *ApoE^{-/-}* (or *Ldlr^{-/-}*) background. Again, we used females in these experiments due to the fact that deletion of LYVE1+ macrophages is more specific in females compared to males (see data provided below for reviewer only).

We show that iWD accelerates atherosclerosis in WT (*Csf1r^{flox/flox}*) mice compared to cWD, but iWD does not accelerate atherosclerosis in mice with Lyve1+ resident macrophage deletion (*Lyve1^{Cre/+} Csf1r^{flox/flox}* mice). The results are presented in new Extended data Fig. 9a-9j.

Please note that we have already provided data for *Nrp1* (a prototypical resident macrophage gene downregulated in iWD) under both cWD and iWD (Fig. 4i).

Taken together, the data strongly support our hypothesis.

3. The data regarding *Nrp1* (and the justification of use in GWAS) is not related to primary hypothesis. In fact, there is no phenotype in the iWD acceleration. The novelty here being explored is acceleration of athero in the iWD. The authors appear to have separate pieces of data, with resident MFs loss in iWD model, a phenotype with *LysM-Cre* in the cWD model, another phenotype if *Lyve1-Cre* in cWD (now on APoE background). What is needed is a more straightforward, controlled approach to define the mechanism involved.

Response: It is unclear why the reviewer believes that “*Nrp1* (and the justification of use in GWAS) is not related to primary hypothesis”. *Nrp1* is one of the most highly downregulated genes in our data set. We, and others, show that *Nrp1* is a prototypical gene of resident macrophages both in mice and in humans (Fig. 4d-4e). *Nrp1* is downregulated in macrophages of symptomatic versus asymptomatic carotid atherosclerotic plaques (Fig. 4c). *Nrp1* is among the genes causally involved in coronary artery disease in GWAS studies (Fig. 4a and Supplementary Table 6). Therefore, we conclude that *Nrp1* is highly relevant (please also see response to point 4 below). The reviewer believes that the fact that “there is no phenotype in the iWD acceleration” (when deleting *Nrp1* in macrophages) is against our hypothesis. This is not how we interpret the data. Our reasoning is the following: iWD accelerates atherosclerosis compared to cWD; iWD is associated with a substantial reduction (log₂Fold change < -5) of *Nrp1* expression in macrophages compared to cWD; Deletion of *Nrp1* in macrophages (to reproduce the profound reduction of *Nrp1* in macrophages of iWD-fed mice) accelerates atherosclerosis in cWD; Thus, the profound reduction of *Nrp1* expression in macrophages of iWD fed mice may account, at least in part, for the acceleration of atherosclerosis under iWD. *Nrp1* deletion in macrophages did not further accelerate atherosclerosis under iWD because *Nrp1* expression is already substantially reduced in macrophages of iWD-fed mice. Thus, the results support our hypothesis.

4. The authors suggest throughout the study that efferocytosis and actin skeleton changes underlie why macrophage behave differently and how they drive the phenotype. However, this is never actually tested in vivo, or in vitro. In fact, the Nrp1 data via RNA seq in this study is in peritoneal MFs, which is really under-related to athero. References to actin dynamics and efferocytosis - which if true, would be interesting, are currently entirely unsupported. The authors need to significantly expand this section to show a mechanism.

Response: We show that Nrp1 is among the most differentially regulated biological pathways between cWD and iWD, and is particularly related to actin filament organisation. We now provide further evidence for the clinical relevance of the actin filament organisation pathway in the causal pathways leading to atherosclerotic coronary artery disease in humans. In a recent Nature Genetics paper (PMID: 36474045) in which the investigators systematically characterized risk variants and genes for coronary artery disease in over a million participants, 220 candidate causal genes were prioritised combining eight complementary approaches. We applied Gene Ontology analysis on those 220 and found that the most represented biological pathways were pathways related to lipid metabolism, cell migration, and vasculature development (new Supplementary Table 11 and new Extended Fig. 10). Genes related to actin filament organization, including NRP1, were overrepresented in the pathways related to cell migration, further highlighting the relevance of our findings obtained in mice to the pathophysiology of coronary artery disease in humans. In fact, NRP1 is also represented in pathways related to “extrinsic apoptotic signaling” and ‘apoptotic cell clearance” in our analysis of the 220 candidate causal genes identified in GWAS (new Supplementary Table 11).

NRP1 deletion in macrophages has previously been shown to affect macrophage migration (PMID: 25271625), which may alter macrophage accumulation in atherosclerotic arteries, and therefore plaque development and progression. Actin cytoskeleton organisation is also related to efferocytosis, and we provide new data showing that NRP1 deletion in macrophages reduces macrophage efferocytosis in vitro (new Extended data Fig. 11d-11f). This is consistent with the fact that NRP1 is known to activate RAC1, the latter being a known major actor in the efferocytosis pathway. Moreover, our new analyses indicate that RAC1 is overrepresented in the GWAS-CAD pathways related to cell migration (new Supplementary Table 11 and new Extended Fig. 10), and RAC1 is also part of an apoptotic cell clearance pathway which is significantly represented in the GWAS-CAD data analysis (new Supplementary Table 11 and new Extended Fig. 10).

There are additional ways through which NRP1 may play protective roles in atherogenesis. For example, previous work has shown that NRP1 deletion in macrophages promotes inflammatory responses (PMID: 28659345; PMID: 35064377), which may be detrimental in atherosclerosis.

5. The authors have used the LYve1 Cre to deplete resident MFs and see large athero lesions. Given the variability well known to athero studies, the n-value should be increased. Moreover, can the authors include some controls to show resident Lyve1 macrophages are lost? I understand Lyve1 is not detected by IF as well (being heterozygous), but what about other numbers? Are there fewer total MFs? Do they express different markers?

Response: Despite the known variability in atherosclerosis studies, we were able to detect a significant difference in lesion size (and other parameters of plaque composition) between the 2 groups. It is difficult to justify the need for another experiment when the data is already significant (and it would take more than 8 months to produce enough animals and complete

the experiment). Please, also note that as detailed above, we have provided additional experiments in mice with Lyve1+ macrophage deletion under both cWD and iWD.

We previously showed that LYVE1+ macrophages, but not LYVE1- macrophages or monocytes, are specifically depleted in the *Lyve1*^{Cre+/-}*-Csf1*^{flox/flox} mice (Lim et al., Immunity 2018). At

the request of the reviewer, we have included additional data to show that resident macrophages are profoundly depleted in *Lyve1*^{Cre+/-}*-Csf1*^{flox/flox} mice (Extended data Fig. 8a-8b). The vast majority of adventitial macrophages are LYVE1+. Therefore, the total number of adventitial macrophages is also reduced in these mice. We also stained for another macrophage

marker CD206, previously shown by us, and others, to be highly expressed in resident macrophages. We again show that almost all LYVE1+ macrophages are CD206+ and vice versa (see figure for reviewer only). The figure shows representative photomicrographs of staining for LYVE1, CD206, CD31 and DAPI in non-atherosclerotic aortic sections of *Csf1*^{flox/flox} and *Lyve1*^{Cre+/-}*-Csf1*^{flox/flox} mice (scale bar = 200µm), and quantification of CD31⁻ adventitial LYVE1⁺CD206⁺ and LYVE1⁺CD206⁻ macrophages in *Csf1*^{flox/flox} mice, as well as the number of adventitial LYVE1⁺CD206⁺ macrophages in *Csf1*^{flox/flox} and *Lyve1*^{Cre+/-}*-Csf1*^{flox/flox} mice at the end of experiment (n=3 per group).

6. At 3 weeks post diet induction, lesion sized had not changed, and importantly - the authors focused some mechanistic studies at this time point, before the acceleration began. When I review fig S5, the average plaque has increased by 50% in the iWD group, although there are not enough animals to conclude, that the plaque size is similar. To make this claim, that function is being assessed before plaques are larger (environment has changed), additional numbers are needed.

Response: The 50% increase in lesion size at this stage is really very small when it comes to absolute plaque area (<5000 µm²), and this is quite negligible compared to the absolute difference of plaque size between cWD and iWD (~ 120,000 µm²) at the end of the 6 weeks. Please, note that this 3-week experiment has already been repeated in another lab on another continent (in Jesse William's lab in the USA) with n=15 mice in cWD group and n=21 mice in iWD group. At this very early stage, most of the fatty streak area is occupied by macrophages. The quantification of total macrophages in the fatty lesions showed similar numbers of macrophages in cWD and iWD (Extended data Fig. 5e), further consolidating our results of negligible difference, if any, in lesion size at that early stage. We refer to this in our manuscript when we say that "The total number of intimal macrophages (CD45+ CD68+)

was similar between cWD and iWD groups (Extended data Fig. 5e), consistent with the comparable plaque size between the 2 groups at this time point (Extended data Fig. 5a-5b)".

7. Given the effects were markedly reduced in females, the authors should clearly state where male and female mice (and patients) were used all in the figure legends. I had understood the entire study, apart from figure one experiment in the supplement was done in males. In figure 2H, it states females, is this correct? The results to do not highlight this. The authors allude to different mechanisms at play in the discussion, however given the mechanism was sorted out in males only - this should be reflected in the abstract and more strongly throughout (introduction, etc). Additional mechanistic studies demonstrating key aspects of the study should be done in female animals to see if the same programs (impair efferocytosis, etc), are altered (to a lesser degree), or its different mechanism in females. Note that supplement Flg 5A states male in figure, and females in legend, similarly in Flg S6 - states female.

Response: We have now reviewed all figures and figure legends and clearly indicated the sex of the mice in each experiment. We performed additional experiments in *Ldlr*^{-/-} females. iWD leads to a highly reproducible and significant increase in atherosclerosis compared to cWD. We found that the fold-increase in atherosclerosis may vary between experiments (between 1.3- and 2.5-fold increase) (Extended data Fig. 2a-2b). The reasons behind this variability in females are currently unknown and will require additional studies that are beyond the scope of the present manuscript. This is acknowledged in the revised version.

8. Clinical cohort data is not well explored in the Finn study. This requires more dedicated tables (listing all variables, in a univariate fashion that may or may not link to the outcome). In such an abbreviated analysis, it is difficult to ask questions. This should be stratified by male vs female sex in humans.

Response: We have now provided additional tables with patient characteristics (new Supplementary Table 14), and additional tables presenting studied outcomes stratified by sex (new Supplementary Tables 15 to 18). The results are in line with our pre-clinical data.

9. The title of the paper is "Childhood hyperlipidemia.....". Were the animal studies started in childhood? How do animal studies relate to this?

Response: Sexual maturity occurs very early in mice (around 35 days). However, mice of 8 weeks old are still considered relatively young. Please note that within the manuscript, "childhood" is used for human data, and "early" is used for animal data.

10. The authors measured total cholesterol, what about the LDL cholesterol, TG, etc with iWD diet?

Response: LDL cholesterol levels mirror total cholesterol levels in *Ldlr*^{-/-} mice. These have now been added to the supplementary data (Extended data Fig. 1d). TG levels also vary with iWD (see below). The TG data is more variable given that lipid measurements were not done after fasting. HDL-C levels showed no variability (see below).

Reviewer Reports on the First Revision:

Referees' comments:

Referee #1 (Remarks to the Author):

The authors have addressed my point. I would opt for including the new data on the different age groups shown to the reviewer only also in the manuscript. This would be appreciated by the readers. I have no objections and support publication of the study.

Referee #2 (Remarks to the Author):

This revised manuscript has addressed reviewer comments and added additional data in response. Overall the manuscript remains fairly descriptive and lacks an integrating molecular mechanism to explain the fundamental (and interesting) observation that intermittent western-type diet (iWD) feeding in mice results in greater atherosclerosis than continuous WD (cWD) feeding for the same overall period of WD and similar effects on plasma cholesterol.

Specific comments:

1. The human data do not directly address the fundamental issue of intermittent high fat diet.
2. The title of the manuscript 'Childhood hyperlipidemia...' should be changed as it implies a primarily human study even though the manuscript is mostly experiments in mice.

Referee #3 (Remarks to the Author):

The authors have submitted a revised paper that partially addresses my comments. Overall, I feel this paper has improved from the prior version. The novelty of the paper is focused on the fact that intermittent high fat feeding drives the phenotype of accelerated atherosclerosis, which remains in this version. The prior version of the paper alluded to novel mechanisms with a potential spatial focus (although not supported by the data) on intimal macrophages, and their loss of numbers and NRP1 in their system, but this required much more work to show. In this version of the paper the authors have revised their conclusions because they could not differentiate between intimal and adventitial macrophages with the tools available. This lessens the impact to some degree. The mechanism appears to be reduced efferocytosis, which itself is not that novel, although this still remains experimentally, poorly defined in this version. The central problems I see with the study is that deletion and targeting of cells and genes (NRP1, SpiC / NRP1 / CSF1RE does not separate recruited monocytes from macrophages, and that the mechanistic change (efferocytosis in plaque) is proven. Many of the phenotypes in the athero are monocyte driven, and this paper could have moved away from that monocyte focus and

instead explore resident MF function, but that is not proven. The key experiments, such as NRP1 experiments, which link efferocytosis to the phenotype, delete in monocytes as well MFs, which confounds much of the authors conclusions.

Comments

1. In my first original question, I stated that in the iWD they observed loss of resident intimal MFs (using Cx3cr1-CreERT2 - Mac-Air subset), and show a change in SpiC - a heme related TF in macrophages in bulk RNA-seq data. They subsequently deleted SpiC - not in resident MFs, but in all macrophages and monocytes and neutrophils and some DCs using LysM-Cre in a different model (continuous WD). These experiments were problematic for many reasons in terms of defining the role of SpiC in the resident MF compartment (which was a potential major novel mechanism). The answer they gave in the rebuttal is insufficient - the LysMcre cannot be used to study the role of the SpiC in resident macrophages, it deletes in many cell types. The authors added LysM-cre data with iWD now (previously only in continuous WD group), but this question is still not answered - we do not know what the role for SpiC is in resident arterial macrophages - the focus of their study. What is needed is deletion with Cx3cr1-CreERT or other similar tool in the resident MFs.

2. The authors conducts important experiments using Lyve1Cre-csf1r f/f mice, which constitutive delete LYVE1 MFS (adventitial and probably intimal). The presentation of the data in the supplement is in a non standard way, separating control vs Lyve1Cre - so no direct comparison is possible. The authors compare control mice to each other and Cre mice to each other. There is variability in the data and my best guess is there is weak data here to support their conclusion that resident MFs are involved in iWD and not in cWD. If all 4 groups were combined together, there would need to repeat experiments to get enough power to know if this is a specific effect in iWD. My guess is there is not, plaque are blunted in both cWD and iWD with loss of LYVE1 macrophages - the authors need to expand these studies. Moreover, this system cannot separate out recruited monocytes that become LYVE1 MFs from the resident LYVE1+ MFs - and it gets to the issues above - is this a resident MF effect, or is it also a monocyte effect. Lastly, it is essential also to show circulating monocytes numbers in each animal group in this system, but given the very mild (perhaps statistically not significant) phenotype, alternative lines of evidence that resident MFs, there before athero begins, are involved in the process.

3. The authors central hypothesis is that NRP1 loss led to impaired efferocytosis in vivo. The authors have cited a number of other studies that circumstantially make this link in atherosclerosis, which is important justification to explore in their work - but proof is needed as to the mechanism in their hands and in their mice. Is it really intimal and/or adventitial MFs that cannot efferocytosis properly that drive the phenotype, because these cells were never assessed? Why not sort these cells and test - its a standard approach. It is unfortunate that the lysM-Cre was used - as it could easily be infiltrating monocytes that cannot ingest as well. The authors need to address this experimentally, in vivo. Ideas that come to mind would be isolate MFs from the adventitia and intima (WT vs KO) and see before iWD, are they impaired, after iWD? If linked to the cx3cr1-CreERT2 labeling system, they could test individual populations (resident before athero, recruited cells that come afterwards) - these types of experiments

would go a long way in proving what happens in the plaque. In vivo or ex vivo uptake studies can be done then.

Author Rebuttals to First Revision:

Nature manuscript 2023-03-04115A

Referee #2 (Remarks to the Author):

This revised manuscript has addressed reviewer comments and added additional data in response. Overall the manuscript remains fairly descriptive and lacks an integrating molecular mechanism to explain the fundamental (and interesting) observation that intermittent western-type diet (iWD) feeding in mice results in greater atherosclerosis than continuous WD (cWD) feeding for the same overall period of WD and similar effects on plasma cholesterol.

Specific comments:

1. The human data do not directly address the fundamental issue of intermittent high fat diet.

Response: We agree and have acknowledged this limitation. Our data showing that hyperlipidaemia early in life (even in childhood and adolescence) is associated with atherosclerosis at mid-adulthood may have important clinical implication. In response to Referee 4 (expert in epidemiology who specifically reviewed the epidemiological data), we have now extended our analyses to the 2018 follow-up (participants aged 41-56 years, as compared to 21-45 years at the 2001/2007 FU). This has led to approx. 10-fold increase in the number of events studied (carotid plaque presence) from n=88 participants with plaques at the 2001/2007 FU to n=817 participants with plaques at the 2018 FU. Our conclusions that early life hyperlipidaemia is associated with atherosclerosis at mid-adulthood stand. The data are reported in the revised version of the manuscript.

2. The title of the manuscript 'Childhood hyperlipidemia...' should be changed as it implies a primarily human study even though the manuscript is mostly experiments in mice.

Response: We have changed the title to "Early intermittent hyperlipidaemia alters tissue macrophages to boost atherosclerosis".

Referee #3 (Remarks to the Author):

The authors have submitted a revised paper that partially addresses my comments. Overall, I feel this paper has improved from the prior version.

The central problem I see with the study is that deletion and targeting of cells and genes (NRP1, SpiC / NRP1 / CSF1RE) does not separate recruited monocytes from macrophages. Many of the phenotypes in the athero are monocyte driven, and this paper could have moved away from that monocyte focus and instead explore resident MF function, but that is not proven. The key experiments, such as NRP1 experiments,

which link efferocytosis to the phenotype, delete in monocytes as well MFs, which confounds much of the authors conclusions. What is needed is deletion with Cx3cr1-CreERT or other similar tool in the resident MFs.

Response: We note that this is a new request from Reviewer 3 that was not raised during the first submission. Sourcing the Cx3cr1-CreERT2 mice and crossing them to Nrp1 floxed mice, Csf1r floxed mice, and Spic floxed mice, to re-perform whole new experiments under cWD and iWD, would take approx. 18 months for each experiment to be completed.

Importantly, there are several scientific reasons why such experiments would be difficult to perform and interpret in the context of our work on atherosclerosis.

a) Resident TLF-like (Lyve1 high) arterial macrophages are Cx3cr1 low (they express low levels of Cx3cr1) while other resident Lyve1 low macrophages are Cx3cr1 high. This has been reported by many groups, including ours (please see PMID: 30054204). Therefore, the Cx3cr1 creERT2 would delete more in Lyve1 low than Lyve1 high macrophages. It will also delete in microglia with all potential side and undesirable confounding effects.

b) Some investigators have used the Cx3cr1 creERT2 to deplete resident macrophages or delete selective genes in resident macrophages. However, their experiments were short-term (typically 1 to 4 weeks) and their genes/pathways of interest were not particularly enriched in Lyve1 high macrophages. Furthermore, the use of Cx3cr1 creERT2 was associated with substantial undesirable effects on animal health. For example, when Epelman/Lavine et al (PMID: 34363749) used the Cx3cr1 creERT2/DTR to deplete resident macrophages in the heart (in fact resident macrophages were depleted everywhere, including the brain) in a model of short-term AngII infusion (4 weeks), they had to repeat DT injection every 5 days to be able to sustain macrophage depletion, and the mice suffered from important loss of body weight. We expect that the use of such a model for several weeks in a chronic setting like atherosclerosis would be associated with even more undesirable effects. Epelman/Lavine also used the Cx3cr1 creERT2 to delete Igf1 in the same AngII model, but their own data indicate that Igf1 and the “regulation of IGF1-R signaling” were not enriched in Lyve1 high cardiac macrophages but were more enriched in other resident macrophage subsets. This may explain why they were successful in deleting Igf1 in resident cardiac macrophages using the Cx3cr1 creERT2. In our case, NRP1 is highly enriched in arterial Lyve1hi macrophages and the use of Cx3cr1 creERT2 would not be optimal.

c) In the case of Spic, we would face other undesirable issues. The use of Cx3cr1 cre will deplete red pulp macrophages in the spleen and this may lead to severe alteration of iron metabolism which may affect atherosclerosis development and confound data interpretation. However, it is known that LysM-mediated deletion of Spic does not affect red pulp macrophages (see PNAS paper PMID: 30061415).

d) We note that resident arterial macrophages are of both EMP and monocyte origins, even in healthy arteries. In fact, existing data show that EMP-derived and monocyte-derived TLF macrophages are transcriptionally extremely similar, even after arterial

injury (see Schulz et al PMID: 32917889). In atherosclerosis, a substantial percentage of resident TLF-like macrophages has a monocyte-derived contribution and reside in the intima (see PMID: 29545365). Therefore, although it would be interesting to address the question of the specific role of EMP-derived versus monocyte-derived resident TLF-like macrophages in atherosclerosis, we feel and we hope that the reviewer will agree that this is out of the scope of the present work.

e) Finally and importantly, the Cx3cr1 creER is a knock-in/knock-out and therefore lacks endogenous CX3CR1 expression. While this may not affect fate-mapping experiments, the absence of CX3CR1 expression becomes problematic when CX3CR1 plays a non-redundant role in the disease process being examined. Indeed, CX3CR1 plays a major role in atherosclerosis, and our group has previously shown that deletion of only one Cx3cr1 allele (Cx3cr1 heterozygosity as in the Cx3cr1 creER) substantially reduces the development of atherosclerosis, as in the total Cx3cr1 knockout (Circulation PMID: 12600915). We had also highlighted the fact that this finding is supported by human data showing that CX3CR1 M280 heterozygosity is associated with reduced severity of atherosclerosis (Circ Res PMID: 11532900) and a markedly reduced risk of acute coronary ischemic events (Blood PMID: 11264153). Overall, the available data indicate that the use of Cx3cr1 creER is not appropriate to study the process of atherosclerosis and is likely to introduce major bias in data interpretation.

Given all the above, we hope the reviewer will agree that repeating the experiments with the Cx3cr1CreERT2 strain is not essential for the conclusions of this manuscript. In order to be responsive to the Reviewer's point, we tempered our conclusions. We changed the wording to "resident-like" macrophages where necessary and have clearly indicated in the Discussion that "additional work will be required to assess the differential roles of monocyte-, yolk sac- or erythro-myeloid progenitor-derived resident macrophages in this process". We have also tempered our conclusions regarding the use of the Lyz2 versus the Cx3cr1 cre drivers and added this to study limitations.

2. Reviewer: The authors central hypothesis is that NRP1 loss led to impaired efferocytosis in vivo. The authors have cited a number of other studies that circumstantially make this link in atherosclerosis, which is important justification to explore in their work - but proof is needed as to the mechanism in their hands and in their mice. Is it really intimal and/or adventitial MFs that cannot efferocytosis properly that drive the phenotype, because these cells were never assessed? Why not sort these cells and test - its is a standard approach. The authors need to address this experimentally, in vivo. Ideas that come to mind would be isolate MFs from the adventitia and intima (WT vs KO) and see before iWD, are they impaired, after iWD? these types of experiments would go a long way in proving what happens in the plaque. In vivo or ex vivo uptake studies can be done then.

Response: We hope the reviewer will agree that it is technically challenging to isolate macrophages separately from intima and adventitia and put them in culture to assess efferocytosis. We have already presented data on altered efferocytosis in the intima of plaques. Nevertheless, to answer the reviewer's request, we have measured

efferocytosis in vivo in the adventitia of the same arteries. TUNEL positivity was much less prevalent in the adventitia compared to the intima (which is expected) and we found no significant difference between the 2 groups in the efferocytosis index in the adventitia (see figure below). Overall, our data indicate that deletion of LYVE1+ macrophages has a predominant impact on efferocytosis in the intima compared to the adventitia.

Please, note that while it is widely accepted that defective efferocytosis in the intima promotes plaque progression, the impact of defective efferocytosis in the adventitia on plaque progression is not known, and therefore it would not be possible to mechanistically link differences in efferocytic capacity in the adventitia to differences in disease progression.

3. Reviewer: The authors conduct important experiments using Lyve1Cre-csf1r f/f mice, which constitutive delete LYVE1 MFS (adventitial and probably intimal). The presentation of the data in the supplement is in a non standard way, separating control vs Lyve1Cre - so no direct comparison is possible. The authors compare control mice to each other and Cre mice to each other. There is variability in the data and my best guess is there is weak data here to support their conclusion that resident MFs are involved in iWD and not in cWD. If all 4 groups were combined together, there would need to repeat experiments to get enough power to know if this is a specific effect in iWD. My guess is there is not, plaque area is blunted in both cWD and iWD with loss of LYVE1 macrophages - the authors need to expand these studies.

Response: The new data we presented in Extended data Fig. 9 (now Figure 3 of the revised manuscript) are original and further support our hypothesis in an additional model of atherosclerosis (AAV/PCSK9). We show that in this additional model, iWD accelerates atherosclerosis and increases necrotic core size compared to cWD in wild type mice, but fails to do so when the mice are deleted for Lyve1+ macrophages. Our results are robust (P values for differences in plaque size and necrotic cores are 0.0016 and 0.0056 in the presence of Lyve1 macrophages, but move to non-significance P=0.30 and 0.19 in the absence of Lyve1 macrophages).

We have re-analysed the data grouping them together as requested by the reviewer (see below) to examine if Lyve1+ macrophage deletion accelerates atherosclerosis under cWD. The data is again robust and show that indeed Lyve1+ macrophage deletion accelerates atherosclerosis and increases necrotic core under cWD (P values 0.0157 and 0.0259, respectively). However, Lyve1+ macrophage deletion does not accelerate atherosclerosis nor increases necrotic core under iWD (P values 0.882 and 0.4119, respectively), which is in line with the fact that Lyve1+ macrophages are already altered under iWD and their deletion has no additive effects under this setting. Thus, regardless of how we analyse the data, the results are strong and show that deletion of Lyve1+ macrophages accelerates atherosclerosis and increases necrotic core size under cWD but not under iWD.

We would also like to emphasise the fact that we have already presented another set of supportive data in the manuscript and showed that deletion of Lyve1 macrophages accelerates atherosclerosis and increases necrotic core size under cWD using the *ApoE^{-/-}* model. Thus, we have several models of atherosclerosis in which we show very consistent results.

Reviewer Reports on the Second Revision:

Referees' comments:

Referee #4 (Remarks to the Author):

This manuscript explores the early life hyperlipidemia on atherosclerotic cardiovascular disease risk later in life through a series of animal models and epidemiological studies. Although the authors explain briefly the methods of the Young Finns Study, important details are missing in the epidemiological component of the paper.

Study design:

- The authors restricted the analysis to participants who attended either 2001 or 2007 follow-ups of the study. This leads to selection bias in the study. In other words, the analytical sample will be biased if any of the causal factors for the outcome have influenced participants in early dropout (e.g., death due to ASCVD) or refused to participate in 2001 or 2007 follow-up. The impact of this selection bias needs to be discussed in the paper.
- From the Young Finn Study website, it is evident that a follow-up was performed in 2010-2012. The motivation to not include this follow-up is not clear.

Analysis:

- How were the potential confounders for Model 2 chosen? Although the authors list the variables adjusted for in the model, there is no justification for the same. Directed acyclic graphs for the selection of confounders are needed to understand the potential biases induced by wrong adjustments. This has become a standard of practice in modern epidemiology.
- Sample size calculation: For one of the primary outcomes, the presence of carotid plaque, there are only 88 events in the analytical dataset. This low sample size has led to a lack of precision in many of their estimates. Importantly, the relative weights are estimated in BRLM.
- Moreover, the models for the carotid plaque area have only 88 participants in total. This further reduces the power to identify the appropriate life course hypothesis.
- Furthermore, there seems to be a reduction in sample size for Model 1 vs Model 2. Possibly due to missing values in covariates adjusted for in Model 2. This issue needs to be discussed in the paper, as well as the impact of these missing participants on the results. Multiple imputation techniques or a Full Bayesian imputation approach could be taken to assess the sensitivity of the results to the missing observation.
- Bayesian model fitting must be substantiated with model convergence diagnostic (trace plots, R_{hat} values, and other HMC diagnostics). How many chains were run, for how long, etc, needed to be reported.

Results:

- Given that the Young Finns study is a cohort study, risk ratios(RR) need to be reported instead of odds ratios (OR). It is well known that ORs may overestimate RRs in cohort studies. The magnitude of over-

estimation may depend on many factors, including rarity of the outcome.

- Effect measures (ORs and beta) are interpreted in terms of every standard deviation change in the non-HDL-C values. This interpretation is not clinically meaningful. To understand the effect measures appropriately, they need to be translated to the original unit of measurement (mmol/l) of non-HDL-C. In other words, what is the effect of x mmol/l non-HDL-C levels on the outcomes?
- Table 1 presents posterior summaries of the weights. Although mean and credible intervals are useful summaries, the marginal credible intervals can be misleading for weights in BRLM (simplex variable). Instead, it is highly recommended to visualize the posterior distribution. In the case of three life periods of interest, this can be done as ternary plots.
- Posterior probabilities of the life course hypothesis could also be calculated to better inform the interpretation. This can be done as the proportion of posterior samples that lie in the region of interest. For example, the probability that the third period (21 to 24 years of age) is a sensitive period for the association between non-HDL-C and carotid intima thickness.

Interpretation:

- Interpreting the value of effect measures in the text is highly recommended. Simple associated or not-associated is not informative to the reader.
- Moreover, the clinical meaningfulness and conclusiveness of the evidence from the Young Finn Study need to be commented on. For example, how much of an increase in the risk of developing a plaque is clinically meaningful?
- Interpretation on line 431 (page 18): "This strongly suggests that exposure to cholesterol early in life is a major determinant of atherosclerosis in mid-adulthood" seems to be an overstatement. The estimates of weights show that exposure measured closer to the outcome (young adulthood) had more relative importance of CIMT at mid-adulthood. Not a surprising result.
- Clarity is needed in the age translations used to corroborate the results between the animal model and human epidemiological study design. For example, what age periods in humans would correspond to the 10th week to the 16th week of the animal? If one takes the age translation used in anti-tumour research (<https://www.sciencedirect.com/science/article/abs/pii/S0024320519311701>), 10 to 16-week-old mice correspond to 20 to 25-year-old humans. However, based on this translation, the animal model results and human epidemiological results are contradicting. Animal models clearly show that this period is not as important compared to earlier periods. However, results from the Young Finns study show that the period from 21 years to 24 years could be potentially a sensitive period.
- This confusion is present in how early life is defined in the interpretation of results. There needs to be specificity (as in age) when interpreting the animal and human study results together.

Author Rebuttals to Second Revision:

Takaoka et al-Nature manuscript 2023-03-04115B

Responses to REFEREE #4

This manuscript explores the early life hyperlipidemia on atherosclerotic cardiovascular disease risk later in life through a series of animal models and epidemiological studies. Although the authors explain briefly the methods of the Young Finns Study, important details are missing in the epidemiological component of the paper.

RESPONSE

Thank you for providing these detailed comments. We acknowledge that the original submission did not detail all elements of the epidemiological methods used as the primary focus of the manuscript was on the animal models. However, in lieu of your comments we have made changes to the epidemiological component of the paper (see following). We believe this provides the necessary clarity and justification the reviewer is seeking and has strengthened the epidemiological component of this work.

Study design:

- The authors restricted the analysis to participants who attended either 2001 or 2007 follow-ups of the study. This leads to selection bias in the study. In other words, the analytical sample will be biased if any of the causal factors for the outcome have influenced participants in early dropout (e.g., death due to ASCVD) or refused to participate in 2001 or 2007 follow-up. The impact of this selection bias needs to be discussed in the paper.

RESPONSE

Loss to follow-up in the Young Finns Study has been minimal compared with similar studies (Dwyer et al. *Int J Epidemiol.* 2013 Feb;42(1):86-96. doi: 10.1093/ije/dys004) and has largely been non-differential (Raitakari et al. *Int J Epidemiol.* 2008 Dec;37(6):1220-6. doi: 10.1093/ije/dym225). Please note (see following comment) that we have added analyses using outcome data collected in 2018 which provides a different subset of participants in each analysis as the cohort is dynamic—i.e., non-participants at one follow-up are invited to attend subsequent follow-ups. The results are largely consistent when using either sample.

Further, we compared baseline characteristics of participants versus non-participants at the age 2018 and 2001/07 follow-ups in our analyses and found they were similar statistically (P values >0.05) and clinically except that participants were more likely to be female at both outcome time-points (2018: 55 % in participants vs. 46 % in non-participants, $P<0.001$; 2001/07: 54 % in participants vs. 41 % in non-participants, $P<0.001$) and have a family history of premature CVD in 2001/07 (13 % in participants vs. 5 % in non-participants, $P=0.002$). Importantly, values of non-HDL-C at baseline were very similar between participants (3.74 mmol/l) vs. non-participants (3.71 mmol/l), after accounting for the above sex difference. Therefore, we hold the opinion that bias due to differential loss to follow-up is not a substantial concern in our data. However, we acknowledge this as a limitation in the Discussion (page 19).

- From the Young Finn Study website, it is evident that a follow-up was performed in 2010-2012. The motivation to not include this follow-up is not clear.

RESPONSE

While an additional follow-up was conducted in 2010-12, it did not include carotid ultrasound measurements. Hence, we used data from the 2001 and 2007 follow-ups when these data were collected. However, since we first submitted this manuscript, carotid ultrasound data collected as part of the most recent survey conducted in 2018 have been

measured and made available for analysis. As such, we have re-analysed our models using these 2018 data. The results are consistent with our initial results using the 2001/07 data however they provide longer length of follow-up, an increased number of cases with carotid plaque (almost 10 times more), and narrower credible intervals on our estimates, which directly address some of your below concerns.

The following changes were made to the manuscript:

- We have updated Table 1 in the revised manuscript to include the 2018 data and relocated the 2001/07 data to the supplement.
- We updated the methods (page 38) of the revised manuscript to outline all follow-ups in the Young Finns Study and the primary (2018) and secondary (2001/07) samples used for the analyses (pages 38-39).

We believe the inclusion of the 2018 data both addresses the concerns raised and strengthens the epidemiological content of the manuscript.

Analysis:

- How were the potential confounders for Model 2 chosen? Although the authors list the variables adjusted for in the model, there is no justification for the same. Directed acyclic graphs for the selection of confounders are needed to understand the potential biases induced by wrong adjustments. This has become a standard of practice in modern epidemiology.

RESPONSE

While we appreciate the utility of DAGs in clarifying the causal pathway and selection of confounders in epidemiological research, we opted for a theoretical justification based on extensive knowledge of the cohort and established literature. Our choice of covariates for model 2 is based on empirical evidence suggesting their potential influence on both the exposure (non-HDL-C levels) and the outcome (carotid plaque), while not being mediators in the pathway. Specifically, we adjusted for sex, year of birth, BMI, fasting glucose, physical activity, smoking status, family history, and education because each can theoretically influence lipid profiles and atherosclerotic outcomes. Additionally, we included systolic BP not as a traditional confounder but because its associates with the outcome and can cluster with lipid disorders. By adjusting for SBP we account for this possibility.

We acknowledge the concern regarding the adjustment for triglycerides in model 2. Given that triglycerides are a major component of VLDL-C, which is included in the non-HDL-C measurement, adjusting for triglycerides could potentially lead to overadjustment. Therefore, we have decided to remove triglycerides from model 2 to avoid obscuring any effects that are intrinsic to non-HDL-C levels.

The following changes were made to the manuscript:

- We have removed triglycerides as a covariate in our model 2 (all Table footnotes have been updated).
- Additional text has been added to the Methods, now outlining the minimally (Model 1) and fully (Model 2) adjusted models and the covariates (page 41).
- The results have been updated following the removal of triglycerides as a covariate.

- Sample size calculation: For one of the primary outcomes, the presence of carotid plaque, there are only 88 events in the analytical dataset. This low sample size has led to a lack of

precision in many of their estimates. Importantly, the relative weights are estimated in BRLM.

RESPONSE

As mentioned above, we have incorporated additional data from the 2018 follow-up of the Young Finns Study, which has considerably increased the number of cases with carotid plaque. The prevalence of plaque in this updated dataset is 39.6 % in 2018 (817 of 2062 participants) compared to 3.3 % (88 of 2643 participants) in the earlier 2001/07 data. This substantial increase in cases has improved the statistical power of our analyses and led to more precise estimates, as evidence by the narrower credible intervals in the updated results (revised Table 1). We now present the 2018 data as the primary dataset in Table 1, while the earlier 2001/07 data are included in the supplement. This update both addresses your concern about the initial case numbers and provides more reassurance on the findings given the constancy between the 2018 and 2001/07 results.

- Moreover, the models for the carotid plaque area have only 88 participants in total. This further reduces the power to identify the appropriate life course hypothesis.

RESPONSE

Please refer to our response above.

- Furthermore, there seems to be a reduction in sample size for Model 1 vs Model 2. Possibly due to missing values in covariates adjusted for in Model 2. This issue needs to be discussed in the paper, as well as the impact of these missing participants on the results. Multiple imputation techniques or a Full Bayesian imputation approach could be taken to assess the sensitivity of the results to the missing observation.

RESPONSE

Yes, the reduction in sample size from Model 1 to Model 2 was due to missing data for some covariates. While we recognize the potential value of imputation procedures, we were limited by the constraints of the BRLM, which we have found does not readily accommodate such methods. Although this was outlined as being possible in the original description of the BRLM procedure [Madathil S, et al., *Int J Epidemiol.* 2018;47(5):1623-1635], there was no clear guidance on it, or coding, in the paper. We previously reached out to the first author of that work for guidance on incorporating imputation procedures but received a vague response on this matter. We are working on a solution, but it is not yet ready. In the interim, we did use the Individual Growth Curve model to interpolate missing non-HDL-C data at the individual level across the observed age points, which provided a partial mitigation of missing non-HDL-C data. We recognize that this does not fully address the potential bias introduced by missing data, however we have noted the reason for the reduction in sample size in the Methods (page 41) and we mention this as a limitation in the Discussion.

- Bayesian model fitting must be substantiated with model convergence diagnostic (trace plots, Rhat values, and other HMC diagnostics). How many chains were run, for how long, etc, needed to be reported.

RESPONSE

These are now reported in Tables or in the supplement (specifically model 2 for the 2018 outcome analyses).

Results:

- Given that the Young Finns study is a cohort study, risk ratios(RR) need to be reported instead of odds ratios (OR). It is well known that ORs may overestimate RRs in cohort studies. The magnitude of over-estimation may depend on many factors, including rarity of the outcome.

RESPONSE

While we acknowledge the preference for risk ratios in epidemiological studies, unfortunately, the BRLM was not formulated for risk ratios, hence our use of odds ratios. It is important to note that ORs can approximate risk ratios under certain conditions, particular when either the outcome is rare or the magnitude of the effect size is modest (Davies HTO, Crombie IK, Tavakoli M. When can odds ratios mislead? *BMJ*. 1998;316(7136):989-91). In our study, the prevalence of carotid plaque is 3.3 % in 2001/07 (<10 % is considered a rare outcome). Additionally, both the 2001/07 and 2018 datasets show modest effect sizes, suggesting that the ORs we report likely provide a reasonable approximation of risk ratios. Therefore, our estimates likely reasonably approximate risk ratios. We have taken care to ensure that we do not interpret our ORs as direct risks. Instead, we present them strictly as measures of association, reflecting the likelihood of outcomes relative to exposures.

- Effect measures (ORs and beta) are interpreted in terms of every standard deviation change in the non-HDL-C values. This interpretation is not clinically meaningful. To understand the effect measures appropriately, they need to be translated to the original unit of measurement (mmol/l) of non-HDL-C. In other words, what is the effect of x mmol/l non-HDL-C levels on the outcomes?

RESPONSE

The “standard deviation” (SD) refers to the variation in non-HDL-C at each individual age-point (or life-stage) where these are referred to in the text. We transformed non-HDL-C values into Z-scores specific to age and sex to account for the physiological alterations that occur in non-HDL-C associated with age and sex across the life-course. This means that a per unit increase in absolute non-HDL-C (mmol/l) is not equivalent across various studied age points and life-stages. The non-HDL-C distribution varies by age and sex, and our method accounts for these variations. This is conceptually important when studying relative contributions in the context of our analyses because if we did not standardize non-HDL-C values, we would run the risk of assigning a higher relative proportion of the total effect to ages or life-stages where an absolute increase in non-HDL-C accounts for a greater amount of the distribution. As such, using the standardized distribution offers a uniform metric for comparison across age points and life-stages. For the above reasons, we maintain our use of this approach in the primary analysis in the revised manuscript, but we have provided further explanation and justification of it.

Further, while we advocate for our primary approach (as we detailed above) on a conceptual basis, we agree that the clinical interpretation could be aided by using absolute non-HDL-C values and units that are clinically relevant. Therefore, we repeated our analyses using a per 1 mmol/l increment in non-HDL-C. These results are included in the Supplement Tables 18, 20, 22 and 24.

In the revised manuscript, we have made the following additions:

-Main text, Statistics and reproducibility, page 40, further reasoning added for non-HDL-C standardization: “In our primary analyses we show the association for a one standard deviation increase in non-HDL-C z-score to account for variations in non-HDL-C that naturally occur with age and sex. We have done this to ensure a uniform metric for

comparison across each life-stage. However, we also provide data for a per 1 mmol/l increase in non-HDL-C in the supplement, which might be more clinically relevant.”

In summary, we have maintained our original methodological approach, given its conceptual importance, while also integrated an additional analysis using clinically relevant non-HDL-C units to enhance interpretability.

- Table 1 presents posterior summaries of the weights. Although mean and credible intervals are useful summaries, the marginal credible intervals can be misleading for weights in BRLM (simplex variable). Instead, it is highly recommended to visualize the posterior distribution. In the case of three life periods of interest, this can be done as ternary plots.

RESPONSE

Ternary plots for the main analyses have been added to the manuscript in Supplementary Methods.

- Posterior probabilities of the life course hypothesis could also be calculated to better inform the interpretation. This can be done as the proportion of posterior samples that lie in the region of interest. For example, the probability that the third period (21 to 24 years of age) is a sensitive period for the association between non-HDL-C and carotid intima thickness.

RESPONSE

We have provided this information in the supplement.

Figure: Comparison of posterior probabilities between life stages for A) carotid plaque presence and B) carotid plaque area.

In these models that examine the association between non-HDL-C and the presence of carotid plaque in 2018, there was a 4% probability that the relative importance of non-HDL-C exposure during childhood surpassed that of adolescence and young adulthood combined. Additionally, the probability that the impact of non-HDL-C exposure during adolescence exceeded the combined influences from childhood and young adulthood was 29%. There was a 40% probability that the relative importance of non-HDL-C exposure in young adulthood was greater than that from both childhood and adolescence combined. In the BRLMs examining the association between non-HDL-C and carotid plaque areas in 2018, there was a 7% probability that the relative importance of non-HDL-C exposure during childhood surpassed that of adolescence and young adulthood combined. Additionally, the probability that the impact of non-HDL-C exposure during adolescence exceeded the combined influences from childhood and young adulthood was 55%. There was a 10% probability that the relative importance of non-HDL-C exposure in young adulthood was greater than that from both childhood and adolescence combined.

Interpretation:

- Interpreting the value of effect measures in the text is highly recommended. Simple associated or not-associated is not informative to the reader.

RESPONSE

We have used collective language in the Results to streamline the text, but all data are shown in the Tables for the reader to review. The addition of the metrics in response to the above comments provides more nuance surrounding the interpretation of the life-course models.

- Moreover, the clinical meaningfulness and conclusiveness of the evidence from the Young Finn Study need to be commented on. For example, how much of an increase in the risk of developing a plaque is clinically meaningful?

RESPONSE

Quantifying a clinically meaningful change in the odds of developing carotid plaque is inherently challenging due to the multifactorial nature of atherosclerotic cardiovascular disease. In our primary analyses, we observed an ~50 % increase in the odds for carotid plaque presence in mid-adulthood for each standard deviation increase in early life (child, adolescent, young adulthood combined) non-HDL-C. This increase is likely substantial given that elevated non-HDL-C is just one of many risk factors for the outcome, and that the exposure is limited to early life.

In response to an earlier comment, we have included in the supplement the results for a per 1 mmol/l increase in non-HDL-C. Although we maintain the need to show a per standard deviation change on a theoretical basis to determine the best life-course model, we agree that providing results for an absolute unit increase in non-HDL-C offers a more clinically tangible estimate. These results show an ~70% increase in odds per 1 mmol/l increase in the 'lifetime' averaged non-HDL-C levels.

- Interpretation on line 431 (page 18): "This strongly suggests that exposure to cholesterol early in life is a major determinant of atherosclerosis in mid-adulthood" seems to be an overstatement. The estimates of weights show that exposure measured closer to the outcome (young adulthood) had more relative importance of cIMT at mid-adulthood. Not a surprising result.

RESPONSE

To clarify, our statement was specifically based on the analysis of plaque presence and plaque area, which directly represent atherosclerosis, rather than on the cIMT data. As detailed above, we have removed all cIMT data from the manuscript so should not provide this confusion. As such, we have made no change to the manuscript.

However, in consideration of your feedback, we have slightly rephrased this statement to: "This suggests that exposure to cholesterol early in life *contributes to the development of* atherosclerosis in mid-adulthood." This revision aligns with the evidence we provide but acknowledges the observational nature of our epidemiological data.

Moreover, as mentioned above, all cIMT data have been removed from the manuscript to avoid confusion and focus on the more directly relevant measures of atherosclerosis for our conclusions.

- Clarity is needed in the age translations used to corroborate the results between the animal model and human epidemiological study design. For example, what age periods in humans would correspond to the 10th week to the 16th week of the animal? If one takes the age translation used in anti-tumour research (<https://www.sciencedirect.com/science/article/abs/pii/S0024320519311701>), 10 to 16-week-old mice correspond to 20 to 25-year-old humans. However, based on this translation, the animal model results and human epidemiological results are contradicting. Animal models clearly show that this period is not as important compared to earlier periods. However, results from the Young Finns study show that the period from 21 years to 24 years could be potentially a sensitive period. This confusion is present in how early life is defined in the interpretation of results. There needs to be specificity (as in age) when interpreting the animal and human study results together.

RESPONSE

Our aim was not to have an exact translation of ages between mice and humans. This is not possible, and this is why we have not discussed this aspect in the manuscript. As shown in this documentation from the Jackson Laboratories (<https://www.jax.org/research-and-faculty/research-labs/the-harrison-lab/gerontology/life-span-as-a-biomarker>), 6-10 weeks of age in mice (the age at which mice were initiated on iWD) would correspond to childhood/adolescence (6 to 18 years of age in the Young Finns Study). 16-20 weeks of age in mice (the age at which cWD was started) would correspond to the start of mid-adulthood. Thus, the age translation fits rather quite well. Early life in humans is before adulthood, and we have now indicated that what we meant by early life was childhood and adolescence.

Reviewer Reports on the Third Revision:

Referees' comments:

Referee #1 (Remarks to the Author):

My comments on the manuscript have all been answered. The addition of strong epidemiological data supports the overall message of the story. It is important that such relevant research is broadly communicated. I support publication of this manuscript.

Referee #2 (Remarks to the Author):

None

Referee #3 (Remarks to the Author):

The authors have been sufficiently responsive to my comments.

If they cannot do the in vivo efferocytosis experiments, they should caveat their study in the discussion, and state that efferocytosis was not directly assessed in vivo.

Accumulation of dead cells (TUNEL) is a potential surrogate for ingestion, and not a direct measure.

I appreciate retrieving intimal cells is challenging - although can be done, and has been done by various groups. Efferocytosis experiments require very few cells - I do hold proving this to be the case in vivo would markedly improve the study, but leave this decision for the editors.

Referee #4 (Remarks to the Author):

The inclusion of a new wave of the cohort study is a commendable effort and helps clarify the overall picture. However, the conclusion presented in the text and the estimates in the table are contradicting. Furthermore, the results from the tables for the human study contradict the animal study results.

1) Contradicting interpretations

The estimates of relative weights of life periods in Table 1 and the posterior probabilities reported in response to the review show the presence of a sensitive period in young adulthood for Carotid Plaque and Adolescence for Plaque Area. For example, the authors report that there was a 40% posterior probability that the weight estimated for the young adulthood period was higher than the other two periods. Note that the prior probability for this hypothesis is only 33% under a non-informative prior the authors used. The support for the other two periods being sensitive was lower in the posterior, further

clarifying this. Although mentioned in the response that the ternary plots are included in the supplementary methods, I could not find them in any of the submitted documents.

Yet the authors interpreted these results as supportive of the accumulation model (all periods are equally important) in line 397 and on lines 401 and 460 as "[...] exposure to non-HDL-C levels very early in life (childhood and adolescence) still contributed substantially".

And on line 405, "[...] this strongly suggests that exposure to cholesterol early in life contributes to the development of atherosclerosis in mid-adulthood".

The analysis of the 2001 or 2007 outcome measurement models and the 2018 outcome measurement model shows that non-HDL-C exposure closer to the outcome is more relevant than farther away.

Following the rough conversion of animal age to human age (according to the author's response), the animal model results suggest that iWD at childhood and adolescence is more important than mid-adulthood. However, the epidemiological study only investigated the periods of exposure from childhood to young adulthood, which (as elaborated above) shows that exposure in young adulthood is more important than the other periods.

Interestingly, although reported in the response that the main manuscript has been modified as "This suggests that exposure to cholesterol early in life contributes to the development of atherosclerosis in mid-adulthood", this sentence does not appear in the main text.

2) Dropping of cIMT results:

The cIMT results are dropped from this submission. This seems to be an odd choice as all of those results are also pointing to the same conclusions as elaborated above. The justification given in the response to the review section that this was done to avoid confusion is not sufficient. Yet, the practice of dropping a finding post-hoc because it may or may not contradict is not recommended. It is well known that this leads to the issue, popularly known as the "Garden of forking paths." Note that this issue is not limited to the Frequentist approach as explained by Dr. Gelman (http://www.stat.columbia.edu/~gelman/research/unpublished/p_hacking.pdf).

3) Reporting of statistical models used: BRLM is not a statistical model but rather a modelling approach. The original paper by Madathil et al. presented the method using logistic regression. However, the method itself does not define the statistical model. Kindly report the statistical model used for each outcome variable.

4) Reporting Odds ratios instead of Risk ratios: The justification provided for not reporting risk ratios for a cohort study is that the prevalence of outcome is rare in the cohort. However, the data from 2018 shows that the prevalence of outcome is 39%, which cannot be considered rare. The second justification is that the BRLM model does not allow the estimate of a risk ratio. I am afraid that is factually not right. BRLM is not a statistical model but rather a modelling approach and can be included in any statistical model (e.g., logistic regression, linear regression, poisson regression). For example, the authors themselves used some form of continuous outcome model for the plaque area and reported estimates using a log scale. The traditional approach of estimating the relative risk in a cohort study is via a log-binomial or Poisson model (<https://academic.oup.com/aje/article/157/10/940/290159>). Furthermore, it is trivial to calculate the relative risk in any Bayesian logistic regression model (regardless of whether it uses the BRLM approach or not) by computing it from the posterior distributions of the coefficients. I

strongly recommend reporting relative risks.

5) Issue of missing values: The justification for not including a missing value imputation method is that the codes are not available. This is also factually wrong. Dr. Madathil's thesis has reported methods of imputing missing values at

([https://github.com/MadathilSA/BayesianRelevantLifeCourseExposure/blob/master/Manuscript III/HeNCeIndiaSmkModelImput.jags](https://github.com/MadathilSA/BayesianRelevantLifeCourseExposure/blob/master/Manuscript%20III/HeNCeIndiaSmkModelImput.jags)). However, an imputation approach can be considered as a sensitive analysis and if the authors choose to do so. Given the extent of confounding seen in the tables, the results may not change much.

6) Line 1112 – Factually wrong statement. BRLM does allow for interactions. See the code here: [https://github.com/MadathilSA/BayesianRelevantLifeCourseExposure/blob/master/Manuscript IV/BRLM_Protracted_Spline.stan](https://github.com/MadathilSA/BayesianRelevantLifeCourseExposure/blob/master/Manuscript%20IV/BRLM_Protracted_Spline.stan) However, the stratified analysis is justified as the results are clear in the stratified models.

Author Rebuttals to Third Revision:

Nature manuscript 2023-03-04115E-Z

Referee #3 (Remarks to the Author):

The authors have been sufficiently responsive to my comments.

Accumulation of dead cells (TUNEL) is a potential surrogate for ingestion, and not a direct measure. If they cannot do the in vivo efferocytosis experiments, they should caveat their study in the discussion, and state that efferocytosis was not directly assessed in vivo.

I appreciate retrieving intimal cells is challenging - although can be done, and has been done by various groups. Efferocytosis experiments require very few cells - I do hold proving this to be the case in vivo would markedly improve the study, but leave this decision for the editors.

RESPONSE

Regarding efferocytosis, we calculated an internalisation index (phagocytosis of apoptotic cells by plaque macrophages), as proposed and done by Ira Tabas and colleagues, and subsequently by many other investigators. However, we fully agree with the reviewer that this remains only a potential surrogate for efferocytosis, and not a direct measure. Direct measure of efferocytosis was done using bone marrow-derived cells.

We agree that, ideally, efferocytosis should be measured directly using plaque-derived macrophages. We would love to be able to do so. However, this is very challenging. We were intrigued by the reviewer's statement that such efferocytosis experiments have been done by various groups. We therefore contacted the world leading experts on efferocytosis in atherosclerosis, the group of Ira Tabas and the group of Nick Leeper, who published most of the major papers on the role of efferocytosis in atherosclerosis, including papers published in Nature (see PMID: 27437576). To our question (by email): "Have you ever done (or do you know of anyone else who has done) efferocytosis experiments on macrophages retrieved from the intima of mouse atherosclerotic lesions?" Both of them (and their group members) replied that they didn't know anyone who has done such an experiment, adding: "If you have ideas about how that could work, I'd love to brainstorm with you". I have then organised a zoom meeting with Ira Tabas, who confirmed to me that, even though this sounds like an interesting experiment (and we all agree on that), it will be a very challenging experiment to put in place and validate.

The number of live intimal macrophages that we retrieve from *ldlr*^{-/-} mice after 6 weeks of Western diet is less than 1000 cells (generally closer to 500 per aorta). Ira Tabas' and Nick Leeper's groups have never carried out an efferocytosis experiment with less than 100,000 (bone marrow-derived or peritoneal) macrophages per well. Even if we go down to 50,000 macrophages per well in 96-well plates (in fact experts do not recommend less than 100,000 cells per well given the need for close cell-cell contacts for such experiments), we would need at least 100 mice for n=1 experimental condition. We hope the reviewer will agree that this is not an easy task.

Given the above, we now understand that the reviewer's statement that "in vivo efferocytosis experiments" using tissue macrophages "have been done by various groups" does not apply to mouse atherosclerotic plaques. We do not dispute that such experiments are feasible using macrophages from other tissues, particularly when the organ is relatively big and harbours a higher number of macrophages.

We reviewed the cardiovascular literature and found that, overwhelmingly, efferocytosis experiments were done using bone marrow-derived or peritoneal macrophages. We found that the group of Slava Epelman have published an in vitro efferocytosis experiment using retrieved cardiac macrophages (PMID: 24439267). From their paper, we can see that the number of macrophages that can be recovered from the mouse heart is at least 10 times more than what we can recover from the intimal layer of the aorta. Unfortunately, the authors did not provide the number of cardiac macrophages used for the in vitro efferocytosis experiment, so we are unable to compare to our case.

In summary, we agree with the reviewer that the efferocytosis experiment on intima-derived macrophages would be ideal. However, we hope the reviewer will agree with us that the experiment is extremely challenging and has never been validated before. As such, we hope the reviewer will agree to waive the experiment. We have added a clear study limitation indicating that efferocytosis was not directly assessed in vivo using plaque-derived macrophages (lines 500-501).

Referee #4 (Remarks to the Author):

The inclusion of a new wave of the cohort study is a commendable effort and helps clarify the overall picture.

Response: Thank you. This was a huge effort from the Young Finns Study team who analysed the recently available 2018 dataset, and provided original data on a very substantial number of outcome events (n=817 compared with n=88 previously), substantially strengthening the validity of our results.

However, the conclusion presented in the text and the estimates in the table are contradicting. Furthermore, the results from the tables for the human study contradict the animal study results.

1) Contradicting interpretations

The estimates of relative weights of life periods in Table 1 and the posterior probabilities reported in response to the review show the presence of a sensitive period in young adulthood for Carotid Plaque and Adolescence for Plaque Area. For example, the authors report that there was a 40% posterior probability that the weight estimated for the young adulthood period was higher than the other two periods. Note that the prior probability for this hypothesis is only 33% under a non-informative prior the authors used. The support for the other two periods being sensitive was lower in the posterior, further clarifying this. Although mentioned in the response that the ternary plots are included in the supplementary methods, I could not find them in any of the submitted documents.

Yet the authors interpreted these results as supportive of the accumulation model (all periods are equally important) in line 397 and on lines 401 and 460 as "[...] exposure to non-HDL-C levels very early in life (childhood and adolescence) still contributed substantially".

And on line 405, "[...] this strongly suggests that exposure to cholesterol early in life contributes to the development of atherosclerosis in mid-adulthood".

The analysis of the 2001 or 2007 outcome measurement models and the 2018 outcome measurement model shows that non-HDL-C exposure closer to the outcome is more relevant than farther away.

Following the rough conversion of animal age to human age (according to the author's response), the animal model results suggest that iWD at childhood and adolescence is more important than mid-adulthood. However, the epidemiological study only investigated the periods of exposure from childhood to young adulthood, which (as elaborated above) shows that exposure in young adulthood is more important than the other periods.

Interestingly, although reported in the response that the main manuscript has been modified as "This suggests that exposure to cholesterol early in life contributes to the development of atherosclerosis in mid-adulthood", this sentence does not appear in the main text.

Response: We apologise for the omission of the ternary plots in the previous submission. We have now included these plots for all models (see Supplementary Methods and Figures).

The ternary plots for the main models indicate higher density toward the upper (adolescence) and right lower (young adulthood) vertices, suggesting higher weights for these periods compared with childhood. However, the credible intervals are wide and overlap across the life periods. Our conclusion for the best life-course model is based on the Euclidian distance, which for most models supported the accumulation hypothesis (see Supplementary Tables 16-22). While some instances showed similar Euclidian distances for a sensitive period vs. accumulation model, most models indicated accumulation. As noted by Ben-Shlomo et al., even if the life-course model is a sensitive period, it is embedded within accumulation (Life course epidemiology. In: Ahrens W, Pigeot I (eds). Handbook of Epidemiology. New York, NY: Springer, 2014).

We would like to re-emphasize that our mouse data indicate that exposure to cholesterol **over a lifetime** (starting early in life) is more relevant to atherosclerotic plaque development than just **late exposure**. Therefore, an accumulation model nicely aligns with our hypothesis. Also, please note that the endpoint in the animal model is atherosclerotic **plaque size**, rather than presence or absence of plaques (all mice in all groups have plaques). Thus, the human data on carotid plaque area is the most relevant. This data fits with an accumulation model, with some suggestion that exposure to non-HDL-C at adolescence was contributing most to the observed lifetime effect (see Table 1 and Supplementary Table 18), which still nicely fits with the animal studies.

Our core point in the previous response to the reviewer was to emphasise that exposure to non-HDL-C at all observed life-stages contributes to mid-adulthood carotid plaque and that even the ‘very early in life’ periods, referring to both childhood AND adolescence, contribute over 50% of the total effect. We acknowledge that this detail was not apparent in our original text and have revised it accordingly (Lines 423-428). We would like to emphasize again that this aligns with the animal model. Please, also note that the whole exposure period from Childhood to Young Adulthood (6 to 24 years), could well be considered as “early” exposure when put in the context of the timepoint of plaque size assessment, which is around the age of 50 in the 2018 visit. In the case of the 2018 visit (41 to 56 years), 17 to 44 years have elapsed since the end of the exposure period. We hope the reviewer will agree that our finding that exposure to non-HDL-cholesterol levels between 6 and 24 years is associated with plaque presence and plaque size assessed at around the age of 50 (even very far from the Young Adulthood period), is quite remarkable, and may have important clinical implications. Indeed, screening and intervention programs for children and adolescents primarily target those with very high cholesterol levels associated with familial hypercholesterolemia. Our data together suggest that screening and prevention efforts in the general population (beyond those with familial hypercholesterolemia) should be occurring much earlier in life (compared to what it is currently done, i.e. generally after the age of 40) to reduce plaque burden associated with cholesterol levels.

However, please note that we are not formulating any recommendation in this regard and we have made every effort to remain factual in our data interpretation.

The previously indicated change that was not included has been made and updated in lieu of the above (Lines 401-428):

Original text: “Despite some variation in the relative contribution of non-HDL-C levels at each life stage, the Euclidian distance and ternary plots consistently indicated that the accumulation life course model best represented the association with mid-adulthood plaque presence and plaque area (Supplementary Tables 17-24, and Supplementary Methods). Data stratification by sex suggested that the associations were slightly stronger for males than for females (Supplementary Tables 17-24). Although exposure to non-HDL-C at each life stage was contributing to the lifetime effect, it is intriguing to note that exposure to non-HDL-C levels very early in life (i.e., childhood and adolescence) still contributed substantially to plaque outcomes in mid-adulthood (Table 1 and Supplementary Table 16). This strongly suggests that exposure to cholesterol early in life contributes to the development of atherosclerosis in mid-adulthood.”

Revised text: “Data stratification by sex suggested that the associations were slightly stronger for males than for females (Supplementary Tables 16, 17, 19-22). For each model, the Euclidian distance generally indicated that the accumulation life course model best represented the association with mid-adulthood plaque presence and plaque area (Supplementary Tables 16, 17, 19-22, and Supplementary Methods and Figures). This was despite some variation in the relative contributions of non-HDL-C levels at each life stage, as shown in Table 1, Supplementary Tables 16-22, and accompanying ternary plots (see Supplementary Methods and Figures), noting that the credible intervals for each life stage were wide and overlapped. Although there was some suggestion that exposure to non-HDL-C at adolescence and young adulthood was contributing most to the observed lifetime effects, exposure to non-HDL-C levels before adulthood (i.e., childhood and adolescence) consistently contributed more than half of the lifetime effect to plaque outcomes in mid-adulthood. This suggests that exposure to cholesterol prior to adulthood contributes to the development of atherosclerosis in mid-adulthood”.

2) Dropping of cIMT results:

The cIMT results are dropped from this submission. This seems to be an odd choice as all of those results are also pointing to the same conclusions as elaborated above. The justification given in the response to the review section that this was done to avoid confusion is not sufficient. Yet, the practice of dropping a finding post-hoc because it may or may not contradict is not recommended. It is well known that this leads to the issue, popularly known as the "Garden of forking paths." Note that this issue is not limited to the Frequentist approach as explained by Dr. Gelman (http://www.stat.columbia.edu/~gelman/research/unpublished/p_hacking.pdf).

Response: We want to clarify that the cIMT findings were not dropped for the reasons mentioned. Our decision to focus on plaque is based on its alignment with our animal model outcomes (atherosclerotic plaque size), not to avoid contradictory results.

While cIMT is associated with carotid plaque, they are distinct phenotypes (Touboul et al. *Cerebrovasc Dis* (2012) 34 (4): 290–296), with cIMT considered an imperfect marker of atherosclerosis given the inclusion of the medial layer in the measurement (Spence, *Atherosclerosis*. 2020; 312: 117-118; Raggi et al. *Atherosclerosis*. 2020; 312: 119-120). In summary, it is now widely accepted that cIMT is NOT atherosclerosis. In this context, adding the cIMT data would only confuse the reader. Initially, both cIMT and plaque data were included due to the limited number of plaques in the 2011 outcome data from the Young Finns Study. However, with the use of the updated 2018 outcome data, plaque prevalence was substantially higher, making additional supplemental analysis with cIMT unnecessary. Given that our mouse models use plaque as the primary atherosclerosis phenotype, we chose to focus on plaque to maintain consistency across human and animal studies.

For transparency, we have included all cIMT results below for your information (Table 1 and Figures 1 and 2 for high cIMT, Table 2 and Figures 3 and 4 for continuous cIMT). The results are largely consistent with the main findings, particularly for plaque presence. We still feel that the cIMT data should not be included in the manuscript, and we hope the reviewer will agree with us. However, if the reviewer and the editors prefer, we can add these findings to the supplemental material.

[REDACTED]

[REDACTED]

[REDACTED]

[REDACTED]

[REDACTED]

[REDACTED]

[REDACTED]

[REDACTED]

[REDACTED]

[REDACTED]

[REDACTED]

[REDACTED]

[REDACTED]

3) Reporting of statistical models used: BRLM is not a statistical model but rather a modelling approach. The original paper by Madathil et al. presented the method using logistic regression. However, the method itself does not define the statistical model. Kindly report the statistical model used for each outcome variable.

Response: Thank you for this clarification. We used linear regression to model the continuous outcomes and Poisson regression for the dichotomous outcomes (also see our response to comment #4). We have updated the manuscript to clearly specify these statistical models with each outcome: Old text: “The Bayesian relevant life course exposure model (BRLM) was used to identify relative contribution of non-HDL-C measured in childhood (aged 6, 9 and 12 years), adolescence (aged 15 and 18 years), and young adulthood (aged 21 and 24 years) on the atherosclerosis outcome variables in mid-adulthood (carotid plaque presence and plaque area).” New text: “The Bayesian relevant life course exposure model (BRLM) was used to identify relative contribution of non-HDL-C measured in childhood (aged 6, 9 and 12 years), adolescence (aged 15 and 18 years), and young adulthood (aged 21 and 24 years) on the atherosclerosis outcome variables in mid-adulthood (carotid plaque presence **with Poisson regression** and plaque area **with linear regression**).” Throughout the manuscript we have also changed any reference to odds ratios to *relative risks* or *risk* given our use of Poisson regression (see our response to comment #4).

4) Reporting Odds ratios instead of Risk ratios: The justification provided for not reporting risk ratios for a cohort study is that the prevalence of outcome is rare in the cohort. However, the data from 2018 shows that the prevalence of outcome is 39%, which cannot be considered rare. The second justification is that the BRLM model does not allow the estimate of a risk ratio. I am afraid that is factually not right. BRLM is not a statistical model but rather a modelling approach and can be included in any statistical model (e.g., logistic regression, linear regression, poison regression). For example, the authors themselves used some form of continuous outcome model for the plaque area and reported estimates using a log scale. The traditional approach of estimating the relative risk in a cohort study is via a log-binomial or Poisson model (<https://academic.oup.com/aje/article/157/10/940/290159>). Furthermore, it is trivial to calculate the relative risk in any Bayesian logistic regression model (regardless of whether it uses the BRLM approach or not) by computing it from the posterior distributions of the coefficients. I strongly recommend reporting relative risks.

Response: Thank you for highlighting this issue. Based on your guidance, we have addressed this by re-running our analyses using a Poisson model for our dichotomous outcomes to estimate relative risks (RR). The lifetime and individual life-period effects remain consistent with our previous findings, indicating that non-HDL-C levels at each life stage contribute to mid-adult plaque. The relative weights have slightly changed, but the life-course hypothesis of accumulation observed in the earlier analyses remains. In the revised manuscript we have updated Table 1 and Supplementary Tables 16-22 using Poisson regression to estimate RRs.

5) Issue of missing values: The justification for not including a missing value imputation method is that the codes are not available. This is also factually wrong. Dr. Madathil's

thesis has reported methods of imputing missing values at ([https://github.com/MadathilSA/BayesianRelevantLifeCourseExposure/blob/master/Manuscript III/HeNCEIndiaSmkModelImput.jags](https://github.com/MadathilSA/BayesianRelevantLifeCourseExposure/blob/master/Manuscript%20III/HeNCEIndiaSmkModelImput.jags)). However, an imputation approach can be considered as a sensitive analysis and if the authors choose to do so. Given the extent of confounding seen in the tables, the results may not change much.

Response: Thank you for pointing out the availability of imputation approaches within the BRLM framework. Following the reviewer's guidance, we have now included methods to impute missing data and presented these results as a sensitivity analysis (see Supplementary Table 18, and Supplementary Methods and Figures). Additional text has been added to the Methods section (Lines 1122-1129) and to the supplement outlining our approach (see Supplementary Methods and Figures): "Methods for imputing missing values have been described ([https://github.com/MadathilSA/BayesianRelevantLifeCourseExposure/blob/master/Manuscript III/HeNCEIndiaSmkModelImput.jags](https://github.com/MadathilSA/BayesianRelevantLifeCourseExposure/blob/master/Manuscript%20III/HeNCEIndiaSmkModelImput.jags)). Multiple imputation was undertaken for the covariates of youth age, ever smoked, family history of cardiovascular disease, education (number of years studied), and cumulative body mass index, systolic blood pressure and high-density cholesterol, using predictive mean matching via the R library, mice. Three imputations were made, and the means and credible intervals were calculated using the draws across all imputations."

As the reviewer expected, we found minimal changes to our findings with imputation of missing values but observed slightly narrower credible intervals and a stronger tendency toward a sensitive exposure period in adolescence for the model with plaque area (see Supplementary Table 18) compared to the complete case analyses, which is still consistent with the animal studies.

6) Line 1112 – Factually wrong statement. BRLM does allow for interactions. See the code here:https://github.com/MadathilSA/BayesianRelevantLifeCourseExposure/blob/master/ManuscriptIV/BRLM_Protracted_Spline.stan However, the stratified analysis is justified as the results are clear in the stratified models.

Response: Thank you for pointing out this error. We appreciate the opportunity to correct it. In the revised manuscript, we have removed the incorrect statement about the BRLM not allowing for interactions (Lines 1290-1). Old text: "As the BRLM does not allow for interaction, all models were additionally performed stratified by sex." New text: "All models were additionally performed stratified by sex." As the reviewer indicates, our stratification remains sufficient to address this in the analysis.

Reviewer Reports on the Forth Revision:

Referees' comments:

Referee #1 (Remarks to the Author):

I support publication of this important work.

Referee #2 (Remarks to the Author):

None

Referee #3 (Remarks to the Author):

The authors have explored my point of efferocytosis through discussions with Ira Tabas, that is reasonable.

What i had meant by checking in vivo - was to isolate the cells from the aortas, put them with labelled target dying cells for a few hours and report the findings. If they were unable to do it (ie no cells survived or could be detected), that is fine, but I suspect if 2500 cells were isolated, they would lasts a few hours in culture to do the experiment.

Referee #4 (Remarks to the Author):

The authors have sufficiently justified their choices with modelling and have addressed all my concerns with updated models, imputation approach and corrected interpretations.

Author Rebuttals to Forth Revision:

Nature manuscript 2023-03-04115E-Z

Referee #3 (Remarks to the Author):

The authors have explored my point of efferocytosis through discussions with Ira Tabas, that is reasonable.

What i had meant by checking in vivo - was to isolate the cells from the aortas, put them with labelled target dying cells for a few hours and report the findings. If they were unable to do it (ie no cells survived or could be detected), that is fine, but I suspect if 2500 cells were isolated, they would last a few hours in culture to do the experiment.

RESPONSE

We note that unfortunately, the reviewer still does not provide any evidence from the literature that such an experiment is feasible and reproducible using 2500 cells. We have already provided a study limitation statement in the manuscript.